# Autophagy induction promoted by m⁶A reader YTHDF3 through translation upregulation of FOXO3 mRNA

WeiChao Hao [1,2,11], MeiJuan Dian[3,4,11], Ying Zhou[2,4], QiuLing Zhong[5], WenQian Pang[2], ZiJian Li[2], YaYan Zhao[1], JiaCheng Ma[6], XiaoLin Lin[2,7], RenRu Luo[8], YongLong Li[2,4], JunShuang Jia[2], HongFen Shen[2], ShiHao Huang[2,4], GuanQi Dai[2,4], JiaHong Wang [2] ✉, Yan Sun [9] ✉ & Dong Xiao [2,4,10] ✉

Autophagy is crucial for maintaining cellular energy homeostasis and for cells to adapt to nutrient deficiency, and nutrient sensors regulating autophagy have been reported previously. However, the role of eiptranscriptomic modifications such as m⁶A in the regulation of starvation-induced autophagy is unclear. Here, we show that the m⁶A reader YTHDF3 is essential for autophagy induction. m⁶A modification is up-regulated to promote autophagosome formation and lysosomal degradation upon nutrient deficiency. METTL3 depletion leads to a loss of functional m⁶A modification and inhibits YTHDF3-mediated autophagy flux. YTHDF3 promotes autophagy by recognizing m⁶A modification sites around the stop codon of FOXO3 mRNA. YTHDF3 also recruits eIF3a and eIF4B to facilitate FOXO3 translation, subsequently initiating autophagy. Overall, our study demonstrates that the epitranscriptome regulator YTHDF3 functions as a nutrient responder, providing a glimpse into the post-transcriptional RNA modifications that regulate metabolic homeostasis.

Autophagy is an evolutionarily conserved mechanism for eukaryotic cells to maintain homeostasis and renewal. Morphologically, autophagy is characterized by the formation of transient double-membrane vesicles called autophagosomes, which engulf cytoplasmic components and transport them into lysosomes for degradation[1]. Autophagy plays a critical role in eliminating misfolded or aggregated proteins in an orderly way, clearing damaged or unnecessary organelles, and allowing degraded cytoplasmic components to be recycled, thereby acting as a cytoprotective system[2]. Autophagy regulates cellular processes vital to life, including maintaining stemness[3], adjusting embryonic development[4], shaping innate cellular immunity[5], and impacting ageing and longevity[6]. Autophagy dysfunction contributes to cancer development[7], neurodegenerative diseases[8], aberrant inflammation[9], various metabolic disorders[10], and decreased lifespan[11]. Traditionally, autophagy is considered a set of cytoplasmic events at the protein level; in recent years, however, compelling evidence has

[1]Department of Oncology, The First Affiliated Hospital of Guangdong Pharmaceutical University, 510080 Guangzhou, China. [2]Cancer Research Institute, School of Basic Medical Sciences, Southern Medical University, 510515 Guangzhou, China. [3]Department of Thoracic Surgery, Nanfang Hospital, Southern Medical University, 510515 Guangzhou, China. [4]Institute of Comparative Medicine & Laboratory Animal Center, Southern Medical University, 510515 Guangzhou, China. [5]Department of Neurobiology, School of Basic Medical Sciences, Southern Medical University, 510515 Guangzhou, China. [6]Tsinghua-Peking Center for Life Sciences, School of Life Sciences, Tsinghua University, 10084 Beijing, China. [7]Cancer Center, Integrated Hospital of Traditional Chinese Medicine, Southern Medical University, 510315 Guangzhou, China. [8]School of Medicine, Shenzhen Campus of Sun Yat-sen University, 518107 Guangdong, China. [9]Guangdong Provincial People's Hospital, Guangdong Academy of Medical Sciences, 510080 Guangzhou, China. [10]National Demonstration Center for Experimental Education of Basic Medical Sciences, Southern Medical University, 510515 Guangzhou, China. [11]These authors contributed equally: WeiChao Hao, MeiJuan Dian. ✉e-mail: wjh1987@smu.edu.cn; sunyan@gdph.org.cn; xiaodong@smu.edu.cn

revealed that nuclear transcription and epigenetic regulations also have profound impacts on this process[12]. Recent breakthroughs discovered that widespread post-transcriptional RNA modifications can dictate cellular function and cell fate, which led to an exciting exploration of the frontiers of epitranscriptomics and revealed a novel layer of gene regulation[13,14]. Yet, whether and how autophagy is regulated at the level of epitranscriptomics remains elusive.

Autophagy plays a central role in how cells accommodate metabolic shifts caused by nutrient deficiency and other metabolic perturbations[15]. To date, some nutrient responders have been shown to regulate autophagy, thereby maintaining cellular metabolism and energy homeostasis[15,16]. For instance, the kinase activity of AMPK is boosted when a cell's energy charge decreases, further stimulating autophagy and regulating metabolic circuits through multiple mechanisms[15,17,18]. mTORC1 senses amino acid deficiency and responds by reversing the inhibitory phosphorylation of ULK1 and subunits of the PIK3C3/VPS34 complex[17,19]. EIF2AK4/GCN2 is activated by amino acid starvation and promotes the translation of ATF4, transactivating many genes involved in autophagy and the integrated stress response[20]. Transcription factor TFEB is dephosphorylated and translocated to the nucleus in response to nutrient deficiency, promoting the transcription of lysosomal and autophagy-related genes[21,22]. The histone deacetylase SIRT1 is activated in nutrient-free conditions and triggers autophagy by activating FOXO transcription factors and core autophagy genes[15,23,24]. These nutrient responders coordinate changes in various molecular mechanisms, which involve nutrient-sensing kinases, transcription, and histone modifications, to ensure an autophagic response and adaptation to metabolic changes. However, no epitranscriptome player has been identified as a nutrient responder to activate autophagy.

A series of studies revealed apparent dynamic changes in epitranscriptomics during nutrient deficiencies and other cellular stress responses, raising the possibility that such changes are closely tied to the modulation of autophagy. For instance, m$^1$A in mRNA exhibits enrichment in the 5'UTR, and near the start codon, in response to nutrient starvation and peroxide stress[25]. Internal mRNA m$^7$G is enriched in the CDS and 3'UTRs, while also depleted in 5'UTRs, upon heat shock and oxidative stress[26]. m$^6$A in the mRNA 5'UTR is induced in response to heat shock and permits selective cap-independent mRNA translation[27,28]. Heat shock or peroxide treatment induces pseudouridine (Ψ) sites and enhances transcript stability[29,30]. Therefore, we supposed that the dynamic reversibility of RNA modifications enables rapid gene regulation in response to changing nutrient cues, that this has a profound impact on autophagy, and that some epitranscriptome players may have a key role in this process.

In this work, we profiled global proteome changes in mouse embryonic fibroblasts (MEFs) with nutrient starvation. In searching for potential epitranscriptome players that might play critical roles in regulating autophagy, we noticed that YTHDF3, an m$^6$A reader, was significantly upregulated during nutrient deficiency. We further characterized how YTHDF3 served as a nutrient responder by regulating autophagy, providing insights into the paradigm that post-transcriptional RNA modifications response to nutrient deficiency stress via controlling autophagy.

## Results

### YTHDF3 up-regulation is required for autophagy induction

To identify the potential epitranscriptome players involved in autophagy, we performed proteomic analysis to screen proteins that were up-regulated due to nutrient deficiency in MEFs. Intriguingly, we found that levels of m$^6$A reader YTHDF3 significantly increased (Fig. 1a). We used a western blot to verify this up-regulation. During nutrient deprivation, levels of YTHDF3 greatly increased, while simultaneously, there was no significant difference in levels of other YTH family proteins (Fig. 1b). This finding is different from recent research reported that hypoxia upregulated YTHDF1 expression[31].

To analyze whether YTHDF3 induction is related to autophagy occurrence, we knocked down YTHDF3 in MEFs using two distinct shRNA-expressing lentiviruses (Supplementary Fig. 1a). Analyzing LC3-II levels in the presence or absence of the lysosomal inhibitor bafilomycin A1 (Baf.A1) revealed that autophagy flux is impaired when YTHDF3 expression is silenced. As another indicator of autophagy, p62 is an autophagosome cargo protein that degrades in autolysosomes. While nutrient starvation-induced p62 degradation in control cells, YTHDF3 ablation resulted in p62 accumulation (Fig.1c), also suggesting decreased autophagy flux. Furthermore, we investigated whether YTHDF3 depletion affected the mRNA levels, mRNA stability as well as protein translation efficiency of p62. Our data showed that silencing YTHDF3 didn't alter mRNA levels or mRNA stability of p62 under starvation (Supplementary Fig. 2a, b). Meanwhile, compared to the control group, the translation efficiency of p62 in YTHDF3 knockdown (KD) cells did not change significantly according to the polysome-profiling assay (Supplementary Fig. 2c). These results exclude the possibility that the attenuated decrease of p62 protein levels under starvation in YTHDF3 silencing cells is due to altered mRNA stability or protein synthesis. Further, we performed rescue experiments by ectopically expressing a full-length YTHDF3 in KD MEFs (Supplementary Fig. 1b). Western blot analysis showed that the level of endogenous LC3-II and p62 degradation were both significantly increased in nutrient-deficient MEFs with restored YTHDF3, as compared to the KD counterpart (Fig.1d), demonstrating that YTHDF3 rescued shRNA-induced inactivation of autophagy. YTHDF3 depletion consistently and significantly diminished cytosolic GFP-LC3 puncta (Fig.1e, f), and this defect could be rescued by re-expressing YTHDF3 (Fig.1g, h). Moreover, we found that YTHDF3 overexpression markedly potentiated LC3-II expression and decreased p62 levels (Fig.1i and Supplementary Fig. 1c), indicating that YTHDF3 is not only necessary for maintaining physiological autophagy but also mediates autophagy enhancement. This finding was also confirmed by a GFP-LC3 assay, which showed an increased number of cytosolic GFP-LC3 puncta through ectopic expression of YTHDF3 (Fig.1j, k). With a CRISPR/Cas9 system, we generated YTHDF3$^{-/-}$ MEFs from E13.5 YTHDF3$^{-/-}$ mouse embryos (Supplementary Fig. 1d). Transmission electron microscopy (TEM) data confirmed that YTHDF3 loss causes a robust decrease in the number of autophagosomes and autolysosomes (Fig.1l). Collectively, these data indicate that YTHDF3 is a positive regulator, one that is required for autophagy induction during nutrient deficiency. In addition, we also detected the effects of YTHDF1 and YTHDF2 depletion on autophagy flux, respectively. Silencing of YTHDF1 dampened the LC3-II elevation and p62 degradation during nutrient starvation, but this effect was observed to be much less significant than silencing YTHDF3 (Supplementary Fig. 3a, b). On the other hand, silencing YTHDF2 had the opposite effect on autophagy flux as compared to inhibition of YTHDF1 and YTHDF3 (Supplementary Fig. 3a, c).

### YTHDF3 depletion impairs autophagosome formation and lysosomal function

The process of autophagy can be summarized as five continuous events: initiation, nucleation, expansion, fusion, and cargo degradation. Accordingly, we further investigated which step of autophagy is interrupted in YTHDF3 KD cells. During autophagic vesicle maturation, autophagosomes fuse with lysosomes to form autolysosomes containing active proteases. By using a tandem fluorescent-tagged mCherry-GFP-LC3 reporter, we measured the abundance of autophagosomes and autolysosomes. Because the mCherry fluorophore is more tolerant to a drop in pH than GFP is, non-acidic phagophores and autophagosomes were labeled with both GFP and mCherry; after fusing with lysosomes, autolysosomes become acidic and GFP fluorescence is quenched. Under conditions of nutrient deprivation, YTHDF3 loss caused a dramatic decrease in the number of both non-acidic (mCherry$^+$GFP$^+$) and acidic (mCherry$^+$GFP$^-$) punctate, demonstrating

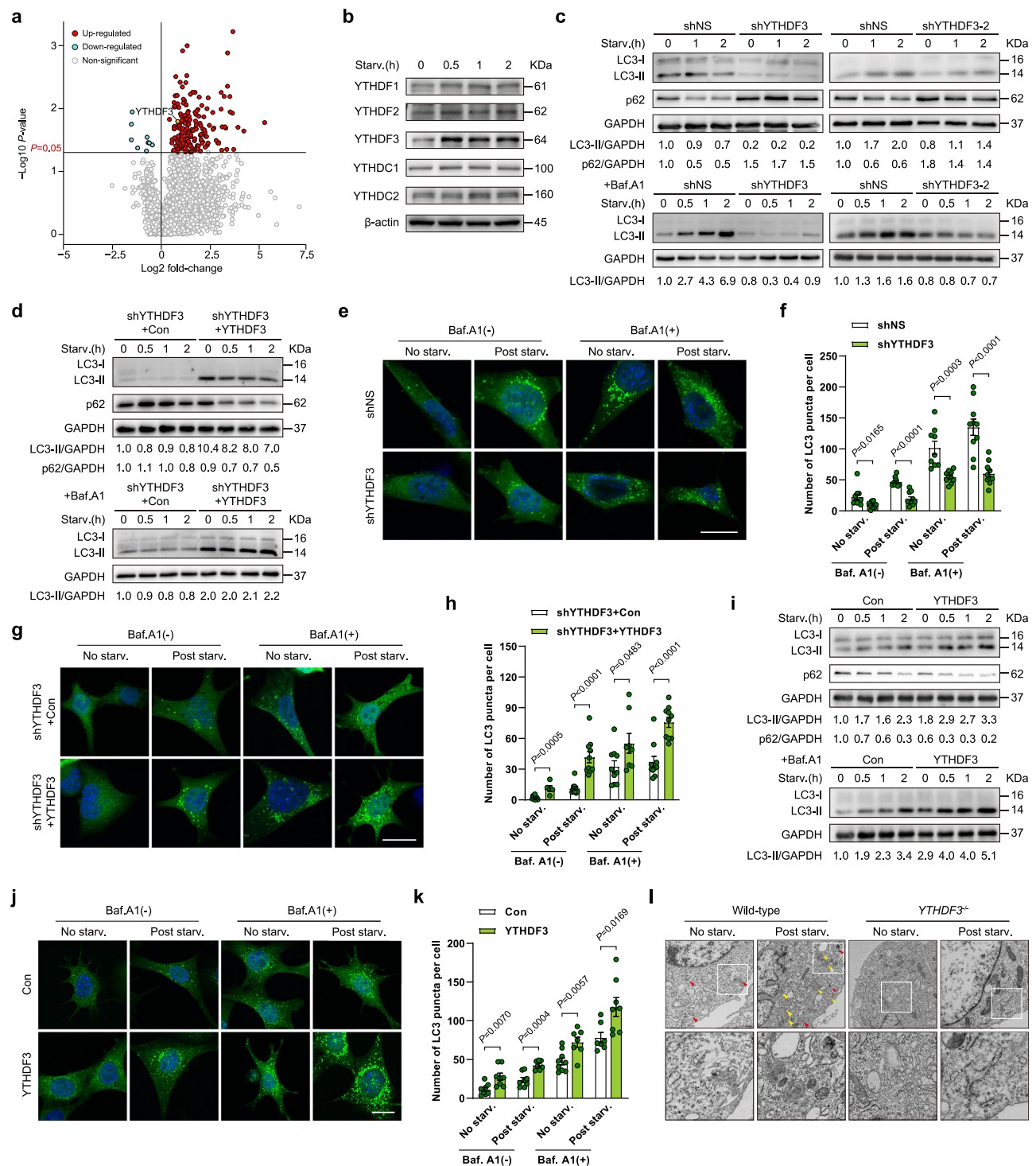

that the formation of autophagosomes and autophagy flux is severely compromised (Fig. 2a, b).

In mammals, autophagy initiation is induced by the ULK1 complex. When activated, the ULK1 complex phosphorylates the components of class III PtdIns3K complexes[32], forming specialized PtdIns3P-enriched subdomains on the endoplasmic reticulum (ER) membrane[33]. To probe whether YTHDF3 contributes to the early steps of autophagosome formation, we examined the localization of representative factors from the two complexes under nutrient deprivation. Knocking down YTHDF3 suppressed starvation-induced puncta formation of ULK1 and ATG13 (Fig. 2c, d), indicating that ULK1 complex translocation to the phagophore initiation site was compromised. ATG13

reportedly bridges the ULK1 and PtdIns3K complexes. This interaction enables ULK1 to phosphorylate ATG14, stimulating the kinase activity of the PtdIns3K complex, and thus promoting autophagosome nucleation[32]. In line with ATG13, knocking down YTHDF3 significantly decreased dot formation of ATG14, as well as PtdIns3P-binding protein DFCP1, after nutrient starvation (Fig. 2c, d). These results demonstrate that YTHDF3 depletion impairs the early steps of autophagosome formation, including initiation and nucleation.

Since reduced amounts of autophagosomes can correspondingly reduce the autolysosomes produced by autophagosome and lysosome fusion, we analyzed whether YTHDF3 regulates autophagy flux by simultaneously affecting lysosomal activity. Acridine orange (AO) is a

**Fig. 1 | YTHDF3 up-regulation is required for autophagy induction. a** Volcano plot depicting the intracellular proteins, detected by LC-MS/MS, that are differentially expressed in MEFs under nutrient-deprived versus normal conditions. The fold change values were calculated by dividing LFQ intensities measured in starved MEFs by intensities measured in normal MEFs. The significance threshold is set at a $P$-value below 0.05, from two-tailed unpaired $t$-tests performed on three biological replicates. Significantly up- and down-regulated proteins are denoted by red and blue dots, respectively. **b** Immunoblot analyses of YTH domain-containing proteins in MEFs in response to nutrient deficiency analyzed after the indicated time periods. β-actin is shown as a loading control. **c** Immunoblot analyses of LC3-II and p62, in shNS and two independent shYTHDF3 MEFs, following nutrient starvation for the indicated time periods, with and without Baf.A1 treatment (20 nM). GAPDH is used as a loading control. **d** Immunoblot analyses of YTHDF3 restored (shYTHDF3 + YTHDF3) and control (shYTHDF3 + Con) MEFs following nutrient starvation for the indicated time periods, with and without Baf.A1 treatment (20 nM). GAPDH is used as a loading control. **e**, **f** Representative confocal images of GFP-LC3 puncta formation (**e**) and quantification of GFP-LC3 puncta per cell (**f**) in shNS and shYTHDF3 MEFs. Nuclei were counterstained with DAPI. Scale bar, 20 μm. **g**, **h** Representative

confocal images of GFP-LC3 puncta formation (**g**) and quantification of GFP-LC3 puncta per cell (**h**) in YTHDF3 restored and control MEFs. Nuclei were counterstained with DAPI. Scale bar, 20 μm. **i** Immunoblot analyses of LC3-II and p62 in control and YTHDF3 overexpressing MEFs, following nutrient starvation for the indicated time periods, with and without Baf.A1 treatment (20 nM). GAPDH is used as a loading control. **j**, **k** Representative confocal images of GFP-LC3 puncta formation (**j**) and quantification of GFP-LC3 puncta per cell (**k**) in control and YTHDF3 overexpressing MEFs. Nuclei were counterstained with DAPI. Scale bar, 20 μm. **l** Representative TEM images of autophagosomes (yellow arrow) and autolysosomes (red arrows) in wild-type and *YTHDF3*[−/−] MEFs, with and without nutrient starvation. High magnification images of the boxed areas are displayed on the right-hand side. Scale bar, 1 μm. For **f**, **h**, **k**, mean numbers of puncta per cell from each randomly selected fields over three independent experiments were plotted (dots). Bars represent mean ± SEM. Two-tailed unpaired multiple $t$-tests with two-stage step-up correction (Benjamini, Krieger, and Yekutieli) were used to estimate significance. $P$-values are indicated in the figure. Source data are provided as a Source Data file.

cell-permeant dye that emits green fluorescence; however, when trapped in acidic compartments, AO emits red fluorescence when excited by blue light[34]. Because Baf.A1 is known to inhibit lysosomal acidification, we used cells treated with Baf.A1 as a positive control. YTHDF3 KD cells exhibited decreased red-to-green fluorescence intensity ratio (R/GFIR) (Fig. 2e), suggesting that silencing YTHDF3 can increase lysosomal pH. This result was confirmed by LysoTracker Red staining (Fig. 2f, g). Next, we investigated whether silencing YTHDF3 affects the hydrolytic function of lysosomes. We loaded both control and YTHDF3 KD cells with DQ-Red BSA, which releases fluorescent monomers when degraded by lysosomal proteases. Attenuated red fluorescence in YTHDF3 KD cells indicated a decrease in intracellular proteolysis (Fig. 2h, i). Because differential endocytic rates may affect DQ-Red BSA trafficking to lysosomes, we used cell-permeant Magic Red to measure the activity of cathepsin B, a lysosomal marker enzyme. Results showed that YTHDF3 KD cells had lower Magic Red fluorescence (Fig. 2j, k), indicating impaired cathepsin activity in lysosomes. Collectively, these results indicate that YTHDF3 depletion impairs autophagosome formation and causes lysosomal dysfunction.

## mTORC1 and AMPK signaling are not affected by YTHDF3

Autophagosome initiation is regulated by mTORC1 and AMPK signaling according to nutrient cues[17,35]. The fact that silencing YTHDF3 impairs the formation of the earliest autophagic structures prompted us to consider whether YTHDF3 affects the activities of upstream mTORC1 and AMPK signaling. We analyzed mTORC1 activity by measuring the phosphorylation states of mTOR at S2448 and mTORC1 downstream targets, including p70S6K at T389 and 4E-BP1 at T37/46. In both control and YTHDF3 KD cells, phosphorylation of mTOR S2448, p70S6K T389, and 4E-BP1 T37/46 were markedly reduced upon nutrient deficiency (Fig. 2l). We then assessed whether YTHDF3 regulates AMPK activity by measuring the phosphorylation levels of AMPKα at T172 and AMPK substrate RAPTOR at S792. Phosphorylation of AMPKα T172 and RAPTOR S792 was markedly increased upon nutrient starvation, while patterns of change in YTHDF3 KD cells were the same as those in control cells (Fig. 2m). These data illustrate that mTORC1 and AMPK signaling are not affected by YTHDF3.

## YTHDF3 requires METTL3-mediated m⁶A modification to promote autophagy

Since YTHDF3 is an m⁶A reader, we asked whether YTHDF3 requires m⁶A to promote autophagy. To assess m⁶A changes during autophagy, we performed an immunofluorescence assay using an antibody that recognizes m⁶A-modified nucleic acids. Intriguingly, we found that m⁶A signals accumulated in the cytoplasm during nutrient deficiency (Fig. 3a, b). Meanwhile, YTHDF3 also accumulated in the cytoplasm

during this process (Fig. 3a, b). m⁶A is harbored in both mRNA and non-coding RNAs[36] and can be recognized by an m⁶A antibody; we focused on mRNAs to investigate whether they are hypermethylated upon nutrient deficiency. Poly(A) + RNA purified with oligo(dT) displayed a significant increase in m⁶A modification upon nutrient deficiency (Fig. 3c), suggesting that m⁶A levels of mRNA are elevated during autophagy induction.

We then wished to identify the m⁶A encoder responsible for m⁶A hypermethylation in mRNAs during this process. In mammalian cells, the m⁶A modification is installed by a writer methyltransferase complex, composed of METTL3 and METTL14, and removed by the erasers ALKBH5 and FTO[37–39]. Of these four factors, only METTL3 was significantly induced after nutrient deficiency (Fig. 3d and Supplementary Fig. 4a, b). Knocking down METTL3 impaired the m⁶A signals accumulating in the cytoplasm (Supplementary Fig. 5a, b) and reduced mRNA hypermethylation during autophagy induction (Fig.3e and Supplementary Fig. 5a), suggesting that METTL3 is necessary for m⁶A induction upon nutrient deficiency. We further validated the dynamic increment of METTL3 in response to nutrient deficiency by glucose starvation, albeit variations were observed in METTL3 induction among different cell types (Supplementary Fig. 4c). Interestingly, METTL3 induction was not observed during short-term starvation in the hepatocellular carcinoma cell line HepG2, but its expression increased after a longer starvation induction (Supplementary Fig. 4d). These data suggest that METTL3 induction might be a general protection mechanism against diverse types of nutrient scarcity.

To explore METTL3's catalyzing activity was truly activated under starvation, we directly measured the m⁶A catalytic activity of METTL3 within cell extracts from both control and starved cells. We employed the S-(5′-Adenosyl)-L-methionine-d₃ (d₃-SAM) to quantify METTL3 methyltransferase activity with RNA probes containing the consensus motif 'GGACU', as previously described[40,41]. Our data showed that the cellular METTL3 obtained from starved cells achieved a higher molar ratio of d₃-m⁶A to RNA probe than that of control cells (Fig. 3f), suggesting nutrient starvation enhances N6-adenosine methylation efficiency. One possible mechanism to explain this phenotype is the different effects of decreased S-adenosine methionine (SAM) and S-adenosylhomocysteine (SAH) levels on METTL3 methyltransferase activity. Briefly, SAM, generated from ATP and methionine, provides methyl groups and constitutively activates METTL3's catalytic activity in the m⁶A methylation process. In contrast, SAH, generated from the methylation reactions, in turn, inhibits METTL3's methyltransferase activity[42]. Since cellular SAM levels are much higher than their $K_m$ (substrate concentration at half-maximum reaction rate) for METTL3[43], the reduction of ATP or SAM might have no significant effect on METTL3's activity at the early stage of nutrient starvation. Meanwhile,

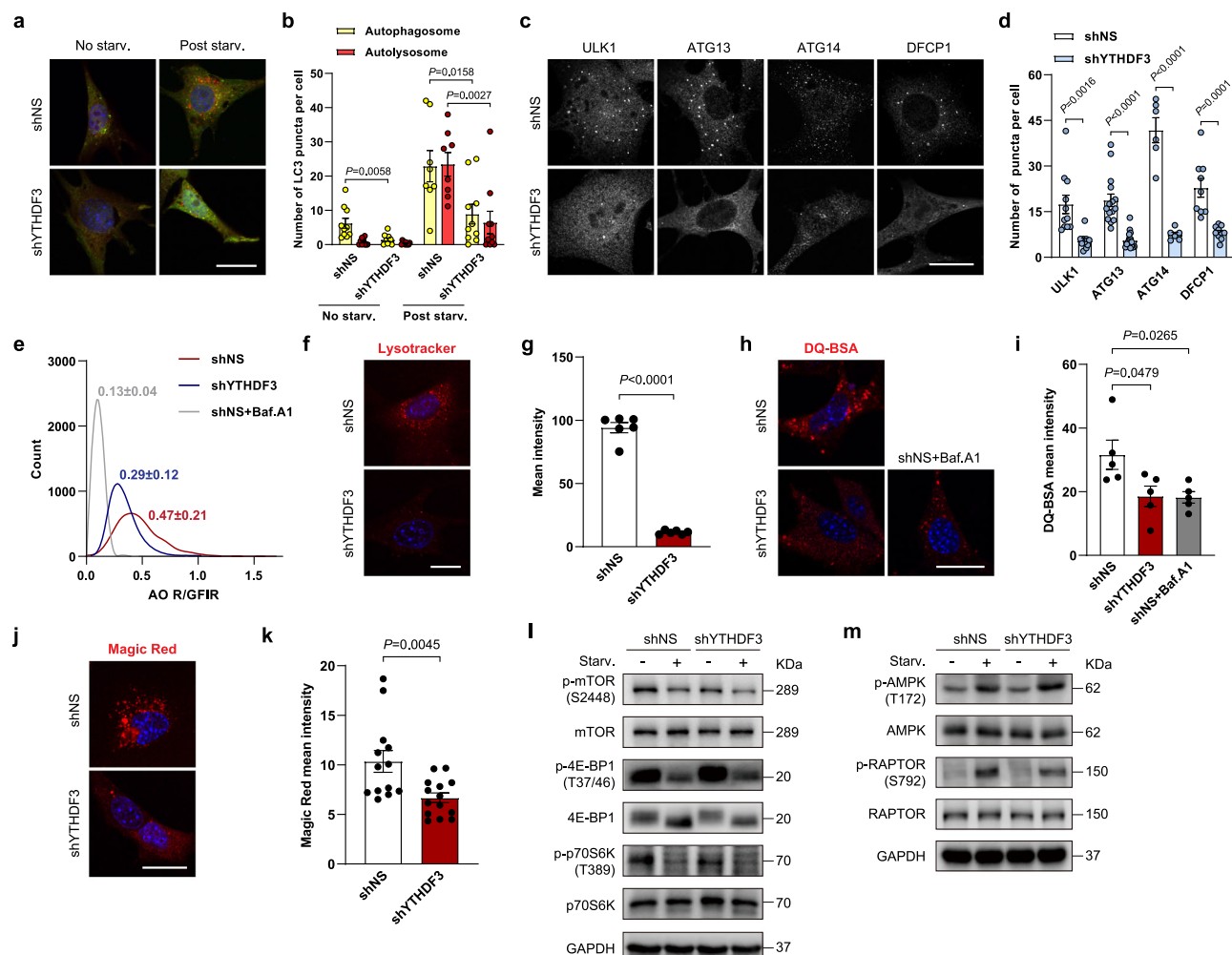

**Fig. 2 | YTHDF3 depletion impairs autophagosome formation and lysosomal function. a**, **b** mCherry-GFP-LC3 was transfected into shNS and shYTHDF3 MEFs, and autophagosome (yellow) and autolysosome (red) formation was examined. Scale bar, 20 μm. **c**, **d** Endogenous ULK1, ATG13, ATG14, and DFCP1 puncta in shNS and shYTHDF3 MEFs were immunostained after nutrient deficiency, visualized with confocal microscopy (**c**), and quantified (**d**). Scale bar, 20 μm. **e** Histograms of R/GFIR of 3000 events analyzed by flow cytometry after AO staining in shYTHDF3 MEFs and shNS MEFs, with or without 20 nM Baf.A1 for 4 h. Values above the histogram indicate mean R/GFIR ± SEM of three experiments. **f**, **g** Representative images (**f**) and quantification (**g**) of total LysoTracker Red in shNS and shYTHDF3 MEFs. Scale bar, 20 μm. **h**, **i** Representative images (**h**) and quantification (**i**) of intracellular proteolysis by DQ-BSA in shYTHDF3 MEFs and shNS MEFs, with or without 20 nM Baf.A1 for 4 h. Scale bar, 20 μm. **j**, **k** Live imaging of Magic Red dye,

which detects active Cathepsin B, in shNS and shYTHDF3 MEFs (**j**). Quantification of Magic Red intensity using ImageJ Software (**k**). Scale bar, 20 μm. **l**, **m** Immunoblot analyses of phosphorylation of mTOR (S2448), 4E-BP(T37/46) and p70S6K(T389) (**l**), and phosphorylation of AMPK(T172) and RAPTOR(S792) (**m**), in shNS and shYTHDF3 MEFs, with or without nutrient starvation. GAPDH is included as a loading control. For **b**, **d**, **g**, **i**, **k**, mean numbers of puncta per cell (**b**, **d**), or mean fluorescence intensity of treated cells (**g**, **i**, **k**), from each randomly selected fields over three independent experiments were plotted (dots). Bars represent mean ± SEM. Two-tailed unpaired multiple *t*-tests with two-stage step-up correction (Benjamini, Krieger, and Yekutieli) (**b**, **d**), or two-tailed unpaired *t*-tests (**g**, **i**, **k**), were used to estimate significance. *P*-values are indicated in the figure. Source data are provided as a Source Data file.

the level of cellular SAH is relatively close to its $IC_{50}$ (half-maximal inhibitory concentration) on METTL3[43], thus the reduction of SAH under nutrient starvation would alter METTL3's activity much earlier than the reduction of SAM. As starvation time prolongs, a significant decrease in SAM levels towards its $K_m$ might result in a significant decrease in its activating effect on METTL3's activity. In line with this, we observed the molar ration of $d_3$-m6A to RNA probe decreased 6 h post starvation (Fig. 3f), indicating that the catalytic activity of METTL3 was diminished. Interestingly, METTL3 protein levels did not change significantly during prolonged starvation (Fig. 3g). We speculated this might be a compensatory mechanism for cells under nutrient exhaustion. As noticed, the autophagy activity of MEFs was only transiently activated upon nutrient starvation. The LC3-II levels with Baf.A1 increased at 1-2 h starvation but declined after 6 h (Supplementary Fig. 6a). Consistent with this, the viability of MEFs did not

change so much within 2 h starvation, however, longer starvation caused cellular viability loss with morphological changes (Supplementary Fig. 6b, c). Furthermore, short-term starvation increased cellular ATP levels, but with a longer starvation period, the ATP levels decreased apparently (Supplementary Fig. 6d). These results suggested that MEFs within short-term starvation might be under a compensatory status that rapid autophagy could favor its living and metabolism; however, prolonged starvation might lead a decompensated status where reduced autophagy was beneficial to delay metabolic exhaustion and cell death in MEFs. Therefore, we supposed that METTL3-mediated m6A modification might be critical for YTHDF3 to promote autophagy. Using two distinct shMETTL3 lentiviruses to infect MEFs overexpressing YTHDF3 (Supplementary Fig. 7a–c), we found that nutrient starvation-induced LC3-II accumulation and p62 degradation were remarkably attenuated (Fig. 3h). To establish

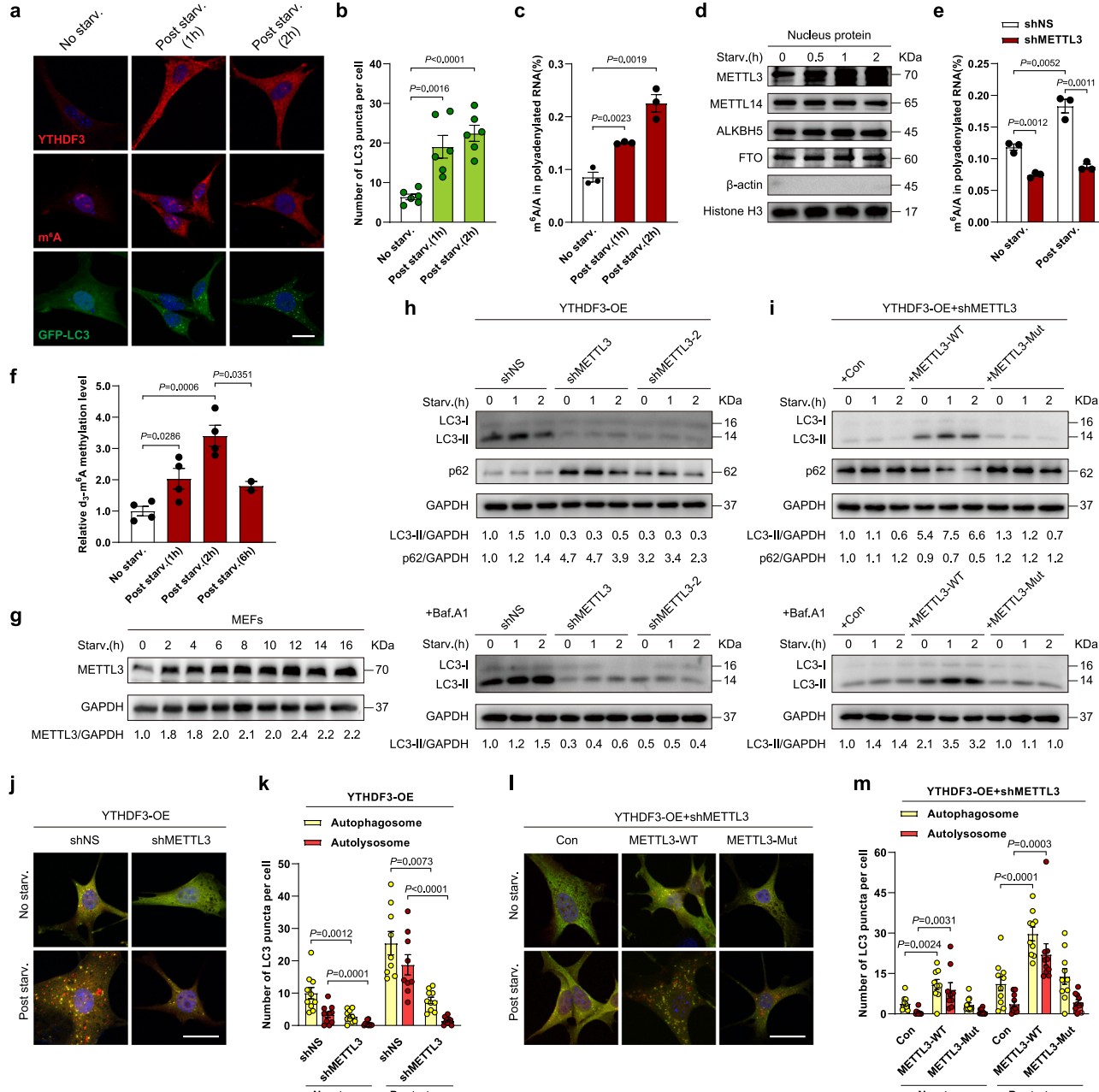

**Fig. 3 | YTHDF3 requires METTL3-mediated m⁶A modification to promote autophagy. a** Representative confocal images of YTHDF3 and m⁶A fluorescence localization were obtained in MEFs following nutrient starvation for the indicated time periods. Nuclei were stained with DAPI. Scale bar, 20 μm. **b** Quantification of GFP-LC3 puncta per cell. **c** LC-MS/MS quantification of m⁶A levels in mRNA extracts from MEFs following nutrient starvation for the indicated time periods ($n = 3$ biological replicates). **d** Immunoblot analyses of nuclear fractions from MEFs following nutrient starvation for the indicated time periods. **e** LC-MS/MS quantification of m⁶A levels in mRNA extracts from shNS and shMETTL3 MEFs, with or without nutrient deprivation ($n = 3$ biological replicates). **f** The relative m⁶A methylation catalytic activities of purified METTL3 from the MEFs starved for the indicated time periods were determined using an RNA probe and $d_3$-m⁶A. The methylation yields were calculated based on the molar ratio of newly formed $d_3$-m⁶A to digested RNA probes ($n = 4, 4, 4, 2$ biological replicates). G was used as an internal control to calculate the amount of RNA probes. **g** Immunoblot analysis of METTL3 in MEFs following nutrient starvation for the indicated time periods. **h** Immunoblot analyses of LC3-II and p62 in YTHDF3-OE MEFs infected with shNS or two independent METTL3 shRNAs (shMETTL3 and shMETTL3-2) following nutrient starvation for the indicated time periods, with or without Baf.A1 treatment (20 nM). GAPDH was used

as a loading control. **i** Immunoblot analyses of METTL3-silenced YTHDF3-OE MEFs transfected with lentiviral vectors (Con), wild-type METTL3 (METTL3-WT), or a catalytic mutant of METTL3 (METTL3-Mut) following nutrient starvation for the indicated time periods, with or without Baf.A1 treatment (20 nM). GAPDH is used as a loading control. **j, k** Measurement of autophagy flux and quantification of autophagosomes (yellow) and autolysosomes (red) by a tandem mCherry-GFP-LC3 reporter assay in shNS and shMETTL3 MEFs of YTHDF3-OE, with or without nutrient deficiency. Scale bar, 20 μm. **l, m** Measurement of autophagy flux and quantification of autophagosomes (yellow) and autolysosomes (red) by a tandem mCherry-GFP-LC3 reporter assay in METTL3-silenced YTHDF3-OE MEFs transfected with METTL3-WT, METTL3-Mut, or Con, with or without nutrient deficiency. Scale bar, 20 μm. For **b, k, m** mean numbers of puncta per cell from each randomly selected fields over three independent experiments were plotted (dots). All bars represent mean ± SEM. Two-tailed unpaired *t*-tests (**b, c, e, f**), or two-tailed unpaired multiple *t*-tests with two-stage step-up correction (Benjamini, Krieger, and Yekutieli) (**k, m**), were used to estimate the significance. *P*-values are indicated in the figure. Source data are provided as a Source Data file.

whether the effect of METTL3 on YTHDF3-promoted autophagy relies on its m⁶A catalytic activity, we re-introduced either the wild-type or catalytic-mutant METTL3 back into the METTL3-silenced cells (Supplementary Fig. 7d, e) and showed that the wild-type METTL3, but not the catalytic-mutant form, could rescue autophagic defects in METTL3-silenced cells (Fig. 3i). In line with these results, a tandem mCherry-GFP-LC3 reporter assay demonstrated that METTL3 depletion compromised the generation of autophagosomes and autolysosomes upon nutrient deficiency in YTHDF3-overexpressing MEFs (Fig. 3j, k), and that wild-type METTL3 could recover this functional inactivation where the catalytic-mutant could not (Fig. 3l, m). Collectively, these data suggest that YTHDF3 requires METTL3-mediated m⁶A modification to promote autophagy.

Given that METTL3-mediated m⁶A may realize its role in autophagy through different readers, we explored how METTL3 modulated autophagy without manipulating YTHDF3 expression. As expected, METTL3 depletion markedly dampened LC3-II accumulation and p62 degradation during nutrient deficiency (Supplementary Fig. 8a), suggesting autophagy flux is impaired. Interestingly, METTL3 overexpression had little effect on LC3-II and p62 levels (Supplementary Fig. 8b), indicating that the METTL3 is redundant in response to nutrient starvation to induce autophagy. To examine whether METTL3 regulates autophagy in an m⁶A-dependent manner, we conducted rescue experiments in METTL3-depleted cells by re-expression of either wild-type or catalytic-mutant METTL3. Similar to those YTHDF3-overexpressing cells, we observed that re-introducing wild-type METTL3, but not the catalytic-mutant, restored nutrient starvation-induced LC3-II accumulation and p62 degradation (Supplementary Fig. 8c), indicating that METTL3's regulated autophagy flux is m⁶A-dependent. These observations were further confirmed by a tandem mCherry-GFP-LC3 reporter assay. Nutrient starvation-induced autophagosomes and autolysosomes were significantly impeded in METTL3-deficient cells (Supplementary Fig. 8d, e), and this autophagic defect was reversed by re-introducing wild-type METTL3, but not the catalytic-mutant form (Supplementary Fig. 8f, g). In addition, no significant effect of METTL3 overexpression was observed on the phenotypes of starvation-induced autophagosome and autolysosome formations (Supplementary Fig. 8h, i). Taken together, these results suggest that METTL3-mediated m⁶A modification is essential for nutrient starvation-induced autophagy. Furthermore, decreased LC3-II levels and p62 accumulation was observed in METTL14-silencing cells compared to control cells (Supplementary Fig. 9a, b). However, since the physical co-operation of METTL14 is necessary for METTL3's methylases catalytic function[37], we cannot exclude the possibility that such impact on autophagy is a direct METTL14-dependent effect or via METTL3 methyltransferase activity.

## Inhibited RPS27a-METTL3 interaction stabilizes METTL3 under starvation

Further, to examine whether METTL3 was transcriptionally regulated, we first analyzed changes in METTL3 mRNA abundance during nutrient deprivation. Our results showed only a slight increase in response to nutrient starvation (Fig. 4a). We next examined whether nutrient starvation could affect METTL3 mRNA stability, but no significant difference was observed (Fig. 4b). Interestingly, METTL3 protein upregulation upon nutrient starvation was almost completely eliminated by treatment of MG132, a classical proteasome inhibitor (Fig. 4c), indicating the protein stability regulation of METTL3 is the main reason for its significant induction under nutrient deficiency. This notion was corroborated by a chase assay using a protein synthesis inhibitor, cycloheximide (CHX), which revealed that nutrient starvation attenuated METTL3 protein degradation (Fig. 4d). In addition, we also noticed that nutrient starvation markedly reduced the ubiquitination of METTL3 (Fig. 4e). To identify the potential regulators which interact with METTL3 and account for its de-ubiquitination in response to

starvation, we overexpressed FLAG-tagged METTL3 in MEFs and subjected the anti-FLAG immunoprecipitates to mass spectrometry. Surprisingly, we found that 6 ubiquitination-related proteins were co-purified with METTL3, including RPS27a, RPL23, RPL11, RPS2, RPS3, and HSPA5 (Fig. 4f). Among these proteins, RPS2, RPS3, and HSPA5 were only reported to be ubiquitinated[44,45], but no evidence indicated they had ubiquitylation regulatory roles. Therefore, we focused on RPS27a, RPL23, and RPL11 for further investigation. Interestingly, our data revealed nutrient starvation attenuated the RPS27a-METTL3 interactions, and increased the interactions of RPL11 and RPL23 with METTL3 (Fig. 4f, g). Moreover, RPS27a, RPL23, and RPL11 expressions were not significantly affected by starvation (Fig. 4h). These results indicated that the interactions of these proteins to METTL3 might affect METTL3 protein ubiquitination and stabilization. To further confirm this notion, we examined the METTL3 expressions in MEFs transfected with siRNAs targeting the RPS27a, RPL23, and RPL11, respectively. The results showed METTL3 was upregulated in siRPS27a MEFs but not significantly changed in RPL11 or RPL23 knocked down MEFs (Fig. 4i). Since RPS27a is a ubiquitin fusion protein, it can release active ubiquitin monomers, which mediate the protein ubiquitously degradation. We then examined whether RPS27a leads to METTL3 ubiquitination. Our result showed the inhibition of RPS27a strongly attenuated METTL3 ubiquitination (Fig. 4j). These results suggest that the impaired RPS27a-METTL3 interaction upon nutrient starvation results in METTL3 ubiquitination suppression, thus increasing METTL3 stabilization.

## YTHDF3 recognizes starvation-induced m⁶A hypermethylation of FOXO3 mRNA

Next, we sought to identify the mRNAs that differentially bind to YTHDF3 upon nutrient deficiency. In two replicates of YTHDF3 RIP-seq, the 4881 and 5013 peaks were more enriched under nutrient deprivation conditions, whereas the 2997 and 2865 peaks were more enriched under normal conditions. By overlapping transcripts from the two replicates, 3424 up-enriched and 1814 down-enriched transcripts were obtained upon nutrient starvation. Among them, the 1041 up-enriched and 535 down-enriched transcripts were thought to be significant (Fig. 5a). Analyzing these binding sites, the m⁶A core motif 'GGAC' was highly detected (Fig. 5b). Most of these binding sites are located in protein-coding transcripts and highly enriched in CDS and 3′ UTR regions, especially near the stop codon (Fig. 5c), coinciding with the distributive pattern of m⁶A peaks[36]. Therefore, we supposed that the mRNAs that are altered binding to YTHDF3 upon nutrient deficiency are mainly m⁶A modified.

Subsequently, we profiled the dynamic changes of m⁶A in mRNAs during nutrient deficiency. Consistent with published studies[36], m⁶A peaks were significantly enriched with the 'GGAC' motif and predominantly located in the CDS and 3′UTR regions, especially near the stop codon, in both control and starved cells (Supplementary Fig. 10a). Relative to normal conditions, a total of 2811 more m⁶A peaks and 2552 fewer m⁶A peaks were identified upon nutrient deficiency (Fig. 5d). The global m⁶A enrichment analysis confirmed a significant up-regulation of m⁶A modification levels in mRNAs after starvation (Supplementary Fig. 10b). To explore why only a sub-class of mRNAs showed an increased m⁶A modification, we first compared the identified hyper-m⁶A-methylated genes with METTL3-bound transcripts from starBase human CLIP-seq data[46]. Irrespective of species and cell type difference, 36% of the hyper-m⁶A-methylated genes were also found in the METTL3-bound transcripts (Supplementary Fig. 10c). Since parts of the m⁶A modifications deposited by METTL3 were also reversed by m⁶A demethylases (FTO or ALKBH5), we next compared the METTL3-bound transcripts without hyper-methylation to the FTO- or ALKBH5-bound transcripts using the starBase human CLIP-seq data[46], and found that 65% of genes without hyper-m⁶A-methylation following starvation were simultaneously bound by FTO or ALKBH5

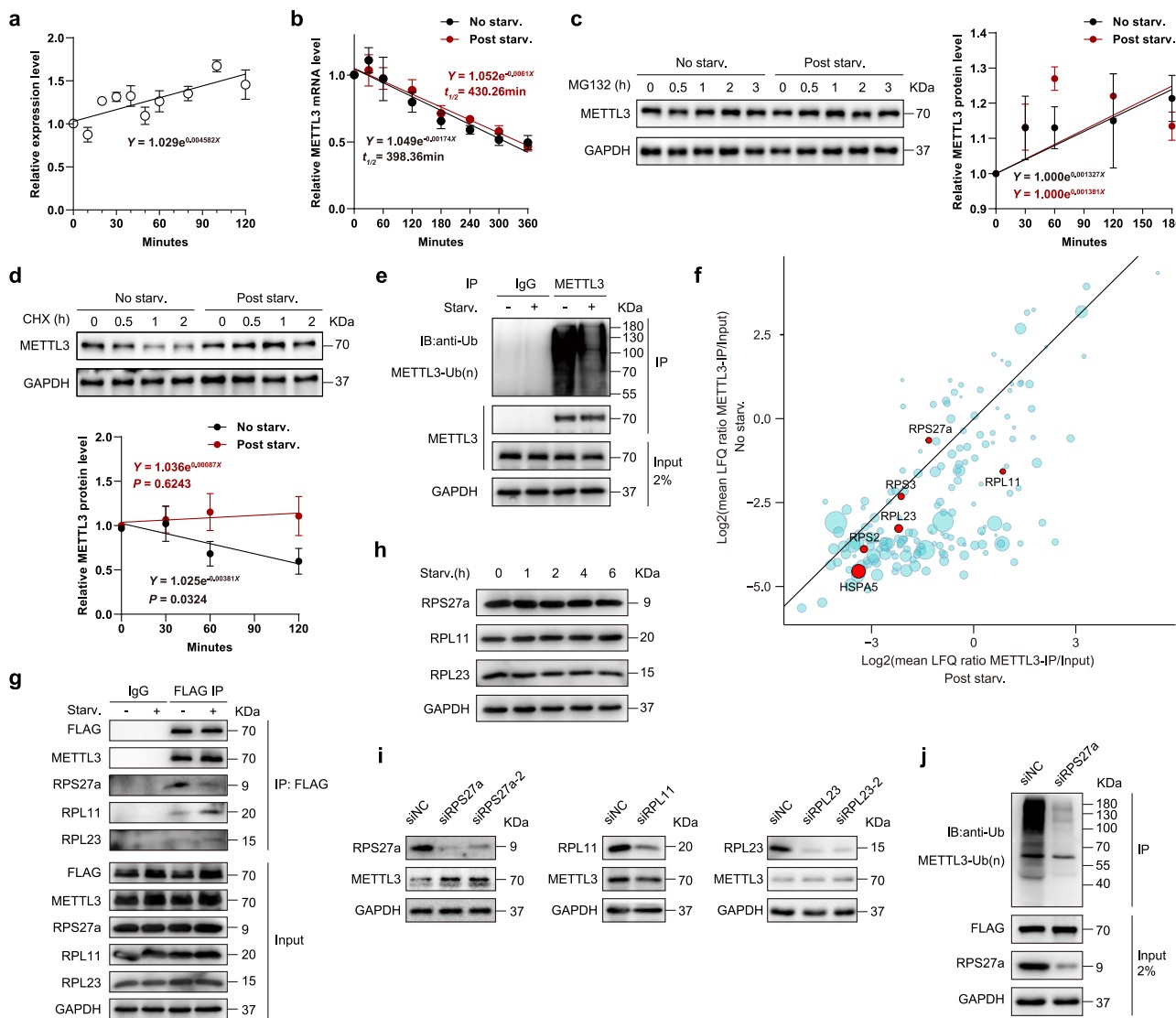

**Fig. 4 | Inhibited RPS27a-METTL3 interaction stabilizes METTL3 under starvation. a** qRT-PCR analysis of METTL3 in MEFs following nutrient starvation for the indicated time periods ($n = 4$ biological replicates). **b** MEFs were treated with Act.D (5 μg/mL) for the indicated times with or without nutrient starvation, respectively. The expression of METTL3 was examined with qRT-PCR ($n = 3$ biological replicates). **c** Left, MEFs with and without nutrient deficiency were treated with 20 μM MG132 for the indicated time periods. Levels of METTL3 were examined by immunoblot analyses. GAPDH was used as a loading control. Right, relative METTL3 protein levels were quantitatively defined ($n = 3$ biological replicates). **d** Up, MEFs with and without nutrient deficiency were treated with 100 μg/ml cycloheximide (CHX) for the indicated time periods. Levels of METTL3 were examined by immunoblot analyses. GAPDH was used as a loading control. Down, protein half-life of METTL3 was quantitatively defined ($n = 3$ biological replicates). Simple linear regression

assessing METTL3 protein decay rates showed significant decay under normal conditions ($P = 0.0324$) versus starved conditions ($P = 0.6243$). **e** METTL3-overexpressing MEFs were treated with or without nutrient starvation after incubation with 20 μM MG132. Ubiquitinated METTL3 were pulled down with an anti-METTL3 antibody, and then subjected to western blotting using anti-ubiquitin antibodies. **f** Identification of METTL3-interacting proteins by quantitative mass spectrometry. The ubiquitination-related proteins are labeled in red. The rest of the proteins are shown in bright blue. **g** Interactions between METTL3 and the indicated proteins were analyzed. **h** Immunoblot analyses of RPS27a, RPL11, and RPL23 in MEFs following nutrient starvation for the indicated time periods. **i** MEFs knocked down of indicated proteins were subjected to immunoblotting. **j** In vivo ubiquitination assay of METTL3 in RPS27a KD and control MEFs. Data are presented as mean values ± SEM. Source data are provided as a Source Data file.

(Supplementary Fig. 10c). Since the m⁶A modification levels are determined by the balance between methylation and demethylation, we, therefore, propose that only a sub-class of mRNAs were highly m⁶A methylated during the starvation process because of up-regulation of demethylation occurred with the other mRNAs. In addition, those transcripts without m⁶A modification sites should not change their m⁶A methylation during starvation.

To identify the potential m⁶A-hypermethylated targets that increase binding to YTHDF3 upon nutrient deficiency, we intersected the up-enriched peaks of YTHDF3 RIP-seq upon nutrient deficiency with the hyper-m⁶A peaks, resulting in 86 peaks (Fig. 5e). Among all 86 peaks, there were 7 genes annotated to GO-term Autophagy

(0006914), including FOXO3, BMF, DDIT3, AP4M1, SESN2, ZFYVE1, and ZFYVE26 (Fig. 5e). In these genes, FOXO3 is one of the most important transcriptional factors which regulates autophagy according to the literatures[47]. By performing RIP-qPCR, we verified that most of these autophagy-related transcripts except ZFYVE26 were more enriched by YTHDF3 upon nutrient deficiency and their interactions were dependent on METTL3. Importantly, the FOXO3 enrichment was most prominent (Fig. 5i). Furthermore, by immunoblotting, we found silencing YTHDF3 could reduce the expressions of FOXO3 and ZFYVE1, while YTHDF3 overexpression had an opposite effect (Fig. 6a and Supplementary Fig. 11b). In contrast, BMF, DDIT3, and SESN2 did not show obviously corresponding changes as FOXO3 in both YTHDF3-deficient

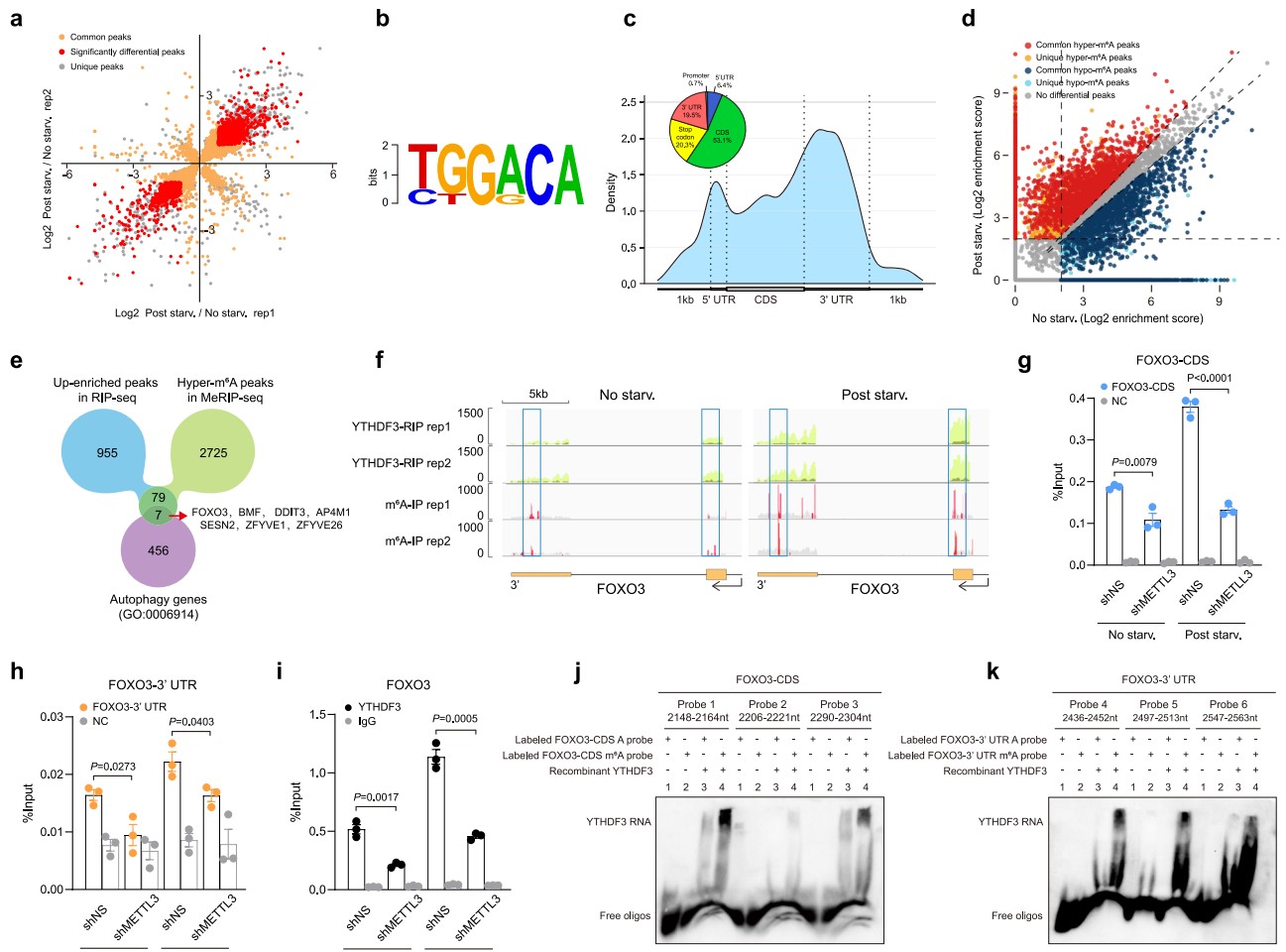

**Fig. 5 | YTHDF3 recognizes starvation-induced m⁶A hypermethylation of FOXO3 mRNA. a** The scatter plot depicts fold changes (log2) of YTHDF3-RIP target peaks in MEFs after nutrient deprivation. Red dots indicate significantly up-enriched and down-enriched peaks with a cutoff fold change of 1.8. **b** The consensus sequence motif identified within significant differentially enriched YTHDF3-binding sites, determined by the HOMER database. **c** Metagene profiles of the significant differentially enriched YTHDF3-binding sites along a normalized transcript, consisting of three rescaled non-overlapping segments: 5'UTR, CDS, and 3'UTR. Pie chart depicting the fraction of significant differentially enriched YTHDF3-binding sites in different transcript segments. **d** Scatter plot showing m⁶A peaks with increased (red) or decreased (blue) levels in response to nutrient deficiency. **e** Venn diagram showing the number of overlapping up-enriched YTHDF3-binding targets and hyper-m⁶A-methylated mRNAs upon nutrient deprivation. Then the resultant 86 peaks were annotated to GO term autophagy (0006914) and

obtained 7 genes. **f** Integrative Genomics Viewer (IGV) tracks displaying YTHDF3-RIP-seq (upper panels) and MeRIP-seq (lower panels) read distribution along the CDS and 3'UTR of FOXO3 mRNA. The squares mark increases in m⁶A peaks in MEFs upon nutrient deficiency. **g, h** Gene-specific MeRIP-qPCR analysis of m⁶A level changes at the CDS (**g**) and 3'UTR (**h**) regions of FOXO3 mRNA transcripts in shNS and shMETTL3 MEFs upon nutrient deficiency ($n = 3$ biological replicates). **i** YTHDF3-RIP followed by qRT-PCR confirmed the interaction between YTHDF3 and FOXO3 mRNA in shNS and shMETTL3 MEFs upon nutrient deficiency ($n = 3$ biological replicates). **j, k** RNA EMSA assays were performed using recombinant YTHDF3 proteins and biotinylated RNA probes containing different sequences of FOXO3's CDS (**j**) or 3'UTR (**k**) with or without m⁶A modifications. Data are presented as mean values ± SEM. Two-tailed unpaired *t*-tests were used to estimate significance. *P*-values are indicated in the figure. Source data are provided as a Source Data file.

and YTHDF3-overexpressed MEFs (Supplementary Fig. 11b). Of note, due to the lack of a proper antibody targeting AP4M1 and the role of AP4M1 in autophagy was not well defined, we did not test AP4M1 expression. In addition, we examined four more autophagy-related genes in the list which was reported previously, including CDKN1B, PLD6, CCR4, and TXNIP. Our data showed silencing or overexpression of YTHDF3 had no significant effect on these genes' expressions (Supplementary Fig. 11c). Thus, FOXO3 and ZFYVE1 were the most promising candidates for mediating YTHDF3-promoted autophagy among the 86 peaks. However, as shown in a previous study that knocking down ZFYVE1 does not suppress autophagy[48], we, therefore, selected and focused on FOXO3 for further investigation. Our data showed that upon nutrient deficiency, the m⁶A hypermethylated peaks in FOXO3 mRNA are located in CDS and 3'UTR regions around the stop codon (Fig. 5f). To verify that the YTHDF3-FOXO3 mRNA interaction relies on the METTL3-mediated m⁶A modification, we knocked down

METTL3 in MEFs. Silencing METTL3 reduced the m⁶A peaks in CDS and 3'UTR regions around the stop codon of FOXO3 transcripts due to nutrient deficiency, as well as m⁶A hypermethylation, as validated by MeRIP-qPCR analysis (Fig. 5g, h). Synchronized with attenuated m⁶A in FOXO3 mRNA, YTHDF3 binding to FOXO3 mRNA decreased accordingly (Fig. 5i). To directly examine whether YTHDF3 binds to the m⁶A locus of FOXO3 transcripts, we performed electrophoretic mobility shift assays (EMSAs) using recombinant YTHDF3 proteins and biotinylated RNA probes containing different sequences of FOXO3's CDS or 3'UTR with or without m⁶A modifications. Our results showed that once m⁶A modifications were removed from the RNA probes, no matter the m⁶A location in either the CDS or 3'UTR of FOXO3, the interactions between YTHDF3 and the probes were significantly attenuated (Fig. 5j, k). Furthermore, since the observed band intensity of probe 1 or 3 combined with YTHDF3 were significantly higher than those of other probes with YTHDF3, we reasoned the m⁶A sites at

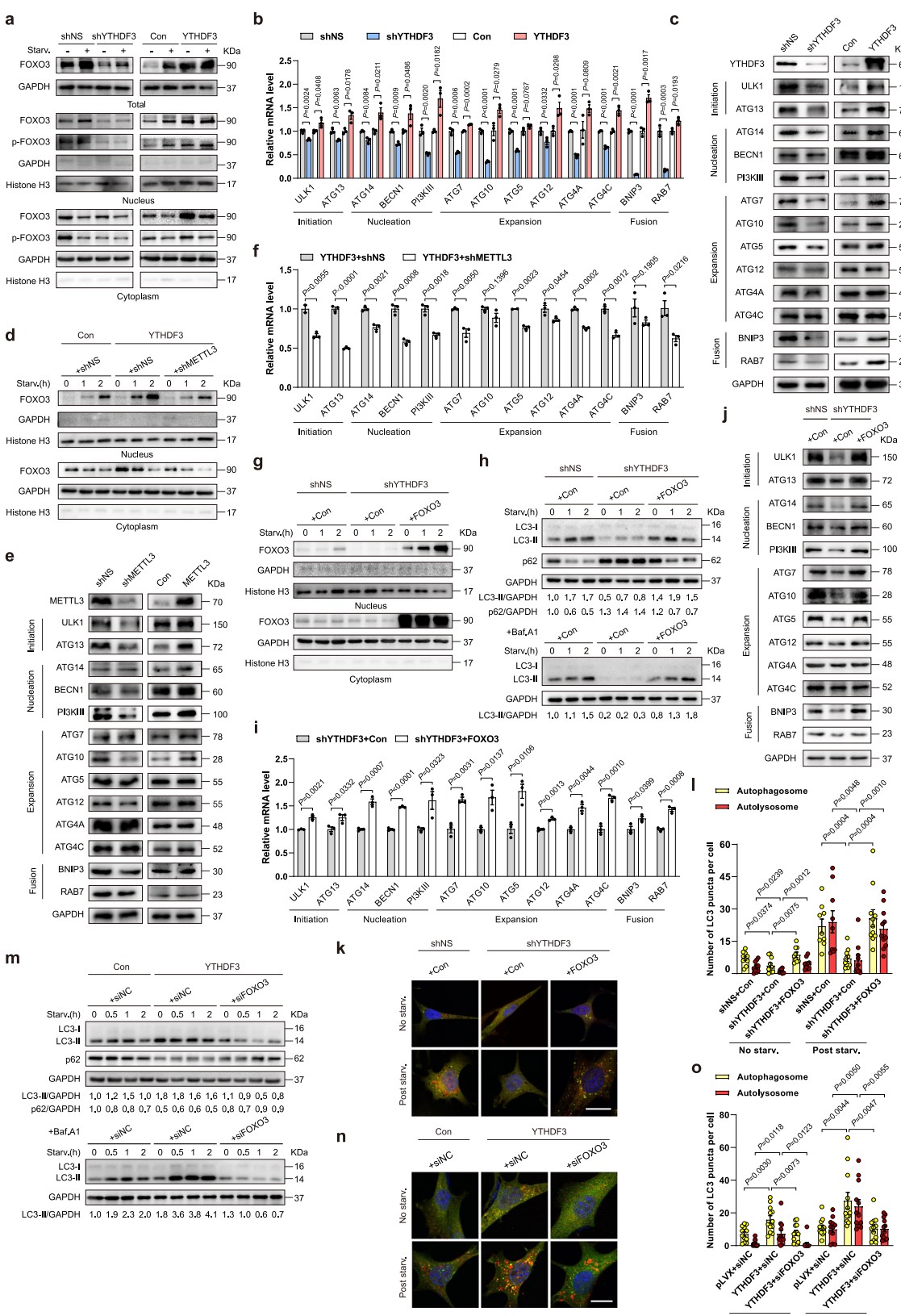

2158nt, 2151nt, 2163nt (probe 1), and 2295nt (probe 3) along the FOXO3-CDS might be more critical for YTHDF3 to recognize FOXO3 stop codon area than other m⁶A sites (Fig. 5j, k). Collectively, these results suggest that METTL3-mediated m⁶A hypermethylation during nutrient deficiency is required for YTHDF3-FOXO3 mRNA interaction.

## FOXO3 is a crucial target for YTHDF3 to promote autophagy

To investigate the role of YTHDF3 in regulating FOXO3 expression, we knocked down YTHDF3 and found that FOXO3 protein levels are remarkably attenuated under both normal and nutrient-starved conditions (Fig. 6a). Correspondingly, YTHDF3 overexpression led to an increase in FOXO3 expression (Fig. 6a). Since FOXO3 is a known

**Fig. 6 | FOXO3 is a crucial target for YTHDF3 to promote autophagy.**
**a** Immunoblot analyses of FOXO3 and p-FOXO3 in shNS, shYTHDF3, control, and
YTHDF3-OE MEFs, with or without nutrient starvation, respectively. **b** qRT-PCR
analysis of mRNA levels of FOXO3 target autophagy-related genes in shNS,
shYTHDF3, control, and YTHDF3-OE MEFs ($n = 3$ biological replicates).
**c** Immunoblot analyses of protein levels of FOXO3-targeted autophagic genes in
shNS, shYTHDF3, control, and YTHDF3-OE MEFs. **d** Immunoblot analyses of nuclear
and cytoplasmic FOXO3 expressions in METTL3-silenced YTHDF3-OE MEFs and
control MEFs following nutrient starvation for the indicated time periods.
**e** Immunoblot analyses of protein levels of FOXO3-targeted autophagic genes in
shNS, shMETTL3, control, and METTL3-OE MEFs. **f** qRT-PCR analysis of mRNA levels
of FOXO3 target autophagy-related genes in METTL3-silenced YTHDF3-OE MEFs
and control MEFs ($n = 3$ biological replicates). **g** Immunoblot analyses of nuclear
and cytoplasmic FOXO3 in FOXO3 rescued MEFs (shYTHDF3 + FOXO3) and control
MEFs (shYTHDF3 + Con). GAPDH is used as a loading control. **h** Immunoblot ana-
lyses of FOXO3 rescued MEFs (shYTHDF3 + FOXO3) and control MEFs
(shYTHDF3 + Con) following nutrient starvation for the indicated time periods,

with or without Baf.A1 treatment (20 nM). **i** qRT-PCR analysis of mRNA levels of
FOXO3 target autophagy-related genes in FOXO3 rescued and control MEFs ($n = 3$
biological replicates). **j** Immunoblot analyses of protein levels of FOXO3 targeted
autophagic genes in FOXO3 rescued MEFs and control MEFs. GAPDH is used as a
loading control. **k, l** mCherry-GFP-LC3 was transfected into FOXO3 rescued and
control MEFs, and the formation of autophagosomes (yellow) and autolysosomes
(red) was examined. Scale bar, 20 μm. **m** Immunoblot analyses of FOXO3-silenced
YTHDF3-OE MEFs (YTHDF3 + siFOXO3) and control MEFs (YTHDF3 + siNC) follow-
ing nutrient starvation for the indicated time periods, with or without Baf.A1
treatment (20 nM). **n, o** mCherry-GFP-LC3 was transfected into FOXO3-silenced
YTHDF3-OE MEFs and control MEFs, and the formation of autophagosomes (yel-
low) and autolysosomes (red) was examined. Scale bar, 20 μm. For **i, o**, mean
numbers of puncta per cell from each randomly selected fields over three inde-
pendent experiments were plotted (dots). All bars represent mean ± SEM. Two-
tailed unpaired multiple t-tests with two-stage step-up correction (Benjamini,
Krieger, and Yekutieli) were used to estimate significance. P-values are indicated in
the figure. Source data are provided as a Source Data file.

autophagy transcriptional regulator[49], we then examined the effect of
YTHDF3 on the expression of key FOXO3 target genes[49,50]. Silencing
YTHDF3 resulted in decreased mRNA levels of FOXO3 targets involved
in autophagy initiation, nucleation, expansion, and autophagosome-
lysosomal fusion, while overexpressing YTHDF3 resulted in upregula-
tion of these genes (Fig. 6b). Changed protein levels of these autop-
hagic genes were further confirmed with western blotting assay
(Fig. 6c). Further, we examined phosphorylation, the most pre-
dominant posttranslational mechanism in regulating FOXO3 activity. It
is reported the phosphorylation of FOXO3 at the S413 residue leads to
transcriptional activity promotion[51]. Therefore, we tested whether
YTHDF3 affected FOXO3 phosphorylation at this site. Our data showed
p-FOXO3(S413) levels were increased in the nucleus and decreased in
the cytoplasm upon nutrient starvation, in a similar manner to the total
fractions (Fig. 6a). On the other hand, we detected a decrease in p-
FOXO3(S413) levels in YTHDF3 KD cells, whereas YTHDF3 over-
expression led to an opposite effect. However, the ratio of phos-
phorylated FOXO3 to the pan-FOXO3 was not obviously affected
(Fig. 6a). These results suggested YTHDF3 might have a key role in
regulating FOXO3 translation rather than FOXO3 posttranslational
modification such as phosphorylation. By silencing or overexpression
of YTHDF3, most of the FOXO3 target genes involved in autophagy
including ULK1, ATG13, ATG14, PI3KIII, ATG7, ATG10, ATG5, BNIP3, and
RAB7 were significantly changed at both the mRNA and protein levels.
However, four genes (BECN1, ATG12, ATG4A, and ATG4C) displayed
very mild changes in protein levels, whereas significant changes were
detected in their transcriptional levels (Fig. 6b, c). We assumed this
might be due to the difference in gene-specific and celltype-dependent
gene expression regulations. Consistently, we found MEFs either
silencing or overexpressing FOXO3 had significant effects on the
transcription of the above-mentioned genes but had no clear effects
on their protein levels (Supplementary Fig. 12a, b). In contrast, other
FOXO3 targets such as ATG14, PI3KIII, ATG7, and BNIP3 were sig-
nificantly changed in protein levels when manipulating FOXO3
genetically, similar to the effect of YTHDF3 (Supplementary Fig. 12b).

We next explored the expression of FOXO3 after METTL3 deple-
tion. Knocking down METTL3 obviously abrogated FOXO3 expression
levels in the nucleus and cytoplasm of YTHDF3-overexpressing MEFs
under both normal and starved conditions (Fig. 6d). Notably, nuclear
FOXO3 in METTL3-silenced cells was much significantly lower than
that in control cells because of its translocation to the nucleus in
response to starvation (Fig. 6d). In addition, most of the protein levels
of FOXO3 target genes involved in autophagy were also attenuated by
METTL3 depletion and enhanced by METTL3 overexpression (Fig. 6e).
However, the discrepancies in expressions of FOXO3 targets were also
detected in the MEFs silencing or overexpressing METTL3 (Fig. 6e).
Nevertheless, since YTHDF3-regulated autophagy was METTL3-

dependent, we observed most of the FOXO3 targets promoted by
YTHDF3 overexpression were reduced when silencing METTL3, even
for those unchanged genes in METTL3-silenced cells (Supplementary
Fig. 12c). Above all, these data further suggest that YTHDF3 regulates
FOXO3 expression and alters the expression of FOXO3-targeted
autophagic genes in a METTL3-dependent manner.

To determine whether impaired FOXO3 expression accounts for
autophagy dysfunction in YTHDF3-deficient cells, we performed res-
cue experiments (Fig. 6g and Supplementary Fig. 13). Ectopic expres-
sion of FOXO3 in YTHDF3-deficient cells rescued reduced levels of
LC3-II in both the presence and absence of Baf.A1, rescued the reduced
degradation of p62 (Fig. 6h), and restored the expression level of
FOXO3 target genes (Fig. 6i, j). Using a mCherry-GFP-LC3 reporter, we
demonstrated that the ectopic expression of FOXO3 greatly recovered
autophagosome formation and defects of autophagy flux caused by
YTHDF3 deficiency (Fig. 6k, l). To further determine the role of FOXO3
in YTHDF3-regulated autophagy, we knocked down FOXO3 and
examined if YTHDF3-promoted autophagy flux was attenuated. Our
data showed that the upregulations of LC3-II levels and p62 degrada-
tion promoted by YTHDF3 were remarkably impeded when silencing
FOXO3 (Fig. 6m). Consistently, the mCherry-GFP-LC3 reporter assay
showed that knocking down FOXO3 suppressed the starvation-
induced autophagosome and autolysosome formations promoted by
YTHDF3 overexpression (Fig. 6n, o). Taken together, these results
indicate that FOXO3 is a key functional target for YTHDF3 to promote
autophagy.

It is known that under physiological conditions, autophagy has a
homeostatic function in the disposal of excessive protein aggregates,
lipids, and dysfunctional organelles in vivo. Attenuated autophagic
response to starvation may lead to metabolic inflexibility. Interest-
ingly, we observed although $YTHDF3^{-/-}$ mice were comparable in body
weight to wild-type mice under normal diet conditions, they exhibited
significantly less body weight loss than their age-matched wild-type
controls after a 24 h fasting (Supplementary Fig. 14a–c). Meanwhile, we
noticed that the weight of the fasted livers seemed heavier in $YTHDF3^{-/-}$
mice than in wild-type mice, although the P-value was not statistically
significant (Supplementary Fig. 14d, e). Indeed, $YTHDF3^{-/-}$ mice showed
more depositions of glycogen and lipid droplets within the hepato-
cytes than wild-type mice upon fasting (Supplementary Fig. 14f–h).
These results indicate that YTHDF3 may play important role in facil-
itating nutrient utilization probably by promoting autophagy. Next, we
examined the expressions of LC3B-II and FOXO3 in livers derived from
wild-type and $YTHDF3^{-/-}$ mice. Compared to wild-type mice, fasted
$YTHDF3^{-/-}$ mice livers showed marked attenuated expressions in LC3B-
II and FOXO3 levels (Supplementary Fig. 14i). Above all, these data
indicate that YTHDF3-regulated autophagy might play important roles
in nutrient homeostasis in vivo.

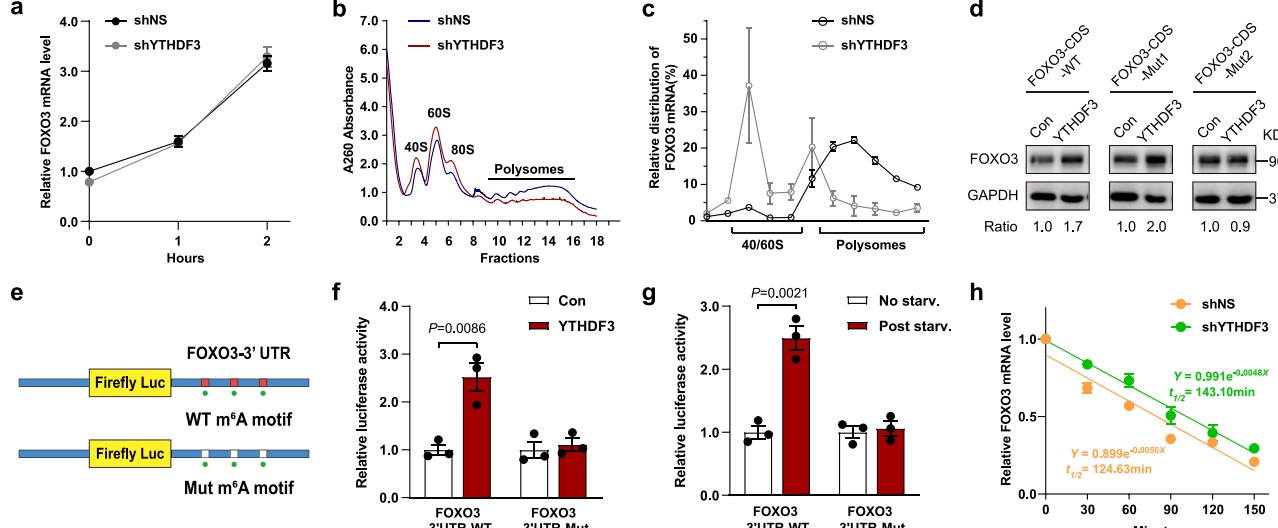

**Fig. 7 | YTHDF3 promotes FOXO3 translation but does not affect its mRNA stability. a** qRT-PCR analysis of FOXO3 in shNS and shYTHDF3 MEFs following nutrient starvation for the indicated time periods. **b** Sucrose gradient-based poly-some profiling of shNS and shYTHDF3 MEFs. **c** FOXO3 mRNAs in each ribosome fraction were quantified through qRT-PCR and plotted as percentages of the total. **d** Lv-FOXO3-CDS-WT, Lv-FOXO3-CDS-Mut1, or Lv-FOXO3-CDS-Mut2 were trans-fected into control and YTHDF3-OE MEFs. Protein expression was measured by western blot analysis and then quantitatively analyzed. **e** Schematic diagram of wild-type or mutant m⁶A sites (A-to-T mutation) in the FOXO3-3′UTR, fused with a dual luciferase reporter. **f** FOXO3-3′UTR-WT or FOXO3-3′UTR-Mut reporters were transfected into control and YTHDF3-OE MEFs for 72 h. Firefly luciferase activity was measured and normalized to Renilla luciferase activity. **g** After transfecting FOXO3-3′UTR-WT or FOXO3-3′ UTR-Mut, MEFs were nutrient starved. Firefly luci-ferase activity was measured and normalized to Renilla luciferase activity. **h** shNS and shYTHDF3 MEFs were treated with Act.D (5 μg/mL) for the indicated times. The expression of FOXO3 was examined with qRT-PCR. Data from three independent experiments are expressed as mean values ± SEM. *P*-values are indicated in the figure. Source data are provided as a Source Data file.

## YTHDF3 promotes FOXO3 translation but does not affect its mRNA stability

YTHDF3 has been reported to facilitate RNA translation by interacting with 40S and 60S ribosomal subunits[52,53] and cooperating with YTHDF2 to reduce mRNA stability[52]. Therefore, we wondered about the mechanism by which YTHDF3 regulates FOXO3 expression. Intri-guingly, knocking down YTHDF3 did not affect the level of FOXO3 mRNA (Fig. 7a), indicating that YTHDF3 may have an effect on FOXO3 translation. Using polysome profiling, we showed that knocking down YTHDF3 resulted in a marked increase in 40S/60S ribosome and 80S monosome fractions, as well as a decrease in polysome fractions (Fig. 7b), corresponding with the results of previous studies[52,53]. The level of FOXO3 mRNA in the translating pool (polysomes) of YTHDF3-deficient cells was lower than that of control cells (Fig. 7c), suggesting that FOXO3 translation was attenuated by YTHDF3 depletion.

Methylated RNA immunoprecipitation sequencing (MeRIP-seq) data revealed two significant peaks in FOXO3 mRNA—the first and second peaks were located in the CDS and 3′UTR region around the stop codon, respectively (Fig. 5f). Using the mammalian m⁶A site pre-dictor SRAMP (sequence-based RNA adenosine methylation site pre-dictor) algorithm as an auxiliary tool[54], we found that the density of m⁶A sites predicted with high confidence was much higher in these locations than in other regions (Supplementary Fig. 15). Therefore, we asked whether YTHDF3 regulates FOXO3 translation by recognizing these m⁶A sites.

To address the effect m⁶A sites in FOXO3-CDS have on YTHDF3-facilitated FOXO3 translation, we generated wild-type and mutant FOXO3-CDS expression constructs. For the mutant form, in m⁶A motifs (RRACH) of the identified peak near the stop codon, we replaced adenosine bases with thymine, thus eliminating the m⁶A modification (FOXO3-CDS-mut1). For comparison, m⁶A motifs of the upstream CDS were also mutated (FOXO3-CDS-mut2). Our data showed that FOXO3-CDS-mut1, rather than FOXO3-CDS-mut2, eliminated YTHDF3-enhanced FOXO3 expression (Fig. 7d), demonstrating that the m⁶A sites near the stop codon in FOXO3-CDS play a critical role in YTHDF3-

promoted FOXO3 translation. To examine the direct regulatory role of the m⁶A sites near the stop codon in the FOXO3-3′UTR region on YTHDF3-facilitated FOXO3 translation, luciferase reporter assays were performed. The identified 3′UTR portions of FOXO3, containing the wild-type and mutant m⁶A sites, were cloned into the 3′ UTR regions of the reporter gene firefly luciferase in pEZX-MT06 vectors (Fig. 7e). With constructs containing the wild-type FOXO3-3′UTR region, YTHDF3 overexpression significantly induced the expression of firefly luciferase, while constructs containing mutated m⁶A sites diminished the effect (Fig. 7f). In accordance with this result, MEFs transfected with the wild-type FOXO3-3′UTR construct showed significantly increased firefly luciferase activity upon nutrient deficiency; this increase was abrogated when the m⁶A sites were mutated (Fig. 7g). For comparison, we examined the m⁶A motifs of the downstream 3′UTR, which were observed also hyper-methylated upon nutrient starvation in MeRIP-seq data. With constructs containing either the wild-type or mutant m⁶A sites, we did not observe a significant difference in firefly luciferase activity in YTHDF3 overexpressing and control MEFs (Sup-plementary Fig. 16a). Moreover, nutrient starvation did not increase the firefly luciferase activity (Supplementary Fig. 16b). Collectively, these results indicate that the m⁶A sites near the stop codon in FOXO3-3′UTR regions are also involved in regulating YTHDF3-facilitated FOXO3 translation.

Furthermore, we investigated whether YTHDF3 regulates FOXO3 mRNA stability. We treated control and YTHDF3-deficient cells with transcriptional inhibitor actinomycin D (Act D). However, no sig-nificant difference was found between the two groups (Fig. 7h), indi-cating that YTHDF3 does not affect the stability of FOXO3 mRNA. Collectively, our data suggest that YTHDF3 promotes FOXO3 transla-tion, depending on its recognition of m⁶A sites around the stop codon in FOXO3 CDS and 3′UTR regions, but YTHDF3 does not affect FOXO3 mRNA stability.

Additionally, we also determined the effects of other YTHDF proteins on FOXO3 expression by western blotting. YTHDF1 KD cells showed decreased expression of FOXO3 proteins compared to their

control counterparts, while YTHDF2 KD slightly increased FOXO3 expression, especially in the nucleus following nutrient starvation (Supplementary Fig. 17a, b). To further investigate their effects on FOXO3 expression, we performed polysome profiling to evaluate the translational efficiency of FOXO3 mRNAs when knocked down YTHDF1 and YTHDF2, respectively. Our data indicated a down-regulation of FOXO3 mRNA level in polysome fractions when knocking down YTHDF1 (Supplementary Fig. 17c, d), while FOXO3 mRNA level was unchanged in YTHDF2 KD cells (Supplementary Fig. 17e, f). RNA stability assays revealed that neither YTHDF1 nor YTHDF2 depletion could prolong the half-life of FOXO3 mRNAs (Supplementary Fig. 17g, h). This finding is inconsistent with two previous findings which show that FOXO3 mRNA stability was enhanced by YTHDF1 overexpression in liver cancer cells under hypoxia[55] or reduced by YTHDF2 overexpression in the luteinized GCs of PCOS patients[56]. Interestingly, although both YTHDF1 and YTHDF2 bind to FOXO3 transcript, nutrient deprivation has the opposite effect on their binding interaction. Our data showed that starvation could significantly strengthen the binding of YTHDF1, but decreased the binding of YTHDF2, to FOXO3 mRNAs (Supplementary Fig. 17i, j). These results suggest that YTHDF1 and YTHDF3 might act synergistically to compete with YTHDF2 to bind FOXO3 mRNAs under nutrient deficiency. Such findings can also explain the observation that YTHDF2 had a mild effect on FOXO3 expression under normal conditions, but upon nutrient starvation, YTHDF2 depletion could clearly upregulate FOXO3 expression (Supplementary Fig. 17b).

### YTHDF3 may interact with eIF3a and eIF4B to promote FOXO3 translation

Translational control occurs most frequently during the initiation stage. In this process, eIF4B enhances the RNA helicase activity of eIF4A. eIF4F, consisting of eIF4A, eIF4E, and eIF4G, unwinds the 5′UTR region of mRNA and binds to the m[7]G cap. eIF3a is the largest subunit of the eIF3 complex. These subunits bind stably with 40S ribosomes to form 43S preinitiation complexes (43S PIC). The 43S PIC complexes are then recruited to the 5′UTR region of mRNA by eIF4G-eIF3 interactions, thereby stimulating the initiation of protein synthesis. From co-immunoprecipitation (co-IP) LC-MS/MS data, we noticed that YTHDF3 is co-purified with multiple translational initiation factors. Among these proteins, eIF3a and eIF4B were the most significantly up-enriched proteins upon nutrient deficiency (Fig. 8a, b). Western blot analysis of different ribosomal fractions revealed that eIF3a and eIF4B shift from heavier to lighter polysome fractions upon YTHDF3 deficiency (Fig. 8c), indicating that eIF3a and eIF4B may play a role in the impact of YTHDF3 on translation. In silico docking analysis[57] suggested potential protein-protein interactions between eIF3a, eIF4B, and YTHDF3. In the docking models with the lowest docked energy (YTHDF3-eIF3a: -1002.8; YTHDF3-eIF4B: -834.7), the following residues were positioned at the modeled interface and were responsible for the interactions: Y424, E426, and T441 in YTHDF3 and R476, R483, and I484 in eIF3a; Y424, T441, and S470 in YTHDF3 and S88, F99, and Y141 in eIF4B (Fig. 8d). To validate the interactions between YTHDF3 and eIF3a or eIF4B, we performed co-IP assays. Endogenous eIF3a and eIF4B were co-precipitated with YTHDF3 in both normal and nutrient-free conditions; nutrient starvation could simultaneously increase amounts of precipitated eIF3a, eIF4B, and YTHDF3. In YTHDF3 KD MEFs, few proteins were precipitated (Fig. 8e). Next, we asked whether these interactions were bridged by RNAs. Adding RNase A to lysates did not reduce the amount of eIF3a and eIF4B isolated with YTHDF3, suggesting that this interaction is RNA independent (Fig. 8f). To further determine the impact of eIF3a and eIF4B on FOXO3 translation, we used shRNA-expressing lentiviruses to knockdown either eIF3a or eIF4B (Fig. 8g). Knocking down either of these two proteins lowered FOXO3 protein levels (Fig. 8g), indicating that FOXO3 translation was attenuated. Collectively, these results suggest that YTHDF3 may

interact with eIF3a and eIF4B to promote FOXO3 translation. In antiviral innate immunity, YTHDF3 interacts with PABP1 and eIF4G2, and promotes FOXO3 translation by binding its mRNA translation initiation region under homeostasis[58]. Our co-IP assays also confirmed the interaction between YTHDF3 and PABP1 or eIF4G2 in MEFs. However, these interactions were significantly reduced following nutrient starvation (Fig. 8h), whereas the interaction between YTHDF3 and eIF3a or eIF4B was increased (Fig. 8e), suggesting the interactions between YTHDF3 and translation initiation regulators might be quite different in diverse cell subtypes and under various cellular stress conditions.

To further explore potential RBPs involved in YTHDF3 binding specificity, we analyzed variations in YTHDF3 protein interactome upon nutrient starvation from our co-IP LC-MS/MS data. We observed that altered-binding proteins were highly enriched in protein catabolic processes, stress response regulation, autophagy, lysosome, and translational initiation using GO and KEGG pathway analyses (Supplementary Fig. 18a, b). Next, we screened the putative YTHDF3 coregulators with known target sequences based on RBP databases (including RBPDB, starBase, and POSTAR3), and obtained 97 up-enriched and 24 down-enriched proteins for further RBP binding sites analyses (Supplementary Fig. 18c). Most of these RBPs contained conserved RNA-binding domains, such as the RNA-recognition motifs (RRMs), the K-Homology (KH) domains, and Zinc Fingers (ZFs) (Supplementary Fig. 18c), suggesting they might play a role in the YTHDF3-RNA recognition process. To further examine how YTHDF3's transcript-binding selectivity was achieved during nutrient deprivation, we focused on those target sequences which were recognized by well-defined RBPs in the list of YTHDF3 interactors. We found that elements with 'GGAC' motifs were significantly enriched in the target sequences of multiple potential YTHDF3 RBP partners, including eIF3a, eIF3d, eIF4B, IGF2BP2, NOP58, FMR1, SRSF7, CNBP, HNRNPUL1, RBM39, TAF15, TARDBP, and MBNL1 (Supplementary Fig. 18d), indicating these RBPs might cooperate with YTHDF3 to selectively recognize its target sites. Moreover, according to published CLIP-Seq data, 46 of the YTHDF3 RBP interactors were found to bind FOXO3 transcripts, suggesting these proteins might cooperate with YTHDF3 to specifically recognize FOXO3 mRNAs (Supplementary Figs. 19–21). We then systematically analyzed the FOXO3 transcript binding sites of each potential YTHDF3 RBP partner and overlapped them with the starvation-induced YTHDF3 binding and hyper-methylated sites. This strategy revealed a bunch of proteins, including ATXN2, CSTF2T, SRSF7, DDX54, CNBP, FBL, FMR1, FXR2, FUS, FAM120A, HNRNPA1, HNRNPC, IGF2BP1, IGF2BP2, NOP58, NOP56, MBNL1, RBM3, TAF15, TARDBP, and U2AF2, bound to the same regions in either the CDS or 3′ UTR as those sites which were hyper-m[6]A-methylated and selectively recognized by YTHDF3 upon nutrient starvation (Supplementary Figs. 19–21). Although we did not provide any functional data showing those above-mentioned RBPs involved in contributing to YTHDF3 binding specificity, we do believe such association results could narrow down the key RBPs in regulating YTHDF3 binding with FOXO3 mRNA upon nutrient deficiency.

## Discussion

How cells respond to metabolic cues to switch on autophagy remains a significant question in studying cellular homeostasis. Accumulating evidence has shown that post-translational modifications (PTMs) and nutrient-sensing kinase cascades, such as the AMPK-TSC1/2-MTOR axis[59], can impact autophagy induction in response to fasting or energy restriction[60]. Multiple transcription factors and histone modifiers have also been proved to play important roles in this process[61]. However, little is known about the role epitranscriptomic modifications play in autophagy regulation. Recently, it was shown that the m[6]A eraser FTO could increase the stability of ULK1, ATG5, and ATG7 transcripts, up-regulating autophagy in a YTHDF2-mediated manner[62,63]. Yet, neither study emphasized the role of epitranscriptome players in autophagy

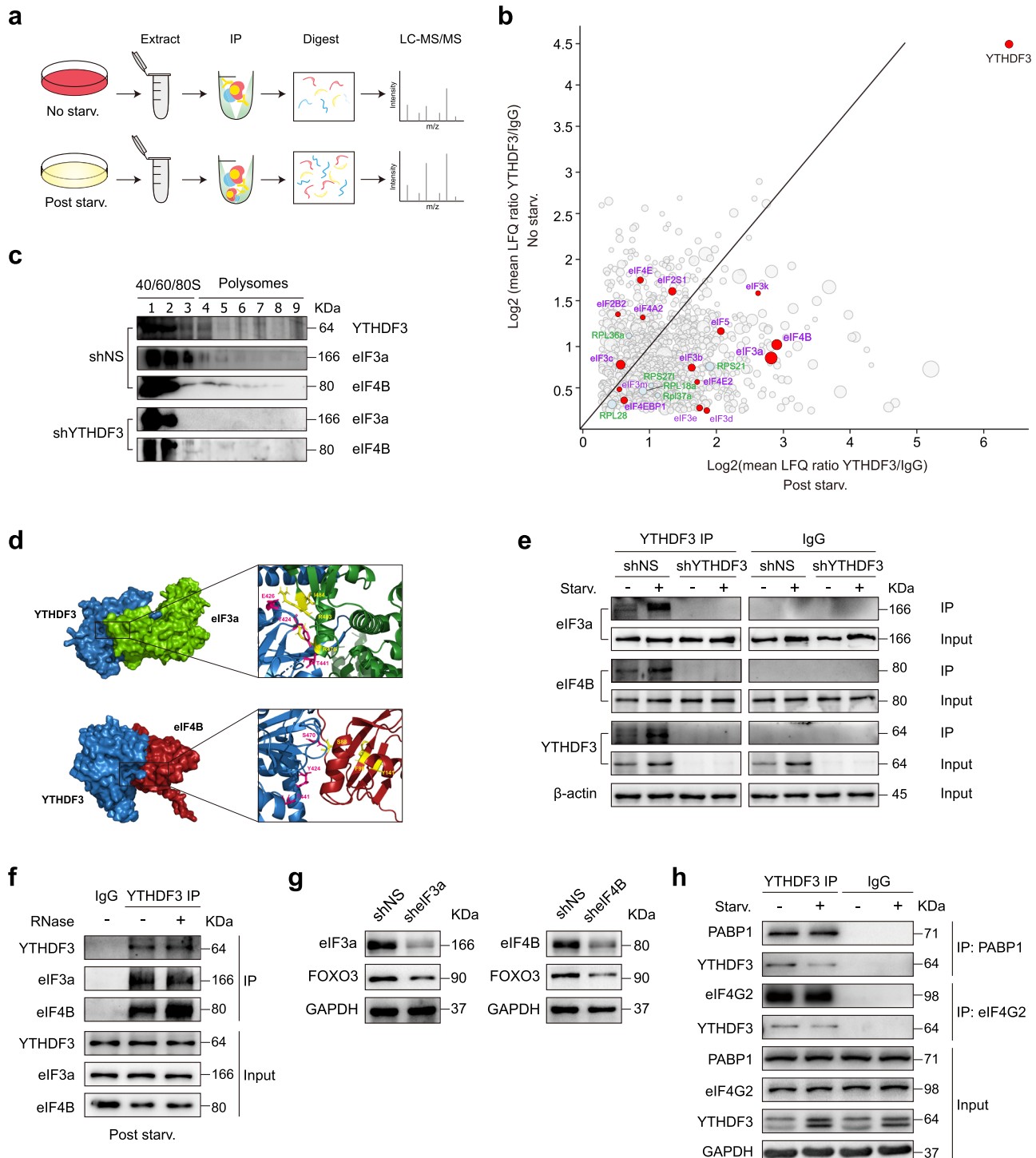

**Fig. 8 | YTHDF3 interacts with eIF3a and eIF4B to promote FOXO3 translation.**
**a** YTHDF3 immunoprecipitation workflow in MEFs under normal and nutrient-free conditions. **b** YTHDF3-specific interactors were identified through quantitative mass spectrometry. eIFs are labeled in red. 40S and 60S ribosomal subunits are labeled in light blue. Diameters correlate with mean numbers of unique peptides (n = 2). **c** Immunoblot analyses for eIF3a and eIF4B proteins from polysome fractions in shNS and shYTHDF3 MEFs. **d** Surface view of docked YTHDF3-eIF3a and YTHDF3-eIF4B complexes. YTHDF3, eIF3a, and eIF4B are colored in blue, green, and red, respectively. Inset, magnified views of the interacting residues are drawn in stick representation and labeled (in pink for YTHDF3 and in yellow for eIF3a or

eIF4B). **e** Immunoblot analyses of YTHDF3-immunoprecipitated proteins from shNS and shYTHDF3 MEFs, with or without nutrient deprivation. Total protein amounts of YTHDF3, eIF3a, and eIF4B were used as inputs. **f** Immunoblot analyses of YTHDF3 immunoprecipitation lysates, with or without RNase A treatment. eIF3a and eIF4B were detected. **g** Immunoblot analyses of FOXO3 in sheIF3a, sheIF4B, and control MEFs, respectively. **h** Immunoblot analyses for YTHDF3 in PABP1- or eIF4G2-immunoprecipitated lysates from MEFs with or without nutrient deprivation. Total protein amounts of PABP1, eIF4G2, and YTHDF3 were used as inputs. Source data are provided as a Source Data file.

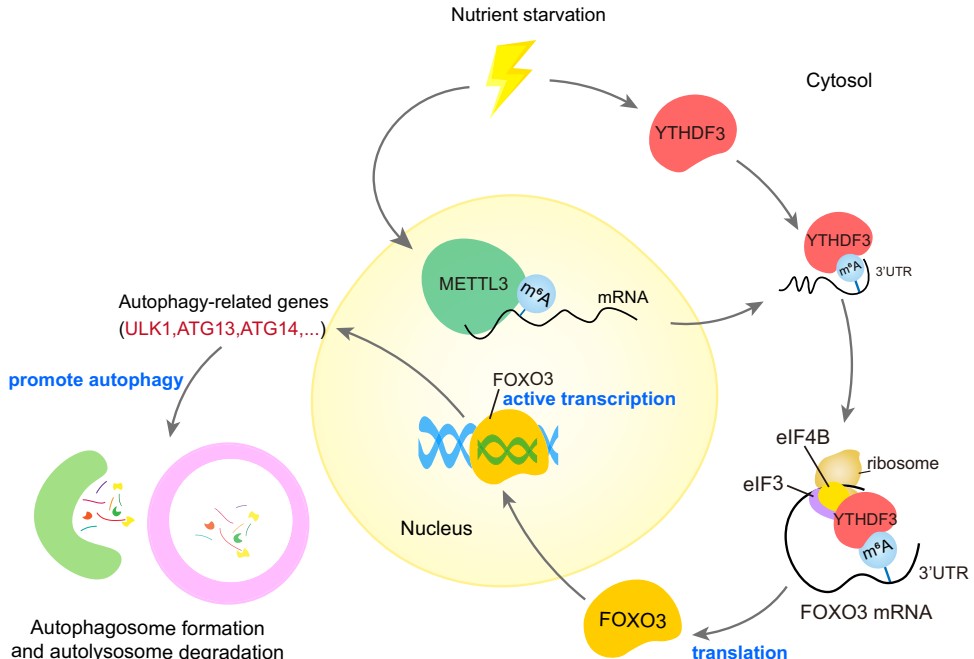

**Fig. 9 | A proposed model of YTHDF3-promoted autophagy induction.** Under nutrient deprivation, the m⁶A reader YTHDF3 and m⁶A writer METTL3 are up-regulated. The latter promotes m⁶A hypermethylation at the CDS and 3′UTR regions around the stop codon of the FOXO3 transcript. YTHDF3 facilitates FOXO3 translation by binding to these m⁶A modifications around the stop codon and recruiting the eIF3a subunit and eIF4B to the 5′ end of FOXO3 mRNA, forming mRNA loops. FOXO3 then activates the transcription of core autophagy-related genes and promotes autophagy.

induction in response to nutrient starvation, nor did they elucidate the mechanism behind how nutrient cues selectively elicit the expression of a variety of autophagy-related genes. Our results showed that the m⁶A reader YTHDF3 and m⁶A writer METTL3 are up-regulated during nutrient starvation. Upon nutrient deficiency, the nutrient responder YTHDF3 augments recognition of METTL3-mediated m⁶A modifications around the stop codon in FOXO3 transcripts, consequently facilitating FOXO3 translation. The biosynthesis of FOXO3 activates the transcription of its target autophagy-related genes, thereby inducing autophagy to maintain energy homeostasis (Fig. 9).

Usually, nutrient deprivation leads to curtailed anabolism associated with growth, greater catabolism, and the use of essential nutrients to support survival. Autophagy is a catabolic process that degrades damaged organelles and misfolded proteins, repurposing them to maximize cellular viability, which is activated upon nutrient deprivation. Mechanistically, the nutrient-sensing signaling network changes within a short time under starvation and overall cellular protein synthesis are attenuated, whereas a set of stress- and autophagy-related mRNAs undergo continued or enhanced translation[64,65]. In the current study, we observed a global increase in m⁶A-methylated mRNAs as well as cytoplasmic m⁶A accumulation due to nutrient deficiency, implying m⁶A methylation may play an important role in cellular survival under such hostile conditions. Under stressful circumstances, controlling m⁶A might be an ideal strategy to control autophagic activity, thus maintaining cell survival and metabolism. The double safeguard mechanism, including adequate METTL3 protein maintenance and its dynamic catalytic activity fine-tuning, ensures proper m⁶A levels at different starvation periods, which is beneficial for cell metabolism and survival. Consistently, a recent study reported that dynamic 5′UTR m⁶A modification in response to nutrient deprivation regulates ATF4 alternative translation by influencing translation initiation site (TIS) selection[66]. Unlike ATF4, we found that FOXO3, the master transcriptional regulator of autophagy genes, exhibits increased m⁶A methylation in the CDS and 3′UTR regions around the stop codon in response to nutrient starvation, and these m⁶A modification events are quite important for FOXO3 translation. This finding expands the current understanding that dynamic m⁶A changes in response to cellular stress mainly occur in 5′UTRs[67].

According to our data, we propose that METTL3 is a stress-responsive gene to nutrient scarcity since the up-regulation of METTL3 upon starvation is due to the decreased ubiquitination of METTL3, thus promoting its stabilization. However, additional mechanisms are likely modulating dynamic METTL3 enzymatic activity and m⁶A during nutrient deprivation. The following possibilities cannot be excluded: first, nutrient deprivation could reduce intracellular levels of SAH and SAM, disrupting their balance and effects on METTL3, and affecting METTL3's methyltransferase activity[43]. Secondly, altered TCA cycle metabolites, such as αKG, could change the enzymatic activities of m⁶A erasers FTO and ALKBH5, leading to changes in cellular m⁶A modifications as well[68,69]. Finally, nutrient starvation could inhibit NADPH production, decrease FTO activity, and upregulate m⁶A levels[70]. How exactly nutrient deprivation affects the enzymatic activities of m⁶A methyltransferases and demethylases requires further investigation. On the other hand, METTL3 might probably regulate genes' expression via different m⁶A readers. For instance, a recent study illustrated METTL3 could induce a decay of ATG7 transcript in a YTHDF2-dependent manner in senescent fibroblast-like synoviocytes (FLSs)[71]. Another study found YTHDF1 promoted BECN1 mRNA stability via recognizing the m⁶A sites within BECN1 transcripts in hepatic stellate cells (HSCs)[72]. Therefore, how METTL3 regulates different autophagy-related genes' expression and autophagic activity might be a comprehensive result via different regulatory mediators and layers, particularly emerging evidence indicated that METTL3 had other functions independent of its m⁶A role[73].

FOXO3 is one of the first transcriptional regulators reported to be linked to autophagy[74]. It shuttles from the cytoplasm to the nucleus, binding to the promoters of a subset of autophagy-related genes to enhance their expression, thereby promoting autophagy[50]. A recent study illustrated how FOXO3 reduced the expression of ATG proteins and decreased autophagic activity in the hypoxic microenvironment of HCC[75], which is opposite to the general role of FOXO3 in regulating autophagy in most cell types[76,77]. To date, the mechanism underlying

FOXO3 regulation has attracted in-depth investigation. Co-regulators interacting with FOXO3, such as SIRT3, CHOP, and PP2A, can increase the target genes' transcriptional activity, inducing FOXO3-dependent gene expression. PTMs of FOXO3, such as AMPK-dependent phosphorylation, PRMT6-induced methylation, and PARP1-mediated PARylation, lead to FOXO3 nuclear translocation and transcriptional activation[78–80]. Transcription factors, including E2F1, p53, and HIF-1a, directly regulate FOXO3 transcription in response to stress stimuli[81–83]. In post-transcriptional regulation of FOXO3, multiple microRNAs target the 3′UTR region of FOXO3 mRNA to suppress FOXO3 activation[84]. Here, we report that FOXO3 mRNA can be m6A hypermethylated by METTL3, and the m6A reader YTHDF3 can act as a switch to affect FOXO3 mRNA translation, thus tuning levels of FOXO3 and the transcription of its targets in response to amino acid starvation. Our findings provide insights into the mechanism behind FOXO3 post-transcriptional regulation. The posttranslational modification of FOXO3 is also important, our data suggested YTHDF3 may have a key role in regulating FOXO3 translation rather than FOXO3 posttranslational modification such as phosphorylation. However, our data could not exclude the effect of all types of posttranslational regulation on FOXO3 activity.

The constitutively enriched m6A modifications around the stop codon and at 3′UTRs have been reported to form a closed mRNA loop when the m6A reader YTHDF1 binds to eIF3, thus promoting mRNA translation[85]. FOXO3 mRNA has a particularly long 3′UTR, accounting for nearly 65% of the transcript's length. The length of this 3′UTR suggests that this region may form an mRNA loop to regulate FOXO3 transcript translation. According to the SRAMP algorithm's prediction results, m6A sites in FOXO3 mRNA predicted with the highest confidence are highly enriched around the stop codon, suggesting these sites may play an important role in mRNA loop formation. In our study, we revealed that the m6A reader YTHDF3, rather than other YTH family proteins, is significantly induced during nutrient starvation, and the m6A writer METTL3 is also up-regulated during this process, resulting in hypermethylation of m6A modifications around the stop codon in FOXO3 transcripts. YTHDF3 further promotes the translation of FOXO3 mRNA by binding to these m6A sites. Previous studies showed that besides the YTHDF1-eIF3 looping model, m6A modifications localized to 5′UTRs can recruit eIF3 directly to induce translation, independent of eIF4E cap binding[28]; the m6A sites close to the stop codon can also form an mRNA loop through the interaction between METTL3 and eIF3h, enhancing mRNA translation[86]. Based on our co-IP LC-MS/MS data, YTHDF3 interacts with 7 subunits of eIF3, as well as eIF4B, and there is increased binding upon nutrient deficiency. This suggests that YTHDF3 promotes translation in a way similar to the METTL3-eIF3h-mediated mRNA loop. In antiviral innate responses, YTHDF3 has been reported to promote FOXO3 mRNA translation in an m6A-independent manner by binding to the translation initiation region of its mRNA, mediating PABP1-eIF4G2 interactions[58]. However, in nutrient starvation-induced autophagy, our experimental evidence supports an alternative closed-loop model: one in which YTHDF3 binds to m6A modifications around the stop codon in FOXO3 transcripts, and then promotes FOXO3 translation by recruiting eIF3a subunits and eIF4B to the 5′ end of FOXO3 mRNA to form mRNA loops.

Besides FOXO3, there were some other genes also identified within the list of YTHDF3-bound hypermethylated genes up-enriched in response to nutrient scarcity. Similar to FOXO3, some of these genes were reported to play a role in regulating autophagy under basal or starved conditions. For instance, CDKN1B was shown to promote autophagy upon nutrient shortage by repressing MTORC1 or directly facilitating lysosomal function[87,88]; SESN2 contributed to autophagy through AMPK-mTORC1 signaling[89]; Mitochondrial PLD6 promoted BNIP3L-induced MTOR-RPS6KB activation and triggered mitophagy[90]. However, since we demonstrated that YTHDF3 had little effect on AMPK or mTORC1 signaling pathways, we speculated that these

aforementioned genes might not be the primary functional targets of YTHDF3 regulating autophagy. Furthermore, several genes were reported to involve in autophagic multi-step or stage-specific regulation. For example, CCR4 deadenylase was reported to play important role in regulating RNA stability of diverse key autophagy-related genes[91]. FNIP2 enhanced FLCN-GABARAP interactions and drove autophagy flux at different stages[92]. BMF interacted with BECN1-BCl2 complexes, or more autophagy-activating pathways, to regulate autophagy[93]. The pro-oxidant protein TXNIP suppressed ATG4B activity and promoted autophagosome maturation through ROS regulation[94]. ZFYVE26 mutations led to defects in autophagosome maturation and autophagic lysosome reformation[95]. Nevertheless, whether and how these abovementioned genes were the important targets of YTHDF3 in regulating autophagy requires further investigation.

Overall, our study reveals an important link between epitranscriptomics and autophagy. We provide evidence that the m6A reader YTHDF3 functions as a nutrient responder to bind m6A hypermethylation, installed by METTL3, around the stop codon of FOXO3 mRNA, that YTHDF3 then recruits eIFs to rapidly promote FOXO3 translation, further transcriptionally activates a subset of core autophagy genes, thus promoting autophagy.

## Methods

### Chemical reagents and antibodies
Bafilomycin A1 (Baf.A1, B101389), acridine orange (AO, A121748), and rapamycin (S115842) were purchased from Aladdin. Actinomycin D (Act D, HY-17559) and cycloheximide (CHX, HY-12320) were purchased from MedChemExpress. N6-methyladenosine (m6A, S3190), Adenosine (S1647), and Guanosine (S2439) were purchased from Selleck. (R,S)-S-Adenosyl-L-methionine-d₃(S-methyl-d₃)Tetra(p-toluenesulfonate) Salt (d₃-SAM, D-4093) was purchased from C/D/N ISOTOPES INC. N6-Methyladenosine-d₃ (d₃-m6A, M275897) was purchased from Toronto Research Chemicals. All antibodies used are listed in Supplementary Table 1.

### Cell culture
MEFs were isolated from mouse embryos at 13.5 days post coitum and cultured in high glucose Dulbecco's modified Eagle's medium (DMEM) supplemented with 10% fetal bovine serum (FBS, PAN), 2 mM gluta-MAX (Gibco), 1% non-essential amino acids (Gibco), 50 μM β-mercaptoethanol (Sigma), and 100 U/ml penicillin-streptomycin (Gibco) at 37 °C under 5% $CO_2$. HEK293T (SCSP-502), 3T3-L1 (SCSP-5038), C2C12 (SCSP-505), and HepG2 (SCSP-510) cell lines were obtained from the Cell Bank of Shanghai Institute of Cell Biology, Chinese Academy of Sciences (Shanghai, China), and maintained in high glucose DMEM containing 10% FBS and 100 U/ml penicillin-streptomycin. All cells used in the study were routinely tested for mycoplasma contamination. For starvation conditions, MEFs were incubated in a nutrient-deprived medium, Hank's balanced salt solution (HBSS, Hyclone), with 10 mM HEPES (Gibco) for the indicated time.

### Plasmids
The lentiviral pLV-EGFP-LC3B and pLV-mCherry-EGFP-LC3B plasmids were generated by subcloning EGFP-LC3B fragments (from pEGFP-LC3B, Addgene, #24920) and mCherry-EGFP-LC3B fragments (from pmCherry-EGFP-LC3B) into pCDH-EF1-Neo lentiviral vectors (System Biosciences, Palo Alto, CA, #CD533A-2). Lentiviral short-hairpin RNA (shRNA) constructs for mouse YTHDF3 and METTL3 were obtained according to the pLKO.1-puro vector protocol (Addgene, Cambridge, MA, USA). Oligonucleotides used for the mouse YTHDF3 knockdown were as follows: shYTHDF3, GGACGTGTGTTTATAATTA; shYTHDF3-2, GACTAGCATTGCAACCAAT. Oligonucleotides used for the mouse METTL3 knockdown were as follows: shMETTL3, GGAGATCCTAG

AGCTATTAAA; shMETTL3-2, GCACACTGATGAATCTTTAGG. YTHDF3 and METTL3 lentiviral expression plasmids were constructed by cloning the full-length ORFs of the mouse YTHDF3 gene (NM_001145919) and the mouse METTL3 gene (NM_019721) into pEZ-Lv242 vectors (GeneCopoeia, Guangzhou, China). The catalytically inactivated mutant of METTL3 (aa395-398, DPPW/APPA) was generated by subcloning the mutant METTL3 fragment from the pFLAG-CMV2-METTL3 (mutant) vector into the pEZ-Lv242 lentiviral vector. The wild-type and mutant FOXO3-CDS lentiviral expression plasmids were generated by subcloning CDS fragments of mouse FOXO3 genes, containing either wild-type $m^6A$ motifs or mutant motifs ($m^6A$ replaced by T), into the pEZ-Lv242 lentiviral vectors. Detailed information regarding the wild-type and two mutant FOXO3 CDS fragments is provided in Supplementary Table 2.

### Generating *YTHDF3*$^{-/-}$ mice and MEFs

*YTHDF3*$^{+/-}$ mice on a C57BL/6 background were generated using the CRISPR-Cas9 system from Cyagen Biosciences (Suzhou, China). Two sgRNAs were designed to target Exon 3 of the YTHDF3 gene. The oligonucleotide sequences are as follows: sgRNA1: AGTCACAAATAGT TACTTGAAGG; sgRNA2: AAACATATACTGTGAAGCGTTGG. *YTHDF3*$^{-/-}$ MEFs were generated from *YTHDF3*$^{-/-}$ mouse embryos (E13.5) by intercrossing *YTHDF3*$^{+/-}$ mice. The genotypes of the generated mouse embryos were identified with the following primers: primers1 (forward: 5′-CTTCAGTGCATGCTAAATACAC-3′; reverse: 5′-CTAAGATTTCAGACA ATTTTCCAC-3′); primers2 (forward: 5′-CTATAAGCTAAGTCATGTG CCAC-3′; reverse: 5′-CTAAGATTTCAGACAATTTTCCAC-3′). All animal experiments were conducted in strict accordance with Southern Medical University guidelines for the Care and Use of Experimental Animals. The animal program was approved by the Animal Experimental Ethics Committee of Southern Medical University. All surgeries were performed under pentobarbital sodium anesthesia, and every effort was made to minimize animal suffering.

### Sample preparation for LC-MS-based proteomic analysis

Proteins were precipitated overnight at -20 °C in trichloroacetic acid (TCA), cleaned with triple excess ice-cold acetone, and lyophilized. The precipitated proteins were solubilized in denaturation buffer [100 mM ammonium bicarbonate and 5% acetonitrile (ACN) mixture], then reduced with 20 mM dithiothreitol (DTT), followed by alkylation with 40 mM Iodoacetamide (IAA). The proteins were then digested with trypsin protease (Pierce) overnight at 37 °C. The resulting peptide mixtures were desalted using C18 spin columns (Pierce) and dried in a vacuum lyophilizer.

### LC-MS-based proteomics analysis

The tryptic peptides were dissolved in 0.1% formic acid (solvent A), and directly loaded onto a reversed-phase analytical column (Acclaim PepMap™,15-cm length, 75 μm i.d.). The gradient was comprised of an increase from 5% to 10% solvent B (0.1% formic acid in 80% acetonitrile) over 28 min, 10-22% in 55 min, 22-30% in 27 min, and climbing to 100% in 5 min then holding at 100% for 5 min, all at a constant flow rate of 300 nL/min on an EASY-nLC 1200 UPLC system (Thermo). The peptides were subjected to NSI source followed by tandem mass spectrometry (MS/MS) in Orbitrap Fusion™ Tribrid™ LC mass spectrometer (Thermo) coupled online to the UPLC. The electrospray voltage applied was 2.3 kV. The $m/z$ scan range was 350–1500 for the full scan, and intact peptides were detected in the Orbitrap at a resolution of 120,000. Peptides were then selected for MS/MS using NCE setting as 30 and the fragments were detected in Ion Trap. Data were acquired in a data-dependent mode that time between master scan was set 3 s. Automatic gain control (AGC) was set at 5E3 and the fixed first mass was set as $120m/z$.

For database search, raw files were processed using MaxQuant (version 1.5.8.3) and Andromeda search engine against the mouse Uniprot database. The spectra were searched with a mass tolerance of 6 ppm for precursors and 0.5 Da for fragment ions. The following parameters were set: 'trypsin/P' was set as enzyme specificity, and 'max missed cleavages' was set to 2. Acetylation of protein N termini and oxidation of methionine were considered as variable and carbamidomethylation of cysteine residues as fixed modification. The options 'match between runs', 'decoy searches of reversed sequences', and 'LFQ' were enabled. 'LFQ min. ratio count' was set to 2. Proteins were identified based on at least one unique peptide with a length of six amino acids and a maximum mass of 4600 Da. Contaminant sequences were included in the search. The false discovery rate for peptide-spectrum matches (PSM), for both the protein and site, were each set at 1%.

For protein quantification and statistical analyses, the MaxQuant output 'proteinGroups.txt' files were loaded into Perseus (version 1.6.2.3). The protein entries were filtered to exclude reverse hits, potential contaminants, and proteins only identified by site. The LFQ intensities were log2 transformed and all missing values were imputed from a fitted normal distribution using default settings in Perseus. For the differential proteomic analysis upon nutrient starvation, $P$ values were calculated using an unpaired two-tailed Student's $t$-test ($n = 3$, assuming equal variance) with a 95% confidence interval ($P < 0.05$). The samples were analyzed in triplicates and in random order.

### Protein extraction and western blot analysis

Total proteins from cultured cells were extracted using a Total Protein Extraction Kit (Keygen), according to the manufacturer's guidelines. Protein concentrations were determined using a BCA Protein Assay Kit (Keygen). Lysates were mixed with the appropriate volume of 5× loading buffer, boiled, and loaded into SDS-PAGE gels for separation. Proteins were transferred to polyvinylidene difluoride (PVDF) membranes using a Mini Trans-Blot™ system (Bio-Rad). The membranes were then blocked with 5% skim milk in TBST, incubated with primary antibodies overnight at 4 °C, and then incubated with HRP-conjugated secondary antibodies for one hour at room temperature. The blots were imaged with Immobilon™ Western Chemiluminescent HRP Substrate (Millipore) and the ChemiDoc™ XRS + imaging system (Bio-Rad). Images were quantified using Bio-Rad Image Lab software. The primary and secondary antibodies used are listed in Supplementary Table 1.

### RNA isolation, reverse transcription, and real-time quantitative reverse transcription PCR (qRT-PCR)

Total RNA was extracted using RNAiso Plus (Takara) according to the recommended protocol. RNA concentrations were measured using a NanoDrop 2000 (Thermo). cDNAs were reverse transcribed with the PrimeScript RT reagent kit (Takara). qRT-PCR was performed using ChamQ Universal SYBR qPCR Master Mix (Vazyme) and a LightCycler™ 480 system (Roche). The primers used are listed in Supplementary Table 3. GAPDH was used as an endogenous control for normalization. The ΔΔCt method was used for relative quantification.

### Immunofluorescence staining

Cells grown on glass coverslips were fixed in 4% paraformaldehyde for 15 min, and then permeabilized in 0.2% Triton X-100 for 30 min. The cells were then blocked in PBS containing 5% goat serum for 12 h, before incubating with the primary antibody at 4 °C overnight. Following incubation with the secondary antibody in PBST at room temperature for 1 h, nuclei were counter-stained with DAPI (Solarbio). The coverslips were then mounted onto slides. Images were visualized and collected using a Nikon A1 confocal microscope.

### Transmission electron microscopy (TEM)

The cells were scraped off, collected, fixed in 0.1× PBS with 2.5% glutaraldehyde and 0.5% osmium tetroxide, dehydrated with ethanol, and embedded in Epon. Thin sections were mounted on grids, counter-

stained with uranyl acetate and lead citrate, and then observed under a Hitachi-7500 electron microscope.

## Autophagy flux measurement

To indicate autophagic degradation, the cargo protein p62 was detected using western blotting. For autophagosome generation, cells were subjected to nutrient deprivation in the presence or absence of bafilomycin A1 (Baf.A1), a lysosomal inhibitor, at a concentration of 40 nM. LC3-II levels were measured using western blots. GFP-LC3 puncta were observed with a confocal microscope, and 10 visual fields were randomly selected to count the number of spots per cell.

## DQ-BSA

Lysosomal hydrolase activity was measured using DQ™ Red BSA (Invitrogen). MEFs cultured on coverslips were incubated with 10 µg/mL DQ™ Red BSA for one hour (37 °C, 5% CO$_2$), followed by counter-staining with Hoechst 33342. Images were collected using a confocal microscope (Nikon A1, Japan). Fluorescence intensities were quantified using ImageJ software.

## Acridine orange assay

Cells were preloaded with 1 µg/ml acridine orange (AO, Aladdin) for 15 min at 37 °C, followed by three fast washes. Cells were then analyzed using a FACSCalibur™ flow cytometer (BD) under green and red channels. The data were processed as previously described[34].

## LysoTracker red staining

Lysosomal acidity was probed with LysoTracker™ Red. Cells cultured on coverslips were incubated with 50 nM LysoTracker™ Red DND-99 (Invitrogen) for 10 min at 37 °C, followed by counter-staining with Hoechst 33342. The fluorescence intensity of different staining results was examined with a confocal microscope (Nikon A1, Japan) and quantified with ImageJ software.

## Cathepsin activity assay

Lysosomal cathepsin B (CTSB) enzymatic activity was detected using Magic Red™ reagents according to the manufacturer's protocol. Cells cultured on coverslips were incubated with Magic Red™ CTSB reagents (Immunochemistry Technology) for 15 min at 37 °C, followed by staining with Hoechst 33342. Images were acquired using confocal microscopy (Nikon A1, Japan).

## Quantification of m$^6$A levels in mRNA by LC-MS/MS

Total RNA was isolated with RNAiso plus (Takara) according to the manufacturer's protocol. mRNA was then extracted using a Dynabeads™ mRNA Purification Kit (Invitrogen) as previously described[96]. For m$^6$A quantification, 200 ng polyadenylated mRNAs were digested by 1.2 U of nuclease P1 (Sigma) dissolved in 25 µl of NH$_4$OAc buffer (20 mM, pH = 5.3) at 42 °C for 2 h. 3 µl of NH$_4$HCO$_3$ (1 M) and 1 U alkaline phosphatase (Sigma) were then added, followed by an additional two hours of incubation at 37 °C. The samples were then diluted with formic acid to 50 µl and filtered with a 0.22 µm syringe filter (Millipore), and 5 µl of the solution was injected into the LC-MS/MS. Nucleosides were separated by a reversed-phase ultra-performance liquid chromatography system on a C18 column, coupled to mass spectrometry detection with an Orbitrap Fusion™ Tribrid™ LC mass spectrometer (Thermo) in positive electrospray ionization mode. Nucleoside quantification was based on retention time and nucleoside-to-base ion transitions: 268-136 for A, and 282-150 for m$^6$A. Using the standard curve generated from reference standards running in the same batch, the concentrations of A and m$^6$A in the samples were calculated. The modification level of m$^6$A was calculated as the percentage of m$^6$A out of the total amount of A, to normalize the amount of mRNA injected from different samples.

## m$^6$A dot blot

An m$^6$A dot blot was performed according to a published protocol[97], with some modifications. Polyadenylated mRNAs were purified using a Dynabeads™ mRNA Purification Kit (Invitrogen). mRNA samples were denatured at 95 °C for 3 min, followed by spotting on a Hybond-N+ nylon membrane (GE). Samples were then UV crosslinked at UV 254 nm, 0.15 J/cm$^2$. After blocking with 5% BSA in TBST for 8 h, the membrane was incubated with anti-m$^6$A antibody (1:1500, Synaptic Systems) overnight at 4 °C. HRP-conjugated goat anti-rabbit IgG (bio-world) was then added to the blots for one hour at room temperature, and the membrane was developed using Immobilon™ Western Chemiluminescent HRP Substrate (Millipore) and a ChemiDoc™ XRS+ imaging system (Bio-Rad). Methylene blue staining was used as a loading control.

## In vitro assay for m$^6$A methyltransferase activity

METTL3 FLAG-tagged proteins were purified from MEFs stably expressing FLAG-tagged METTL3 proteins under normal conditions or following starved conditions. Purifications were performed using a FLAG-tagged Fusion Protein Purification Kit (DIA-AN) according to the manufacturer's protocol. A 5′-UACACUCGAUCUGGACUAAAGCUGC UC-3′ RNA probe containing the canonical RRACH sequence was designed according to published sequences[40] and synthesized in vitro by RiboBio (Guangzhou, China). The methyltransferase activity assay was carried out as previously described[40], in a 50 µl reaction mixture containing the following components: 0.15 nmol RNA probe, 0.15 nmol fresh purified METTL3 protein, 0.8 mM d$_3$-SAM, 80 mM KCl, 1.5 mM MgCl$_2$, 0.2 U µl$^{-1}$ RNasin, 10 mM DTT, 4% glycerol and 15 mM HEPES (pH 7.9). Pre-incubation was performed at 90 °C for 3 min, followed by 40 cycles of -2 °C/cycle within 30 min for annealing. After incubation at 16 °C for 12 h, the resultant RNAs were recovered using RNAiso plus (Takara) and extracted. This was followed by complete digestion of single nucleosides using nuclease P1 and alkaline phosphatase, and then LC/MS/MS analysis. The nucleosides were quantified by using nucleoside-to-base ion mass transitions of 285 to 153 (d$_3$-m$^6$A) and 284 to 152 (Guanosine, G). G served as an internal control for the amount of total RNA probe in each reaction mixture.

## Ubiquitination assay

Protein co-immunoprecipitation was conducted using a Pierce™ Crosslink Magnetic IP/Co-IP Kit (Thermo) according to the manufacturer's protocol with slight modifications. Cells were pretreated with 20 µM MG132 for 4 h, and lysed in HEPES lysis buffer (20 mM HEPES (pH 7.4), 10 mM KCl, 100 mM NaCl, 1 mM MgCl$_2$, 1 mM EDTA, 0.1 mM EGTA, and 0.5 mM CaCl$_2$) for 4 min on ice. Ubiquitin aldehyde was added to the lysate to a final concentration of 1 µM following the mix sample on a rotator at 20 rpm for 15 min at 4 °C. Cell debris was removed by centrifuging at 13,000 × g for 15 min at 4 °C. Protein A/G Magnetic Beads were crosslinked with anti-FLAG antibody (F1804, Sigma-Aldrich). Then cell lysates were combined with the antibody-crosslinked beads through gentle rocking at 4 °C overnight. After washing twice with the IP Lysis/Wash Buffer and washing once with water, the bound proteins were eluted from the beads and then subjected to ubiquitin immunoblotting.

## RNA electrophoretic mobility shift assay (EMSA)

The biotin-labeled RNA oligonucleotides containing A or m$^6$A were obtained from Genecefe Biotech (China). The oligo sequences are provided in Supplementary Table 4. The mouse YTHDF3 recombinant proteins were obtained from Sino Biological (Beijing, China). The RNA EMSA assay was carried out using a LightShift™ Chemiluminescent RNA EMSA Kit (Thermo Scientific) following the manufacturer's instructions and performed as previously described[98]. Briefly, the biotin-labeled RNA probes with A or m$^6$A (20 pmol) were mixed with 6 µg of recombinant YTHDF3 proteins and incubated at room temperature for 30 min. The

mixture was then mixed with 5× loading buffer and separated on a 6% TBE gel in 0.5×TBE buffer at 100 V for 30 min, then transferred to a positively charged Hybond-N + nylon membrane (GE). The membrane was crosslinked at UV 254 nm, 0.15 J/cm$^2$, blocked and incubated with HRP-linked streptavidin. This was then developed using a chemiluminescent HRP substrate and a ChemiDoc™ XRS + imaging system (Bio-Rad).

## Cell fractionation
Nuclear and cytoplasmic fractions were prepared with NE-PER® Nuclear and Cytoplasmic Extraction reagents (Pierce), following the manufacturer's protocol (with slight modifications). Cells were harvested by scraping and centrifuging at 500 × g at 4 °C for 5 min. The cell pellets were then washed once with ice-cold PBS, suspended in CER I, and incubated on ice for 10 min. After adding ice-cold CER II, vigorously vortexing, and incubating on ice for one minute, the insoluble fraction was precipitated with a centrifuge, and supernatant cytoplasmic extracts were collected. After washing three times with ice-cold PBS, the cell pellet was added to the NER reagent, incubated on ice, and vortexed for 15 s every 10 min, for a total of 40 min. Finally, the supernatant was collected through centrifuging and the nuclear extracts were harvested.

## RNA stability assay
MEFs were treated with actinomycin D at a final concentration of 5 μg/mL for the indicated time periods and collected. Total RNAs were extracted and analyzed with qRT-PCR. The half-life ($t_{1/2}$) of mRNA was calculated using ln 2/-slope, and GAPDH was used for normalization.

## Polysome profiling
5-50% linear sucrose gradients were freshly prepared in an ultracentrifuge tube (Beckman) using an automated gradient maker (BioComp). Cells were pre-treated with 100 μg/ml Cycloheximide (CHX) (MCE) at 37 °C for 10 min, washed twice with ice-cold PBS containing 100 μg/ml CHX, and collected. Cells were then lysed in hypertonic buffer containing 20 mM Tris-HCl (pH 7.4), 300 mM NaCl, and 10 mM MgCl$_2$, and 0.5% sodium deoxycholate, 1% Triton X-100, 1 mM DTT, 100 μg/ml CHX and 300 U/ml of RNase inhibitor were added. Cell debris was removed via centrifugation at 16,000 × g for 10 min at 4 °C. 500 μL of supernatant was then loaded into the sucrose gradients, followed by centrifugation at 4 °C and 36,000 rpm for 2 h (SW41Ti rotor, Beckman). Samples were then fractioned and analyzed with a Piston Gradient Fractionator (Bio-Comp) and fraction collector (Gilson). 5 ng of polyadenylated synthetic luciferase mRNA (Promega) was added to each fraction for normalization. RNAs were extracted from each fraction and subjected to qRT-PCR analysis.

## Dual-luciferase reporter assay
Fragments of either FOXO3-3′UTR-WT or FOXO3-3′UTR-Mut (m$^6$A changed to T) were inserted into pEZX-MT06 vectors (GeneCopoeia) to generate pFOXO3-3′UTR-WT and pFOXO3-3′UTR-Mut plasmids. Luciferase activity was detected with a Dual-Luciferase Reporter Assay System (Promega) 72 h after transfecting pFOXO3-3′UTR-WT or pFOXO3-3′UTR-Mut. The relative Fluc/Rluc activity was calculated by normalizing the activity of firefly luciferase to that of renilla luciferase. Each group's assay was repeated in triplicate.

## Protein co-immunoprecipitation
Protein co-immunoprecipitation was conducted using a Pierce™ Crosslink IP Kit (Thermo) according to the manufacturer's protocol. After washing with pre-chilled PBS and 1× coupling buffer, cells were lysed in IP lysis buffer for 5 min on ice. Cell debris was removed by centrifuging at 13,000 × g for 10 min at 4 °C. Protein A/G Plus Agarose was preloaded with anti-YTHDF3 antibodies (sc-377119, SANTA CRUZ).

An immunoprecipitation control was performed using mouse IgG (Millipore). Cell lysates were combined with the antibody-crosslinked agarose through gentle rocking overnight at 4 °C overnight. After washing twice with lysis/wash buffer and washing once with 1× conditioning buffer, precipitates were eluted, followed either by LC-MS sample preparation or equilibration with 5× sample buffer, DTT, and heating for immunoblot analysis.

## RNA immunoprecipitation (RIP)
RIP was performed using a Magna RIP Kit (Millipore), following the manufacturer's instructions. Cells seeded in 15 cm dishes were washed and collected with ice-cold PBS, then lysed in RIP lysis buffer with added protease and RNase inhibitors. Both YTHDF3 antibodies (sc-377119, SANTA CRUZ) and control mouse IgG (Millipore) were conjugated to protein A/G magnetic beads via incubation for 30 min at room temperature, followed by three washes and incubation with cell lysate at 4 °C overnight. After six washes, the beads were resuspended, followed by protein digestion with proteinase K at 55 °C for 30 min. The input and immunoprecipitated RNAs were recovered using RNAiso plus, extracted, and subjected to next-generation sequencing (NGS) or qRT-PCR analysis.

## MeRIP-seq
Total RNA was extracted from cells using RNAiso plus (Takara) according to the manufacturer's protocol. mRNA was further purified using a Dynabeads™ mRNA Purification Kit (Invitrogen). MeRIP-seq was performed by Cloudseq Biotech Inc. (Shanghai, China). Summarizing briefly, m$^6$A RNA immunoprecipitation (IP) was performed using a GenSeq™ m$^6$A RNA IP Kit (GenSeq). Both the input and m$^6$A IP samples were prepared for NGS. The library was constructed using a NEBNext® Ultra II Directional RNA Library Prep Kit (NEB). The quality of the library was evaluated with a BioAnalyzer 2100 system (Agilent). Library sequencing was performed on an Illumina Hiseq instrument with 150 bp paired-end reads.

## Sequencing data analysis
For RIP-seq: both input and immunoprecipitated RNA samples were quality controlled and used to generate RNA-Seq libraries using a NEBNext® Ultra II Directional RNA Library Prep Kit (NEB). The quality of the final library was evaluated with both a QubitTM (Thermo) and a 2200 TapeStation System (Agilent). Sequencing was performed with an Illumina Hiseq platform in paired-end-read mode, with 150 bp per read. Adapters and low-quality reads were trimmed with Trimmomatic (v0.36)[99]. Sequencing reads were aligned to the mouse genome (mm10) with Tophat (v2.0.13)[100]. RIP targets were defined as peaks enriched in the IP (RIP/input ≥ 2 and $p < 0.0001$). RIP peaks were calculated with diffReps (v1.55.6)[101]. Peaks were annotated with the annotatePeaks.pl module of HOMER (v4.9.1), using the default setting. Motifs were found with the findMotifsGenome.pl module, using the "-size 200 -len6" option[102]. Metagene analysis to map mRNA peak distribution was performed using the Guitar Bioconductor package (v1.20.1)[103].

For MeRIP-seq: paired-end reads were harvested from an Illumina HiSeq 4000 sequencer, and quality control was performed via Q30 calculations. After 3′ adapter trimming and low-quality read elimination with Cutadapt software (v1.9.3)[104], all clean reads were mapped to the mouse reference genome (mm10) with HISAT2 software (v2.0.4)[105]. Methylated sites on peaks were identified with MACS2 peak-calling software (v2.1.1)[106], where the corresponding input sample served as a control. Differentially methylated sites were identified with diffReps (v1.55.6)[101]. Peaks identified by overlapping mRNA exons were determined and chosen with our own original scripts. Motifs enriched with m$^6$A peaks were identified with DREME (v5.4.1)[107]. m$^6$A peak distributions were visualized with the Integrative Genomics Viewer (IGV).

## Gene-specific MeRIP-qPCR

m⁶A modifications on specific genes were determined using Magna MeRIP™ m⁶A Kits (Millipore) according to the manufacturer's instructions. Briefly, 100 µg of total RNA was sheared to an approximate length of 100 nt using metal-ion-induced fragmentation. Before immunoprecipitation, one-tenth of fragmented RNA was saved as input control. The RNA fragments were then incubated with anti-m⁶A antibody-conjugated (202003, Synaptic Systems) or rabbit IgG-conjugated (Millipore) protein A/G magnetic beads in 1×immunoprecipitation buffer, supplemented with RNase inhibitors, at 4 °C for 2 h on a rotating wheel. After three washes, the bound RNA was eluted through competition with free N6-methyladenosine, and then purified using the RNeasy mini kit (QIAGEN). Input and immunoprecipitated m⁶A RNAs were reverse transcribed using a PrimeScript RT reagent kit (Takara), and the RNAs were further analyzed by qRT-PCR with the primers listed in Supplementary Table 3.

## In silico protein-protein docking analysis

Three-dimensional structure models of YTHDF3 (UniProt ID: Q8BYK6), eIF3a (UniProt ID: P23116), and eIF4B (UniProt ID: Q8BGD9) were constructed using the phyre2 web server[108]. The 3D structure model of YTHDF3 was docked onto the homologous model of either eIF3a or eIF4B using the ClusPro web server[57]. YTHDF3 was treated as the receptor, and eIF3a or eIF4B were used as ligands. ClusPro yielded four sets of docked structures using the following scoring schemes: (1) balanced, (2) electrostatic-favored, (3) hydrophobic-favored, and (4) van der Waals + electrostatics. We selected the "balanced" scoring scheme, and the highest ranked protein complex with the lowest energy score, as well as the largest number of members, was selected. Partner-specific protein-protein interface analysis was conducted at a structural level using the BIPSPI online tool[109]. All models were visualized using PyMOL software.

## Statistics and reproducibility

Each experiment was repeated three times independently, and data were presented as means ± standard error of the mean (SEM), unless otherwise specified. Statistical analyses and graphs were conducted with GraphPad Prism (version 9.0.0). Differences were considered statistically significant at $P$-value below 0.05.

## Reporting summary

Further information on research design is available in the Nature Research Reporting Summary linked to this article.

## Data availability

The RIP-seq and MeRIP-seq data generated in this study have been deposited in the GEO database under accession codes GSE158660 and GSE158268. The mass spectrometry proteomics data have been deposited to the ProteomeXchange Consortium (http://proteomecentral.proteomexchange.org) via the iProX partner repository with the dataset identifier PXD025450. All data needed to evaluate the conclusions in the paper are present in the paper and the Supplementary Information. Source data are provided with this paper. Additional data related to this paper may be requested from the authors.

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

## Acknowledgements

We thank Prof. Shuan Rao (Nanfang Hospital, Southern Medical University) and Chloe Yang (University of North Carolina at Chapel Hill) for critically reading our manuscript and for giving detailed comments and suggestions; Prof. Ying Yang and Prof. YunGui Yang (Beijing Institute of Genomics, Chinese Academy of Sciences) for technical assistance on MeRIP-seq; Dr. MengKe Chen, Dr. HongBo Li, and Dr. XiaoDan Ma (Sun Yat-sen University) for technical assistance on polysome profiling; Prof. Aibing Wu for offering material support; the staff members of the Central Laboratory of Southern Medical University for proteomics data collection and analysis; and the staff members of the Cancer Research Institute of Southern Medical University for support during data collection and discussion. This work was supported by grants from the National Natural Science Foundation of China (81872209, 82173299, 81672689, 81372896, 81172587 to D.X.; 81600086, 81770100 to Y.S.; 81600488, 81870602 to X.L.L.; 81702778 to J.S.J.); the Natural Science Foundation of Guangdong Province of China (2022A1515012477, 2014A030313294 to D.X.; 2022A1515012467 to J.S.J.; 2022A1515010018 to A.W.); the Science and Technology Planning Project of Guangdong Province of China (2017A010105017, 2013B060300013, 2009B060300008 to D.X.; 2017A030303018 to J.S.J.; 2015A030302024 to X.L.L.); the Medical Scientific Research Foundation of Guangdong Province of China (A2022107 to W.C.H.; B2020010 to J.H.W.; A2017420 to J.S.J.; B2014238

to H.F.S.); the Guangzhou Basic and Applied Basic Research Foundation (202102020697 to J.H.W.; 202201010139 to W.H.); the China Post-doctoral Science Foundation (2018T110884, 2017M622740, 2016T90792, 2015M572338 to X.L.L.).

## Author contributions

W.C.H. and M.J.D. designed the experiments. W.C.H., M.J.D., Y.Z., Q.L.Z., W.Q.P., Z.J.L., Y.Y.Z., J.C.M., R.R.L., and J.H.W. performed the experiments and data analyses. J.H.W. performed the bioformatics analyses. D.X., X.L.L., Y.L.L., S.H.H., G.Q.D., and W.C.H. prepared and carried out the animal experiments. W.C.H., M.J.D., and D.X. wrote the manuscript. D.X., Y.S., and J.H.W. conceived the ideas, organized and supervised the study, and provided advices and comments. D.X., Y.S., J.H.W., X.L.L., J.S.J., and H.F.S. provided material support.

## Competing interests

The authors declare no competing interests.
