## [Peer Review File · Nature Communications]

Autophagy induction promoted by m6A reader YTHDF3 through translation upregulation of FOXO3 mRNAREVIEWER COMMENTS

Reviewer #1 (Remarks to the Author):

This a quite thorough work showing important roles of YTHDF3 and its binding to mRNA m6A on autophagy. The authors have designed and executed a series of experiments to prove the effect of YTHDF3 on starvation-induced autophagy. This effect is m6A dependent. The model can be seen in Fig. 8. I have a few minor suggestions:

1. Does starvation induce FOXO3 m6A methylation? Did the authors test if other YTHDF proteins can bind FOXO3 mRNA and if so any translation or stability regulation upon KD of other YTHDF proteins.
2. How does YTHDF3 achieve transcript binding selective? Any potential RBP partners based on potential binding of other RBPs to the same site (analysis of CLIP-seq data of all known RBPs)?This may not be easy.
3. I would suggest adding discussion of other YTHDF3 targets. FOXO3 could be a main target accounting for a large portion of the observed phenotype but other targets could be important as well.
4. Title could change to "Autophagy induction promoted by a novel nutrient responder, YTHDF3, through translation upregulation of FOXO3 mRNA" or "Autophagy induction promoted by YTHDF3 through translation upregulation of FOXO3 mRNA"

Reviewer #2 (Remarks to the Author):

The manuscript submitted by Hao et al demonstrated that starvation in MEF cells activates METTL3-YTHDF3-FOXO3 signaling axis which facilitates the autophagy process. Overall, the findings are interesting, but need more experiments to validate their results.

- (1) The title should be reorganized; Its current form is exaggerated and does not contain scientific merit. The actual finding is just m6A reader YTHDF3 binding to methylated Foxo3, which mechanism has been established by many other published works. The title must be revised to a similar title: "m6A reader YTHDF3 binds to FOXO3 then regulates cellular autophagy".

(2) Fig. 1D&E, both mRNA m6A and expression levels of p62 under starvation combined with YTHDF3 depletion should be tested to exclude the possibility that this effect might be induced by an altered mRNA stability instead of changed protein degradation.

(3) Fig.3A&D findings are interesting. It is supposed that cellular functions under starvation are most inactivated, what is the biological significance of an enhanced mRNA m6A? The present study used HBSS for starvation medium, the authors are recommended to test the starvation-induced METTL3 activation using different serum-free medium, such as DMEM without glucose, and also different cell types. Additionally, autophagy induction post METTL3 depletion should be tested.

(4) It is known that nutrient deprivation induces a decreased ATP production and subsequently a quenched enzyme activity. METTL3 induction by starvation is regulated transcriptionally or post-transcriptionally? To prove METTL3's catalyzing activity is truly activated under starvation condition, a minigene (with m6A sites) report assay should be performed to validate their findings. If METTL3 enzyme activity is enhanced by starvation, how it possibly happens that only a sub-class of mRNAs showing an increased m6A modification. In addition, METTL3 expression level under a prolonged starvation should be tested.

(5) Fig.5A: FOXO3 expression regulated by YTHDF3 should be validated post METTL3 depletion. In 5B&C, protein level of autophagy genes (can choose several representative ones) should be included.

(6) YTHDF3 has been shown to promote translation of the transcription corepressor-FOXO3 through binding to its translation initiation region of mRNA, then cooperatively suppressing the expression of a list of downstream genes (Zhang Y, et al. PNAS, 2019). How YTHDF3 selectively binds to stop codon area of FOXO3 to promote its translation under starvation, and FOXO3 is supposed to exhibit suppressive effect on translation of its target genes, which is apparently opposite with the findings of this study.

(7) Minors:

Page3-119, typo errors: "This funding"; Fig.7F, RNase: wrong labeling

Reviewer #3 (Remarks to the Author):

Hao et al report that nutrient deficiency upregulates YTHDF3, an m6A reader, and METLL3, an enzyme that induces m6A modification in mRNAs. In response to starvation, mRNA of Foxo3 was subjected to m6A modification via METLL3 dependent manner. The m6A modification promotes its translation in YTHDF3 dependent manner, thereby inducing autophagy.

General:

METLL3-induced m6A modification of Foxo3 mRNA and autophagy has been reported (Lin et al EMBO J 2020). The fact that YTHDF3 upregulates translation of FoxO3 also has been reported (Zhang et al 2019). The interaction between YTHDF3 and eIF is also known. Thus, the overall story reported in this manuscript can be predicted easily.

Although the authors propose that nutrient starvation-induced upregulation of METLL3 and YTHDF3 regulates autophagy through upregulation of FoxO3, loss of function studies were conducted with nearly complete downregulation of METLL3 or YTHDF3, which would downregulate FoxO3 and autophagic machinery below physiological levels. Suppression of autophagy in such a condition does not necessarily mean that modest upregulation of these molecules that occurs in response to starvation actively mediates autophagy. In addition, parallel loss of function studies were not conducted to study the role of other YTHDFs or methylases to rule out their contributions to autophagy at baseline.

Other issues:

- 1) The effect of METLL3 knockdown was examined with YTHDF3 overexpression. However, whether YTHDF3 overexpression promotes autophagy is not shown. The authors should address this. In addition, whether METLL3 knockdown without YTHDF3 overexpression inhibits autophagy should be shown.
- 2) Whether METLL3 overexpression promotes autophagy, and if so, whether downregulation of YTHDF3 inhibits METLL3-induced autophagy should be tested.
- 3) Whether m6A defective Foxo3 mutant fails to bind to YTHDF3 should be shown.
- 4) Please describe the baseline phenotype of YTHDF3 knockout mice.
- 5) Please provide better photos of the EM analysis in Fig. 1J.
- 6) in Fig. 3IK, the labeling of control and starvation is missing.

7) In Figure 4, the authors should show that peaks for YTHDF3 and m6A hypermethylation occur at exactly the same place on FoxO3 mRNA.

8) The authors should show the level of FoxO3 protein, particularly in the nucleus in Figure 5C.

Reviewer #4 (Remarks to the Author):

I was asked specifically to evaluate the proteomics portion of this manuscript, so my comments below relate to that section of the paper.

The authors performed a label-free quantitative proteomics experiment to determine how nutrient deficiency in MEFs impacts autophagy. Unfortunately, the authors did not provide sufficient detail to evaluate the validity of these results.

The materials and methods section does not contain sufficient details; missing information is listed below.

There is no description of the LC-MS acquisition method, nor is there a reference to a previous report of said method.

The details of the database search are incomplete. (no parent or fragment ion mass error tolerances, missing number of proteins in the database)

The quantification and statistical analysis section is also incomplete, as is the description of quantification in Perseus. Were the P-values corrected (i.e. was multiple testing correction performed)? In what order were the samples analyzed? Were the replicates randomized for analysis?

Minor point: I think the authors meant to say "reversed" sequences rather than "revert" sequences.

Based on the missing details described above, some revision of this manuscript is required.

Point-by-point responses (NCOMMS-21-11358)

You will find our responses to each of the reviewer's points and suggestions. The reviewers' comments are in *italics*.

Reviewer #1 (Remarks to the Author):

This a quite thorough work showing important roles of YTHDF3 and its binding to mRNA m6A on autophagy. The authors have designed and executed a series of experiments to prove the effect of YTHDF3 on starvation-induced autophagy. This effect is m6A dependent. The model can be seen in Fig. 8. I have a few minor suggestions:

Our response: It is highly appreciated that the reviewer showed his support and positive comments to our manuscript.

1. Does starvation induce FOXO3 m6A methylation? Did the authors test if other YTHDF proteins can bind FOXO3 mRNA and if so any translation or stability regulation upon KD of other YTHDF proteins.

Our response: We thank the reviewer for proposing such extra analyses to investigate other YTHDF proteins. According to our MeRIP-seq data, FOXO3 mRNA was hyper-m⁶A-methylated upon nutrient starvation and the m⁶A hypermethylated peaks in FOXO3 mRNA were located in CDS and 3'UTR regions around the stop codon (Please refer to revised Fig.4f for more details). We further confirmed this finding by performing FOXO3-specific MeRIP-qPCR, and verified the m⁶A hypermethylation in FOXO3 transcripts induced by nutrient deficiency is METTL3-dependent (Fig.4g).

As suggested, the effects of other YTHDF proteins on FOXO3 expression were determined by Western blotting. YTHDF1 KD cells showed decreased expression of FOXO3 proteins compared to their control counterparts, while YTHDF2 KD slightly increased FOXO3 expression, especially in the nucleus following nutrient starvation (Fig.S13a, b). To further investigate their effects on FOXO3 expression, we performed polysome profiling to evaluate the translational efficiency of FOXO3 mRNAs when knocked down YTHDF1 and YTHDF2, respectively. Our data indicated a down-regulation of FOXO3 mRNA level in polysome fractions when knocking down YTHDF1 (Fig.S13c, d), while FOXO3 mRNA level was unchanged in YTHDF2 KD cells (Fig.S13e, f). RNA stability assays revealed that neither YTHDF1 nor YTHDF2 depletion could prolong the half-life of FOXO3 mRNAs (Fig.S13g, h). This finding is inconsistent with two previous findings which showed that FOXO3 mRNA stability is enhanced by YTHDF1 overexpression in liver cancer cells under hypoxia¹ or reduced by YTHDF2 overexpression in the luteinized GCs of PCOS patients². Interestingly, although both YTHDF1 and YTHDF2 bind to FOXO3 transcripts, nutrient deprivation has the opposite effect on their binding interaction. Our data showed that starvation could significantly strengthen the binding of YTHDF1, but decreased the binding of YTHDF2, to FOXO3 mRNAs (Fig. S13i, j). These results suggest that YTHDF1 and YTHDF3 might act synergistically to compete with YTHDF2 to bind FOXO3 mRNAs under nutrient deficiency. Such findings can also explain the observation that YTHDF2 had a mild effect on FOXO3 expression under normal

conditions, but upon nutrient starvation, YTHDF2 depletion could clearly upregulate FOXO3 expression (Fig.S13b).

Figure 1 for reviewers. (4f) Integrative Genomics Viewer (IGV) tracks displaying YTHDF3-RIP-seq (upper panels) and MeRIP-seq (lower panels) read distribution along the CDS and 3'UTR of FOXO3 mRNA. The squares mark increases in m⁶A peaks in MEFs upon nutrient deficiency. (4g) Gene-specific MeRIP-qPCR analysis of m⁶A level changes at the CDS and 3'UTR regions of FOXO3 mRNA transcripts in shNS and shMETTL3 MEFs upon nutrient deficiency. (S13a) Nuclear and cytoplasm fractions from shNS and shYTHDF1 MEFs following nutrient starvation for the indicated time periods were subjected to immunoblotting. (S13b) Nuclear and cytoplasm fractions from shNS and shYTHDF2 MEFs following nutrient starvation for the indicated time periods were subjected to immunoblotting. (S13c) Sucrose gradient-based polysome profiling of shNS and shYTHDF1 MEFs. (S13d) FOXO3 mRNAs in each ribosome fraction were quantified through qPCR and plotted as percentages of the total input from shNS and shYTHDF1 MEFs. (S13e) Sucrose gradient-based polysome profiling of shNS and shYTHDF2 MEFs. (S13f) FOXO3 mRNAs in each ribosome fraction were quantified through qPCR and plotted as percentages of the total input from shNS and shYTHDF2 MEFs. (S13g) shNS and shYTHDF1 MEFs were treated with Act D (5µg/mL) for the indicated times. The expression of FOXO3 was examined with qRT-PCR. Data from three independent experiments are expressed as means ± SEM. (S13h) shNS and shYTHDF2 MEFs were treated with Act D (5µg/mL) for the indicated times. The expression of FOXO3 was

examined with qRT-PCR. (S13i) YTHDF1-RIP followed by qRT-PCR detected the interaction between YTHDF1 and FOXO3 mRNA in MEFs before and after nutrient starvation. (S13j) YTHDF2-RIP followed by qRT-PCR detected the interaction between YTHDF2 and FOXO3 mRNA in MEFs before and after nutrient starvation. Two-tailed unpaired Student's t-test. Data from three independent experiments are expressed as means \pm SEM. *P < 0.05, ***P < 0.001.

2. How does YTHDF3 achieve transcript binding selective? Any potential RBP partners based on potential binding of other RBPs to the same site (analysis of CLIP-seq data of all known RBPs)? This may not be easy.

Our response: We thank the reviewer's suggestion to check potential RBPs involved in YTHDF3 binding specificity. By analyzing variations in YTHDF3 protein interactome upon nutrient starvation from our co-IP LC-MS/MS data. We observed that altered-binding proteins were highly enriched in protein catabolic processes, stress response regulation, autophagy, lysosome, and translational initiation using GO and KEGG pathway analyses (Fig.S14a, b). Next, we screened the putative YTHDF3 coregulators with known target sequences based on RBP databases (including RBPDB³, starBase⁴, and POSTAR3⁵), and obtained 81 up-enriched and 11 down-enriched proteins for further RBP binding sites analyses (Fig.S14c). Most of these RBPs contained conserved RNA-binding domains, such as the RNA-recognition motifs (RRMs), the K-Homology (KH) domains, and Zinc Fingers (ZFs) (Fig.S14c), suggesting they might play a role in the YTHDF3-RNA recognition process. To further examine how YTHDF3's transcript-binding selectivity was achieved during nutrient deprivation, we focused on those target sequences which were recognized by well-defined RBPs in the list of YTHDF3 interactors. We found that elements with 'GGAC' motifs were significantly enriched in the target sequences of multiple potential YTHDF3 RBP partners, including eIF3a, eIF4B, eIF4D, IGF2BP2, UPF1, and SRSF7 (Fig.S14d), indicating these RBPs might cooperate with YTHDF3 to selectively recognize its target sites. Moreover, according to published CLIP-Seq data^{4,5}, 33 of the YTHDF3 RBP interactors were found to bind FOXO3 transcripts, suggesting these proteins might cooperate with YTHDF3 to specifically recognize FOXO3 mRNAs (Fig.S15-17). We then systematically analyzed the FOXO3 transcript binding sites of each potential YTHDF3 RBP partner and overlapped them with the starvation-induced YTHDF3 binding and hyper-methylated sites. This strategy revealed a bunch of proteins, including RBFOX2, CELF1, SRSF7, DDX54, CNBP, FBL, FMR1, FXR2, ELAVL1, FUS, IGF2BP2, NOP58, UPF1, U2AF2, and RBFOX1, bound to the same regions in either the CDS or 3'UTR as those sites which were hyper-m⁶A-methylated and selectively recognized by YTHDF3 upon nutrient starvation (Fig.S15-17). Although we didn't provide any functional data showing those above-mentioned RBPs involved in contributing to YTHDF3 binding specificity, we do believe such association results could narrow down the key RBPs in regulating YTHDF3 binding with FOXO3 mRNA upon nutrient deficiency. As the reviewer pointed out "This may not be easy", we aim to identify these RBPs in the future study.

Figure 2 for reviewers. (S14a) GO biological process enrichment analysis of the 955 proteins differentially bound by YTHDF3 upon nutrient starvation. (S14b) KEGG pathway enrichment analysis of the 955 proteins differentially bound by YTHDF3 upon nutrient starvation. (S14c) Schematic representation of the domain architecture of putative YTHDF3 RBP partner proteins. The domains are named and located according to the CCD protein domain database. Specific domains are shown with boxes of the indicated colors. (S14d) The consensus sequence motif

identified within the indicated YTHDF3 partners' target sequences, according to the starBase and RBPDB databases.

Figure 3 for reviewers. (S15a) IGV tracks displaying the reads of YTHDF3 RIP-seq (upper panels) and MeRIP-seq (lower panels) along the FOXO3 mRNAs. The red squares mark significant hyper-m⁶A peaks from MeRIP-seq of two replicates, the green square marks significant up-enriched peaks of YTHDF3 from RIP-seq of two replicates, the orange square marks the m⁶A peaks predicted by bioinformatics, identified hyper-m⁶A-methylated in MeRIP-seq of one replicate, and functionally verified by EMSA and dual-luciferase reporter assay. **(S15b)** IGV tracks displaying the indicated RBPs' binding sites on FOXO3 mRNAs (pink line) from CLIPdb database mouse data. The light-blue ranges mark the corresponding regions to the hyper-m⁶A-methylated and YTHDF3 up-enriched peaks in response to nutrient starvation on mouse FOXO3 transcript.

S16**S17**
Figure 4 for reviewers. (S16 and S17) IGV tracks displaying the indicated RBPs' binding sites on FOXO3 mRNAs (pink line) from the starBase human data (S16) or CLIPdb database human data (S17). The light-blue ranges mark the corresponding regions to the mouse hyper-m⁶A-methylated and YTHDF3 up-enriched peaks in response to nutrient starvation on mouse FOXO3 transcript.

3. I would suggest adding discussion of other YTHDF3 targets. FOXO3 could be a main target accounting for a large portion of the observed phenotype but other targets could be important as well.

Our response: We thank the reviewer for this kind suggestion to add an extra discussion regarding YTHDF3 other target genes. We have added such a discussion in the revised manuscript. Please also refer to the "Discussion" section of the revised manuscript for more details.

"Besides FOXO3, there were some other genes also identified within the list of YTHDF3-bound hypermethylated genes up-enriched in response to nutrient scarcity. Similar to FOXO3, some of these genes were reported to play a role in regulating autophagy under basal or starved conditions. For instance, CDKN1B was shown to promote autophagy upon nutrient shortage by repressing mTORC1 or directly facilitating lysosomal function^{6,7}; SESN2 contributed to autophagy through AMPK-mTORC1 signaling⁸; Mitochondrial PLD6 promoted BNIP3L-induced MTOR-RPS6KB activation and triggered mitophagy⁹. However, since we demonstrated that YTHDF3 had little effect on AMPK or mTORC1 signaling pathways, we speculated that these aforementioned genes might not be the main functional targets of YTHDF3 regulating autophagy. Furthermore, several genes were reported to involve in autophagic multi-step or stage-specific regulation. For example, CCR4 deadenylase was reported to play important roles in regulating RNA stability of diverse key autophagy-related genes¹⁰. FNIP2 enhanced FLCN-GABARAP interactions and drove autophagy flux at different stages¹¹. BMF interacted with Becn1-Bcl2 complexes, or more autophagy-activating pathways, to regulate autophagy¹². The pro-oxidant protein TXNIP suppressed ATG4B activity and promoted autophagosome maturation through ROS regulation¹³. ZFYVE26 mutations led to defects in autophagosome maturation and autophagic lysosome reformation¹⁴. Nevertheless, whether and how these above-mentioned genes were the important targets of YTHDF3 in regulating autophagy require further investigation."

4. Title could change to "Autophagy induction promoted by a novel nutrient responder, YTHDF3, through translation upregulation of FOXO3 mRNA" or "Autophagy induction promoted by YTHDF3 through translation upregulation of FOXO3 mRNA"

Our response: We thank the reviewer for this kind suggestion. We would propose "Autophagy induction promoted by m⁶A reader YTHDF3 through translation upregulation of FOXO3 mRNA" as the revised title, which might better represent the contents of our study.

Reviewer #2 (Remarks to the Author):

The manuscript submitted by Hao et al demonstrated that starvation in MEF cells activates METTL3-YTHDF3-FOXO3 signaling axis which facilitates the autophagy process. Overall, the findings are interesting, but need more experiments to validate their results.

(1) The title should be reorganized; Its current form is exaggerated and does not contain scientific merit. The actual finding is just m6A reader YTHDF3 binding to methylated Foxo3, which mechanism has been established by many other published works. The title must be revised to a similar title: “m6A reader YTHDF3 binds to FOXO3 then regulates cellular autophagy”.

Our response: We thank the reviewer for this comment which was also raised by reviewer 1. We would propose “Autophagy induction promoted by m6A reader YTHDF3 through translation upregulation of FOXO3 mRNA” as the revised title, which might better represent the contents of our study.

(2) Fig. 1D&E, both mRNA m6A and expression levels of p62 under starvation combined with YTHDF3 depletion should be tested to exclude the possibility that this effect might be induced by an altered mRNA stability instead of changed protein degradation.

Our response: We thank the reviewer for pointing out these important control experiments. As suggested, we investigated whether YTHDF3 depletion affected the mRNA levels, mRNA stability as well as protein translation efficiency of p62. Our data showed that YTHDF3 depletion didn't alter mRNA levels or mRNA stability of p62 under starvation (Fig.S1a, b). Meanwhile, there were no significant changes in p62 translation efficiency detected in YTHDF3 KD cells compared to their control counterparts according to polysome profiling assay (Fig.S1c). These results exclude the possibility that the attenuated decrease of p62 protein levels under starvation in YTHDF3 silencing cells is due to an altered mRNA stability or protein synthesis.

Figure 5 for reviewers. (S1a) qRT-PCR analyses of p62 in shNS and shYTHDF3 MEFs following nutrient starvation for the indicated time periods. (S1b) shNS and shYTHDF3 MEFs were treated with Act D (5µg/mL) for the indicated times, with or without nutrient starvation, respectively. The expression of p62 was examined with qRT-PCR. (S1c) Relative levels of p62 mRNAs in each ribosome fraction were quantified and plotted as a percentage relative to the total input.

(3) Fig.3A&D findings are interesting. It is supposed that cellular functions under starvation are

most inactivated, what is the biological significance of an enhanced mRNA m⁶A? The present study used HBSS for starvation medium, the authors are recommended to test the starvation-induced METTL3 activation using different serum-free medium, such as DMEM without glucose, and also different cell types. Additionally, autophagy induction post METTL3 depletion should be tested.

Our response: We thank the reviewer's positive comment and would like to address these concerns respectively. First of all, nutrient deprivation usually leads to curtailed anabolism associated with growth, greater catabolism, and use of essential nutrients to support survival. Autophagy is a catabolic process that degrades damaged organelles and misfolded proteins, repurposing them to maximize cellular viability, which is activated upon nutrient deprivation. Mechanistically, the nutrient-sensing signaling network changes within a short time under starvation, and overall cellular protein synthesis is attenuated, whereas a set of stress- and autophagy-related mRNAs undergo continued or enhanced translation^{15,16}. In the current study, we observed a global increase in m⁶A methylated mRNAs as well as cytoplasmic m⁶A accumulation due to nutrient deficiency, implying m⁶A methylation might play an important role in cellular survival under such hostile conditions. Consistently, a recent study reported that dynamic 5'UTR m⁶A modification in response to nutrient deprivation regulates ATF4 alternative translation by influencing TIS selection¹⁷. Unlike ATF4, we found that FOXO3, the master transcriptional regulator of autophagy genes, exhibits increased m⁶A methylation in the CDS and 3'UTR regions around the stop codon in response to nutrient starvation, and these m⁶A modification events are quite important for FOXO3 translation. This finding expands the current understanding that dynamic m⁶A changes in response to cellular stress mainly occurs in 5'UTRs¹⁸. In addition, we propose that METTL3 is a stress-responsive gene to nutrient scarcity, since the up-regulation of METTL3 upon starvation is due to the decreased ubiquitination of METTL3, thus promoting its stabilization. However, some other mechanisms can't be excluded that modulate dynamic METTL3 enzymatic activity and m⁶A during nutrient deprivation. For instance, nutrient deprivation could reduce intracellular levels of SAH and SAM, disrupting their balance and effects on METTL3, and affecting METTL3's methyltransferase activity¹⁹. Furthermore, altered TCA cycle metabolites, such as α KG, could change the enzymatic activities of m⁶A erasers FTO and ALKBH5, leading to changes in cellular m⁶A modifications as well^{20,21}. Additionally, nutrient starvation could inhibit NADPH production, decrease FTO activity, and upregulate m⁶A levels²². Therefore, activated m⁶A methylation is an important and general biological process with a complex regulating network to overcome nutrient deprivation and sustain cell viability.

As the reviewer suggested, we validated the dynamic increment of METTL3 in response to nutrient deficiency by glucose starvation, albeit variations were observed in METTL3 induction among different cell types (Fig.S4c). Interestingly, METTL3 induction was not observed during short-term starvation in the hepatocellular carcinoma cell line HepG2, but its expression increased after a longer starvation induction (Fig.S4d). These data suggest that METTL3 induction might be a general protection mechanism against diverse types of nutrient scarcity.

Next, as the reviewer requested, we explored how METTL3 modulated autophagy without manipulating YTHDF3 expression. As expected, METTL3 depletion markedly dampened LC3-II accumulation and p62 degradation during nutrient deficiency (Fig.S7a), suggesting autophagy flux is impaired. Interestingly, METTL3 overexpression had little effect on LC3-II and p62 levels

(Fig.S7b), indicating that the METTL3 was redundant in response to nutrient starvation to induce autophagy. To examine whether METTL3 regulates autophagy in an m⁶A-dependent manner, we conducted rescue experiments in METTL3 depleted cells by re-expression of either wild-type or catalytic-mutant METTL3. Similar to those YTHDF3-overexpressing cells, we observed that re-introducing wild-type METTL3, but not the catalytic-mutant, restored nutrient starvation-induced LC3-II accumulation and p62 degradation (Fig.S7c), indicating that METTL3's regulated autophagy flux was m⁶A-dependent. These observations were further confirmed by a tandem mCherry-GFP-LC3 reporter assay. Nutrient starvation-induced autophagosomes and autolysosomes were significantly impeded in METTL3-deficient cells (Fig.S7d, e), and this autophagic defect was reversed by re-introducing wild-type METTL3, but not the catalytic-mutant form (Fig.S7f, g). In addition, no significant effect of METTL3 overexpression was observed on the phenotypes of starvation-induced autophagosome and autolysosome formations (Fig.S7h, i). Taken together, these results suggest that METTL3-mediated m⁶A modification is essential for nutrient starvation-induced autophagy.

Figure 6 for reviewers. (S4c, d) Immunoblot analyses of METTL3 in the indicated cell types in response to either glucose starvation (S4c) or amino acid starvation (S4d), analyzed after the indicated time periods. GAPDH is shown as a loading control. **(S7a)** Immunoblot analyses of LC3-II and p62 in MEFs infected with nonspecific shRNA (shNS) or shMETTL3 following nutrient starvation for the indicated time periods, with or without Baf.A1 treatment (20nM). GAPDH was used as a loading control. **(S7b)** Immunoblot analyses of LC3-II and p62 in MEFs ectopically expressing either METTL3 or its control vector (Con). GAPDH is shown as a loading control. **(S7c)** Immunoblot analyses of METTL3-silenced MEFs transfected with lentiviral vectors (Con), wild type METTL3 (METTL3-WT), or a catalytic mutant of METTL3 (METTL3-Mut) following nutrient starvation for the indicated time periods, with or without Baf.A1 treatment (20nM). GAPDH is used as a loading control. **(S7d, e)** Measurement of autophagy flux and quantification of autophagosomes (yellow) and autolysosomes (red) by a tandem mCherry-GFP-LC3 reporter assay in shNS and shMETTL3 MEFs. Scale bar, 20 μ m. **(S7f, g)** Measurement of autophagy flux and quantification of autophagosomes (yellow) and autolysosomes (red) by a tandem mCherry-GFP-LC3 reporter assay in METTL3-silenced MEFs transfected with METTL3-WT, METTL3-Mut, or Con, with or without nutrient deficiency. Scale bar, 20 μ m. **(S7h, i)** Measurement of autophagy flux and quantification of autophagosomes (yellow) and autolysosomes (red) by a tandem mCherry-GFP-LC3 reporter assay in MEFs ectopically expressing either METTL3 or its control vector (Con) with or without nutrient deficiency. Scale bar, 20 μ m. Two-tailed unpaired Student's t-test. The data are represented as the means \pm SEM. **P < 0.01, ***P < 0.001, ns indicates no significance.

(4) It is known that nutrient deprivation induces a decreased ATP production and subsequently a quenched enzyme activity. METTL3 induction by starvation is regulated transcriptionally or post-transcriptionally? To prove METTL3's catalyzing activity is truly activated under starvation condition, a minigene (with m6A sites) report assay should be performed to validate their findings. If METTL3 enzyme activity is enhanced by starvation, how it possibly happens that only a sub-class of mRNAs showing an increased m6A modification. In addition, METTL3 expression level under a prolonged starvation should be tested.

Our response: We thank the reviewer for these great inputs. To examine whether METTL3 was transcriptionally regulated, we first analyzed changes in METTL3 mRNA abundance during nutrient deprivation. Our results showed only a slight increase in response to nutrient starvation (Fig.3n). We next examined whether nutrient starvation could affect METTL3 mRNA stability, but no significant difference was observed (Fig.3o). Interestingly, METTL3 protein up-regulation upon nutrient starvation was almost completely eliminated by treatment of MG132, a classical proteasome inhibitor (Fig.3p), indicating the protein stability regulation of METTL3 is the main reason for its significant induction under nutrient deficiency. This notion was corroborated by a chase assay using a protein synthesis inhibitor, cycloheximide (CHX), which revealed that nutrient starvation attenuated METTL3 protein degradation (Fig.3q). In addition, we also noticed that nutrient starvation markedly reduced the ubiquitination of METTL3 (Fig.3r). To summarize, we propose that the inhibited ubiquitination of METTL3 following nutrient starvation is accounted for its increased stability and expression.

As suggested, to explore METTL3's catalyzing activity was truly activated under starvation, we

directly measured the m⁶A catalytic activity of METTL3 within cell extracts from both control and starved cells. We employed the S-(5'-Adenosyl)-L-methionine-d₃ (d₃-SAM) to quantify METTL3 methyltransferase activity with RNA probes containing the consensus motif 'GGACU', as previously described^{23,24}. Our data showed that the cellular METTL3 obtained from starved cells achieved a higher molar ratio of d₃-m⁶A to RNA probe than that of control cells (Fig.3f), suggesting nutrient starvation enhances N⁶-adenosine methylation efficiency. One possible mechanism to explain this phenotype is the different effects of decreased S-adenosyl methionine (SAM) and S-adenosylhomocysteine (SAH) levels on METTL3 methyltransferase activity. Briefly, SAM, generated from ATP and methionine, provides methyl groups and constitutively activates METTL3's catalytic activity in the m⁶A methylation process. In contrast, SAH, generated from the methylation reactions, in turn inhibits METTL3's methyltransferase activity²⁵. Since cellular SAM levels are much higher than their *K_m* (substrate concentration at half maximum reaction rate) for METTL3¹⁹, the reduction of ATP or SAM might have no significant effect on METTL3's activity at the early stage of nutrient starvation. Meanwhile, the level of cellular SAH is relatively close to its IC₅₀ (half maximal inhibitory concentration) on METTL3, thus the reduction of SAH under nutrient starvation would alter METTL3's activity much earlier than the reduction of SAM. As starvation time prolongs, a significant decrease in SAM levels towards its *K_m* might result in a significant decrease in its activating effect on METTL3's activity. In line with this, we observed the molar ratio of d₃-m⁶A to RNA probe decreased 6 hours post starvation (Fig.3f), indicating that the catalytic activity of METTL3 was diminished. Interestingly, METTL3 protein levels did not change during prolonged starvation (Fig.3g).

As the reviewer pointed out, we observed only a sub-class of mRNAs showing an increased m⁶A modification after starvation from MeRIP-seq data (Fig.4d). Nevertheless, the global m⁶A enrichment analysis confirmed a significant up-regulation of m⁶A modification levels in mRNAs after starvation (Fig.S9b). To explore why only a sub-class of mRNAs showed an increased m⁶A modification, we first compared the identified hyper-m⁶A-methylated genes with METTL3-bound transcripts from starBase human CLIP-seq data⁴. Irrespective of species and cell types difference, 36% of the hyper-m⁶A-methylated genes were also found in the METTL3-bound transcripts (Fig.S9c). Since parts of the m⁶A modifications deposited by METTL3 were also reversed by m⁶A demethylases (FTO or ALKBH5), we next compared the METTL3-bound transcripts without hyper-methylation to the FTO- or ALKBH5-bound transcripts using the starBase human CLIP-seq data⁴, and found that 65% of genes without hyper-m⁶A-methylation following starvation were simultaneously bound by FTO or ALKBH5 (Fig.S9c). Since the m⁶A modification levels are determined by the balance between methylation and demethylation, we therefore propose that only a sub-class of mRNAs were highly m⁶A-methylated during the starvation process because of up-regulation of demethylation occurred with the other mRNAs. In addition, those transcripts without m⁶A modification sites should not change their m⁶A methylation during starvation.

Finally, as we mentioned above (second paragraph), our data showed that METTL3 protein levels did not change as long as 16 hours long starvation (Fig.3g).

Figure 7 for reviewers. (3f) The relative m⁶A methylation catalytic activities of purified METTL3 from the MEFs starved for the indicated time periods were determined using an RNA probe and d₃-SAM. The methylation yields were calculated based on the molar ratio of newly formed d₃-m⁶A to digested RNA probes. G was used as an internal control to calculate the amount of RNA probes. *P < 0.05 and ***P < 0.001. (3g) Immunoblot analyses of METTL3 in MEFs following nutrient starvation for the indicated time periods. (3n) qRT-PCR analyses of METTL3 in MEFs following nutrient starvation for the indicated time periods. (3o) MEFs were treated with Act D (5μg/mL) for the indicated times with or without nutrient starvation, respectively. The expression of METTL3 was examined with qRT-PCR. (3p) MEFs with or without nutrient deficiency were treated with 20μM MG132 for the indicated time periods. Levels of METTL3 were examined by immunoblot analyses. GAPDH was used as a loading control. (3q) MEFs with or without nutrient deficiency were treated with 100μg/ml cycloheximide (CHX) for the indicated time periods. Levels of METTL3 were examined by immunoblot analyses. GAPDH was used as a loading control. (3r) METTL3 overexpressing MEFs were treated with or without nutrient starvation after incubation with 20μM MG132. Ubiquitinated METTL3 were pulled down with an anti-METTL3 antibody, and then subjected to Western blotting using an anti-ubiquitin antibody. (S9b) Violin plots of log₂ fold enrichment of all the m⁶A peaks in MEFs with or without nutrient starvation. P=3.2963E-19 by Wilcoxon signed-rank tests. (S9c) Venn diagrams show overlap between hyper-m⁶A genes from our mouse MeRIP-seq data and METTL3-bound transcripts from starBase human CLIP-seq data; and overlap between the former resultant METTL3-bound transcripts without significant hyper-m⁶A-methylated during nutrient starvation, and FTO- or ALKBH5-bound transcripts from starBase human CLIP-seq data.

(5) Fig. 5A: FOXO3 expression regulated by YTHDF3 should be validated post METTL3 depletion. In 5B&C, protein level of autophagy genes (can choose several representative ones) should be included.

Our response: We thank the reviewer for these kind suggestions. As requested, we first explored the expression of FOXO3 after METTL3 depletion. Knocking down METTL3 obviously abrogated FOXO3 expression levels in the nucleus and cytoplasm of YTHDF3-overexpressing MEFs under both normal and starved conditions. Notably, nuclear FOXO3 in METTL3-silenced cells was much significantly lower than that in control cells because of its translocation to the nucleus in response to starvation (Fig.5d). In addition, the protein levels of FOXO3 target genes involved in autophagy were also attenuated by METTL3 depletion (Fig.5e). These data indicate that YTHDF3 regulates FOXO3 expression and alters expression of FOXO3-targeted autophagic genes in a METTL3-dependent manner.

Additionally, as requested by the reviewer, protein levels of autophagic genes were further confirmed with Western blotting assay (revised Fig.5c, i).

Figure 8 for reviewers. (5c) Immunoblot analyses of FOXO3-targeted autophagic genes in shNS, shYTHDF3, control, and YTHDF3-OE MEFs. GAPDH is shown as a loading control. (5d) Immunoblot analyses of nuclear and cytoplasmic FOXO3 expressions in METTL3-silenced YTHDF3-OE (YTHDF3+shMETTL3) and control (YTHDF3+shNS) MEFs following nutrient starvation for the indicated time periods. (5e) Immunoblot analyses of FOXO3-targeted autophagic genes in shNS, shMETTL3, control, and METTL3-OE MEFs. (5i) Immunoblot analyses of FOXO3-targeted autophagic genes in FOXO3 rescued (shYTHDF3+FOXO3) and control (shYTHDF3+Con) MEFs. GAPDH is used as a loading control.

(6) YTHDF3 has been shown to promote translation of the transcription corepressor-FOXO3 through binding to its translation initiation region of mRNA, then cooperatively suppressing the expression of a list of downstream genes (Zhang Y, et al. PNAS, 2019). How YTHDF3 selectively binds to stop codon area of FOXO3 to promote its translation under starvation, and FOXO3 is supposed to exhibit suppressive effect on translation of its target genes, which is apparently opposite

with the findings of this study.

Our response: It is highly appreciated that the reviewer pointed out such inconsistency between our study and Zhang's research, which might expand our understanding of FOXO3 in regulating gene expression. As Zhang et al. reported, in antiviral innate immunity, YTHDF3 interacted with PABP1 and eIF4G2, and promoted FOXO3 translation by binding its mRNA translation initiation region under homeostasis²⁶. Our co-IP assays also confirmed the interaction between YTHDF3 and PABP1 or eIF4G2 in MEFs. However, these interactions were significantly reduced following nutrient starvation (Fig.7h), whereas the interaction between YTHDF3 and eIF3a or eIF4B was increased (Fig.7e), suggesting the interactions between YTHDF3 and translation initiation regulators might be quite different in diverse cell subtypes and under various cellular stress conditions.

Regarding the specificity of YTHDF3 to the stop codon area of FOXO3, we performed electrophoretic mobility shift assays (EMSA) using recombinant YTHDF3 proteins and biotinylated RNA probes containing different sequences of FOXO3's CDS or 3'UTR with or without m⁶A modifications. Our results showed that once m⁶A modifications were removed from the RNA probes, no matter the m⁶A location in either the CDS or 3'UTR of FOXO3, the interactions between YTHDF3 and the probes were significantly attenuated (Fig.4i, j). Furthermore, since the observed band intensity of probe 1 or 3 combined with YTHDF3 were significantly higher than those of other probes with YTHDF3, we reasoned the m⁶A sites at 2158nt, 2151nt, 2163nt (probe 1), and 2295nt (probe 3) along the FOXO3-CDS might be more critical for YTHDF3 to recognize FOXO3 stop codon area than other m⁶A sites (Fig.4i, j).

Additionally, although some examples of FOXO3-mediated suppression of target gene expression have been reported²⁷, the general role of FOXO3 in regulating autophagy in most cell types is to poise cells for rapid autophagy induction following starvation and transactivating autophagy-related genes^{28,29}, which is consistent with our findings.

Figure 9 for reviewers. (7e) Immunoblot analyses of YTHDF3-immunoprecipitated proteins from shNS and shYTHDF3 MEFs, with or without nutrient deprivation. Total protein amounts of YTHDF3, eIF3a, and eIF4B were used as inputs. (7h) Immunoblot analyses for YTHDF3 in PABP1- or eIF4G2-immunoprecipitated lysates from MEFs with or without nutrient deprivation. Total protein amounts of PABP1, eIF4G2, and YTHDF3 were used as inputs. (4i, j) RNA EMSA assays were performed using recombinant YTHDF3 proteins and biotinylated RNA probes containing different sequences of FOXO3's CDS (i) or 3'UTR (j) with or without m⁶A modifications.

(7) *Minors:*

Page3-119, typo errors: "This funding"; Fig.7F, RNase: wrong label

Our response: We apologize for the errors. We have corrected the word and the label.

Reviewer #3 (Remarks to the Author):

Hao et al report that nutrient deficiency upregulates YTHDF3, an m⁶A reader, and METTL3, an enzyme that induces m⁶A modification in mRNAs. In response to starvation, mRNA of Foxo3 was subjected to m⁶A modification via METTL3 dependent manner. The m⁶A modification promotes its translation in YTHDF3 dependent manner, thereby inducing autophagy.

General:

METTL3-induced m⁶A modification of Foxo3 mRNA and autophagy has been reported (Lin et al EMBO J 2020). The fact that YTHDF3 upregulates translation of FoxO3 also has been reported (Zhang et al 2019). The interaction between YTHDF3 and eIF is also known. Thus, the overall story reported in this manuscript can be predicted easily.

Our response: We thank the reviewer for raising such concerns to the novelty of our work and acknowledge that the connection of m⁶A modification, FOXO3 as well as autophagy have been investigated in some previous studies. However, we are confident that the detailed mechanism reported in our manuscript is very different from those two studies mentioned by the reviewer, thus, our study will definitely expand our understanding of the role of YTHDF3/METTL3/FOXO3 in regulating autophagy, particularly under starved conditions. In detail, Lin et al. reported that METTL3 increased the stability of FOXO3 mRNA through a YTHDF1-dependent mechanism in hepatocellular carcinoma (HCC) cells, which in turn inhibited autophagy²⁷. Actually, their data showed that YTHDF1 knockdown had no effect on the expression of FOXO3 protein in blank Bel-7402 cells, although it did compromise the upregulation of FOXO3 protein in METTL3-overexpressed cells. Their polysome profiling data also showed that METTL3 knockdown only decreases the level of FOXO3 mRNA in 40S fraction, suggesting the FOXO3 translation may not be remarkably affected (Lin et al. EMBO J (2020))²⁷. Furthermore, in antiviral innate responses, Zhang et al. reported that YTHDF3 promoted FOXO3 mRNA translation in an m⁶A-independent manner by binding to the translation initiation region of FOXO3 mRNA via PABP1-eIF4G2 interactions (Zhang et al. PNAS (2019))²⁶. Surprisingly, our study demonstrated that YTHDF3 promoted FOXO3 translation via recognizing the stop codon area of FOXO3 in an m⁶A dependent fashion, which was mediated by METTL3 in starvation conditions. Therefore, the main findings in our manuscript will be a great complement to understanding the m⁶A modification/translation regulation/autophagy in response to different cellular stress, e.g., nutrient deprivation.

Although the authors propose that nutrient starvation-induced upregulation of METTL3 and YTHDF3 regulates autophagy through upregulation of FoxO3, loss of function studies were conducted with nearly complete downregulation of METTL3 or YTHDF3, which would downregulate FoxO3 and autophagic machinery below physiological levels. Suppression of autophagy in such a condition does not necessarily mean that modest upregulation of these molecules that occurs in response to starvation actively mediates autophagy. In addition, parallel loss of function studies were not conducted to study the role of other YTHDFs or methylases to rule out their contributions to autophagy at baseline.

Our response: We thank the reviewer for this kind suggestion and we do accept the argument that “Suppression of autophagy in such a condition does not necessarily mean that modest upregulation of these molecules that occurs in response to starvation actively mediates autophagy”. Actually, suppression of YTHDF3 indeed inhibited autophagy (Fig.1d, f, g), which could be restored by re-introducing YTHDF3 expression (Fig.1e, h, i); meanwhile, over-expression of YTHDF3 in both physiological and starved conditions could trigger autophagy (revised Fig.1j, k, l). Paradoxically, knocking down METTL3 inhibited autophagy which could also be rescued by re-expressing wild-type METTL3 (revised Fig. S7a, c-g), but over-expression of METTL3 had no significant effect on autophagy in either physiological or starved conditions (revised Fig. S7b, h, i). Therefore, in the current study, we proved that only up-regulation of YTHDF3 but not METTL3 could trigger autophagy, indicating that YTHDF3-mediated autophagy activation is a gene-specific event upon starvation. Moreover, manipulating YTHDF3 (knocking down & overexpressing) to control autophagy should provide enough evidence that modest upregulation of YTHDF3 that occurs in response to starvation actively mediates autophagy.

As for the other YTHDFs or methylases, we examined the effects of depleting YTHDF1 or YTHDF2 on autophagy, respectively. Silencing of YTHDF1 dampened the LC3-II elevation and p62 degradation during nutrient starvation, but this effect was observed to be much less significant than silencing YTHDF3 (Fig.S3b). On the other hand, YTHDF2 silencing had opposite effects on autophagy as compared to inhibition of YTHDF1 and YTHDF3 (Fig.S3c). Furthermore, decreased LC3-II levels and p62 accumulation was observed in METTL14 silencing cells compared to control cells (Fig.S8b). However, since the physical co-operation of METTL14 is necessary for METTL3’s methylases catalytic function³⁰, we cannot exclude the possibility that such impact on autophagy is a direct METTL14-dependent effect or via METTL3 methyltransferase activity.

Figure 10 for reviewers. (S3b, c) Immunoblot analyses of LC3-II and p62 in shNS and shYTHDF1 MEFs (S3b), and in shNS and shYTHDF2 MEFs (S3c), following nutrient starvation for the indicated time periods, with and without Baf.A1 treatment (20nM). GAPDH is used as a loading control. **(S8b)** Immunoblot analyses of LC3-II and p62 in shNS and shMETTL14 MEFs following nutrient starvation for the indicated time periods, with and without Baf.A1 treatment (20nM). GAPDH is used as a loading control.

Other issues:

1) *The effect of METLL3 knockdown was examined with YTHDF3 overexpression. However, whether YTHDF3 overexpression promotes autophagy is not shown. The authors should address*

this. In addition, whether METTL3 knockdown without YTHDF3 overexpression inhibits autophagy should be shown.

Our response: We thank the reviewer for proposing these detailed control experiments. As the reviewer suggested, we found that YTHDF3 overexpression markedly potentiated LC3-II expression and decreased p62 levels, indicating that YTHDF3 was not only necessary for maintaining physiological autophagy but also mediated autophagy enhancement (Fig.1j). This finding was also confirmed by a GFP-LC3 assay, which showed an increased number of cytosolic GFP-LC3 puncta through ectopic expression of YTHDF3 (Fig.1k, l).

Next, METTL3 depletion markedly dampened LC3-II accumulation and p62 degradation during nutrient deficiency (Fig.S7a), suggesting autophagy flux was impaired when suppressing METTL3. To corroborate the idea that METTL3 mediated autophagy in an m⁶A-dependent manner, we conducted rescue experiments in METTL3 depleted cells using either wild-type or catalytic-mutant METTL3. Similar to those YTHDF3-overexpressing cells, we observed that re-introducing wild-type METTL3, but not the catalytic-mutant, restored nutrient starvation-induced LC3-II accumulation and p62 degradation (Fig.S7c), indicating that METTL3's catalytic activity was dispensable for autophagy flux. These observations were confirmed by a tandem mCherry-GFP-LC3 reporter assay. Nutrient starvation-induced generation of autophagosomes and autolysosomes was significantly impeded in METTL3-deficient cells (Fig.S7d, e), and this autophagic defect was reversed by re-introducing wild-type METTL3, but not the catalytic-mutant form (Fig.S7f, g).

Figure 11 for reviewers. (1j) Immunoblot analyses of LC3-II and p62 in control and YTHDF3 overexpressing MEFs, following nutrient starvation for the indicated time periods, with and without Baf.A1 treatment (20nM). GAPDH is used as a loading control. (1k, l) Representative confocal images of GFP-LC3 puncta formation (1k) and quantification of GFP-LC3 puncta per cell (1l) in control and YTHDF3 overexpressing MEFs. Nuclei were counterstained with DAPI. Scale bar, 20 μ m. (S7a) Immunoblot analyses of LC3-II and p62 in MEFs infected with nonspecific shRNA (shNS) or shMETTL3 following nutrient starvation for the indicated time periods, with or without Baf.A1 treatment (20nM). GAPDH was used as a loading control. (S7c) Immunoblot analyses of METTL3-silenced MEFs transfected with lentiviral vectors (Con), wild type METTL3 (METTL3-WT), or a catalytic mutant of METTL3 (METTL3-Mut) following nutrient starvation for the indicated time periods, with or without Baf.A1 treatment (20nM). GAPDH is used as a loading control. (S7d, e) Measurement of autophagy flux and quantification of autophagosomes (yellow) and autolysosomes (red) by a tandem mCherry-GFP-LC3 reporter assay in shNS and shMETTL3 MEFs. Scale bar, 20 μ m. (S7f, g) Measurement of autophagy flux and quantification of autophagosomes (yellow) and autolysosomes (red) by a tandem mCherry-GFP-LC3 reporter assay in METTL3-silenced MEFs transfected with METTL3-WT, METTL3-Mut, or Con, with or without nutrient deficiency. Scale bar, 20 μ m. Two-tailed unpaired Student's t-test. The data are represented as the means \pm SEM. *P < 0.05, **P < 0.01 and ***P < 0.001.

2) Whether METTL3 overexpression promotes autophagy, and if so, whether downregulation of YTHDF3 inhibits METTL3-induced autophagy should be tested.

Our response: We thank the reviewer for this suggestion. We have confirmed that METTL3 overexpression had little effect on autophagy in the revised manuscript (Fig.S7b). In addition, by tandem mCherry-GFP-LC3 reporter assay, no significant effect of METTL3 overexpression was observed on the phenotypes of starvation-induced autophagosome and autolysosome formation (Fig.S7h, i). These data indicate that the METTL3 is redundant to mediate nutrient starvation-induced autophagy.

Figure 12 for reviewers. (S7b) Immunoblot analyses of LC3-II and p62 in MEFs ectopically expressing either METTL3 or its control vector (Con). GAPDH is shown as a loading control. (S7h, i) Measurement of autophagy flux and quantification of autophagosomes (yellow) and autolysosomes (red) by a tandem mCherry-GFP-LC3 reporter assay in MEFs ectopically expressing either METTL3 or its control vector (Con) with or without nutrient deficiency. Scale bar, 20µm. Two-tailed unpaired Student's t-test. The data are represented as the means ± SEM. Two-tailed unpaired Student's t-test. The data are represented as the means ± SEM. ns indicates no significance.

3) Whether m6A defective Foxo3 mutant fails to bind to YTHDF3 should be shown.

Our response: We thank the reviewer for suggesting this important experiment. We performed electrophoretic mobility shift assays (EMSAs) using recombinant YTHDF3 proteins and biotinylated RNA probes containing different sequences of FOXO3's CDS or 3'UTR with or without m⁶A modifications. Our results showed that once m⁶A modifications were removed from the RNA probes, no matter the m⁶A location in either the CDS or 3'UTR of FOXO3, the interactions between YTHDF3 and the probes were significantly attenuated (Fig.4i, j). Furthermore, since the observed band intensity of probe 1 or 3 combined with YTHDF3 were significantly higher than those of other probes with YTHDF3 (Fig.4i, j), we reasoned the m⁶A sites at 2158nt, 2151nt, 2163nt (probe 1), and 2295nt (probe 3) along the FOXO3-CDS might be more critical for YTHDF3 to recognize FOXO3 stop codon area than other m⁶A sites.

4i

4j

Figure 13 for reviewers. (4i, j) RNA EMSA assays were performed using recombinant YTHDF3 proteins and biotinylated RNA probes containing different sequences of FOXO3's CDS (4i) or 3'UTR (4j) with or without m⁶A modifications.

4) Please describe the baseline phenotype of YTHDF3 knockout mice.

Our response: As suggested, we compared the baseline phenotype between YTHDF3^{-/-} mice and their littermate controls. We found that YTHDF3-deficient mice were comparable in size and body weight compared to their age-matched controls (Fig.S18, A, B). For the major organs, including liver, heart, spleen, lung, kidney, and brain, there was no significant difference in YTHDF3 deficient mice (Fig.S18, C, D). Furthermore, there was no difference in histological characterization of these aforementioned organs by H&E staining between controls and YTHDF3^{-/-} mice (Fig.S18, E). Thus, our data indicated that YTHDF3 deficiency did not significantly affect the baseline phenotypes of mice under normal physiological conditions. Additionally, the Mouse Genome Informatics (MGI) database also showed that there was no dramatic difference in general phenotypes between the controls and YTHDF3^{-/-} mice (<http://www.informatics.jax.org/marker/MGI:1918850>).

Figure 14 for reviewers. The baseline phenotypes of YTHDF3^{-/-} mice under normal physiological conditions. Comparison of overall size of controls and YTHDF3^{-/-} mice (at least 3 mice per group, 2-4 months old were analyzed). **(S18a, b)** The body weight of wildtype and YTHDF3^{-/-} mice. **(S18c, d)** Comparison of organs weight (i.e., liver, heart, spleen, lung, kidney and brain) from wildtype and YTHDF3^{-/-} mice (at least 3 mice per group, 2-4 months old). **(S18e)** Hematoxylin and eosin stained (H&E) evaluation of organ sections, including heart, liver, brain, lung, kidney, spleen of wildtype and YTHDF3^{-/-} mice.

5) Please provide better photos of the EM analysis in Fig. 1J.

Our response: As requested, we updated better photos in revised Fig.1m (original Fig.1J) in the revised manuscript.

Figure 15 for reviewers. (1m) Representative TEM images of autophagosomes (yellow arrow) and autolysosomes (red arrows) in wild-type and YTHDF3^{-/-} MEFs, with and without nutrient starvation. High magnification images of the boxed areas are displayed on the right-hand side. Scale bar, 1 μm.

6) in Fig. 3IK, the labeling of control and starvation is missing.

Our response: We apologize for the error. We have added the label in revised Fig.3k, m (original Fig.3, I and K).

7) In Figure 4, the authors should show that peaks for YTHDF3 and m⁶A hypermethylation occur at exactly the same place on FoxO3 mRNA.

Our response: As requested, we have shown that peaks for YTHDF3 and m⁶A hypermethylation occur at exactly the same place on FoxO3 mRNA in the revised Fig.4f.

Figure 16 for reviewers. (4f) Integrative Genomics Viewer (IGV) tracks displaying YTHDF3-RIP-seq (upper panels) and MeRIP-seq (lower panels) read distribution along the CDS and 3'UTR of FOXO3 mRNA. The squares mark increases in m⁶A peaks in MEFs upon nutrient deficiency.

8) The authors should show the level of FoxO3 protein, particularly in the nucleus in Figure 5C.

Our response: As suggested, the FoxO3 protein levels in both the nucleus and cytoplasm were

provided in the revised Fig.5f.

Figure 17 for reviewers. (5f) Immunoblot analyses of nuclear and cytoplasmic FOXO3 in FOXO3 rescued MEFs (shYTHDF3+FOXO3) and control MEFs (shYTHDF3+Con). GAPDH is used as a loading control.

Reviewer #4 (Remarks to the Author):

I was asked specifically to evaluate the proteomics portion of this manuscript, so my comments below relate to that section of the paper.

The authors performed a label-free quantitative proteomics experiment to determine how nutrient deficiency in MEFs impacts autophagy. Unfortunately, the authors did not provide sufficient detail to evaluate the validity of these results.

Our response: We sincerely apologized for not providing sufficient experimental details in the primary submitted manuscript, the missing information had been included in the revised manuscript.

The materials and methods section does not contain sufficient details; missing information is listed below.

1. There is no description of the LC-MS acquisition method, nor is there a reference to a previous report of said method.

Our response: We thank the reviewer for this comment. We have now added detailed information on the LC-MS acquisition method in the revised manuscript. Please see below and refer to Materials and Methods/LC-MS-based proteomics analysis in the revised manuscript for more details.

“The tryptic peptides were dissolved in 0.1% formic acid (solvent A), directly loaded onto a reversed-phase analytical column (Acclaim PepMap™, 15-cm length, 75 µm i.d.). The gradient was comprised of an increase from 5% to 10% solvent B (0.1% formic acid in 80% acetonitrile) over 28 min, 10% to 22% in 55 min, 22% to 30% in 27 min, and climbing to 100% in 5 min then holding at 100% for 5 min, all at a constant flow rate of 300 nL/min on an EASY-nLC 1200 UPLC system (Thermo). The peptides were subjected to NSI source followed by tandem mass spectrometry (MS/MS) in Orbitrap Fusion™ Tribrid™ LC mass spectrometer (Thermo) coupled online to the UPLC. The electrospray voltage applied was 2.3 kV. The m/z scan range was 350 to 1500 for full scan, and intact peptides were detected in the Orbitrap at a resolution of 120,000. Peptides were then selected for MS/MS using NCE setting as 30 and the fragments were detected in Ion Trap. Data were acquired in a data-dependent mode that time between master scan was set 3 sec. Automatic gain control (AGC) was set at 5E3 and the fixed first mass was set as 120 m/z.”

2. The details of the database search are incomplete. (no parent or fragment ion mass error tolerances, missing number of proteins in the database)

Our response: We thank the reviewer for pointing out these important pieces of information was incomplete. We have added the information in the revised manuscript. Please see below and also refer to Materials and Methods/ LC-MS-based proteomics analysis in the revised manuscript for more details.

“For database search, raw files were processed using MaxQuant (version 1.5.8.3) and Andromeda search engine against the mouse Uniprot database. The spectra were searched with a mass tolerance

of 6 ppm for precursors and 0.5 Da for fragment ions. The following parameters were set: ‘trypsin/P’ was set as enzyme specificity, and ‘max missed cleavages’ was set to 2. Acetylation of protein N termini and oxidation of methionine were considered as variable and carbamidomethylation of cysteine residues as fixed modification. The options ‘match between runs’, ‘decoy searches of reversed sequences’, and ‘LFQ’ were enabled. ‘LFQ min. ratio count’ was set to 2. Proteins were identified based on at least one unique peptide with a length of 6 amino acids and a maximum mass of 4600 Da. Contaminant sequences were included in the search. The false discovery rate for peptide-spectrum matches (PSM), for both the protein and site, were each set at 1%.”

Regarding the differential proteomic profile analysis upon starvation, we identified 3,318 mouse proteins from 46,573 peptides, which were obtained through mapping 445,510 MS/MS spectra to mouse Uniport database and acquiring 86,978 peptide-spectrum matches (PSMs). Regarding the YTHDF3 protein interactome analysis, we identified a total of 2,752 mouse proteins from 12,797 peptides, which were obtained by mapping 118,368 MS/MS spectra to mouse Uniport database and acquiring 18,362 PSMs.

3. The quantification and statistical analysis section is also incomplete, as is the description of quantification in Perseus. Were the P-values corrected (i.e. was multiple testing correction performed)? In what order were the samples analyzed? Were the replicates randomized for analysis?

Our response: We thank the reviewer for pointing out this important incompleteness. We have added detailed information in the revised manuscript. Please see below and also refer to Materials and Methods/ LC-MS-based proteomics analysis in the revised manuscript for more details.

“For protein quantification and statistical analyses, the MaxQuant output ‘proteinGroups.txt’ files were loaded into Perseus (version 1.6.2.3). The protein entries were filtered to exclude reverse hits, potential contaminants, and proteins only identified by site. The LFQ intensities were log₂ transformed and all missing values were imputed from a fitted normal distribution using default settings in Perseus. For the differential proteomic analysis upon nutrient starvation, P values were calculated using an unpaired two-tailed Student’s t-test (n=3, assuming equal variance) with a 95% confidence interval (P<0.05). The samples were analyzed in triplicates and random order.”

Furthermore, we didn’t perform multiple testing corrections in the differential proteomic analysis upon starvation, as it might be too strict to prevent all false-positive errors, thus possibly filtering out the truly-regulated proteins.

Minor point: I think the authors meant to say "reversed" sequences rather than "revert" sequences.

Our response: We apologize for the error. We have corrected the word.

Based on the missing details described above, some revision of this manuscript is required.

1. Lin, Z. *et al.* RNA m(6) A methylation regulates sorafenib resistance in liver cancer through FOXO3-mediated autophagy. *Embo j* **39**, e103181 (2020).
2. Zhang, S. *et al.* Altered m(6) A modification is involved in up-regulated expression of FOXO3 in luteinized granulosa cells of non-obese polycystic ovary syndrome patients. *J Cell Mol Med* **24**, 11874-11882 (2020).
3. Cook, K.B., Kazan, H., Zuberi, K., Morris, Q. & Hughes, T.R. RBPDB: a database of RNA-binding specificities. *Nucleic Acids Res* **39**, D301-8 (2011).
4. Li, J.-H., Liu, S., Zhou, H., Qu, L.-H. & Yang, J.-H. starBase v2.0: decoding miRNA-ceRNA, miRNA-ncRNA and protein-RNA interaction networks from large-scale CLIP-Seq data. *Nucleic Acids Research* **42**, D92-D97 (2014).
5. Zhao, W. *et al.* POSTAR3: an updated platform for exploring post-transcriptional regulation coordinated by RNA-binding proteins. *Nucleic Acids Res* (2021).
6. Nowosad, A. & Besson, A. CDKN1B/p27 regulates autophagy via the control of Ragulator and MTOR activity in amino acid-deprived cells. *Autophagy* **16**, 2297-2298 (2020).
7. Nowosad, A. *et al.* p27 controls Ragulator and mTOR activity in amino acid-deprived cells to regulate the autophagy-lysosomal pathway and coordinate cell cycle and cell growth. *Nature Cell Biology* **22**, 1076-1090 (2020).
8. Parmigiani, A. *et al.* Sestrins Inhibit mTORC1 Kinase Activation through the GATOR Complex. *Cell Reports* **9**, 1281-1291 (2014).
9. da Silva Rosa, S.C. *et al.* BNIP3L/Nix-induced mitochondrial fission, mitophagy, and impaired myocyte glucose uptake are abrogated by PRKA/PKA phosphorylation. *Autophagy* **17**, 2257-2272 (2020).
10. Yamaguchi, T. *et al.* The CCR4-NOT deadenylase complex controls Atg7-dependent cell death and heart function. *Sci Signal* **11**(2018).
11. Dunlop, E.A. *et al.* FLCN, a novel autophagy component, interacts with GABARAP and is regulated by ULK1 phosphorylation. *Autophagy* **10**, 1749-1760 (2014).
12. Delgado, M. & Tesfaigzi, Y. Is BMF central for anoikis and autophagy? *Autophagy* **10**, 168-169 (2013).
13. Qiao, S. *et al.* A REDD1/TXNIP pro-oxidant complex regulates ATG4B activity to control stress-induced autophagy and sustain exercise capacity. *Nature Communications* **6**(2015).
14. Vantaggiato, C. *et al.* ZFYVE26/SPASTIZIN and SPG11/SPATACSIN mutations in hereditary spastic paraplegia types AR-SPG15 and AR-SPG11 have different effects on autophagy and endocytosis. *Autophagy* **15**, 34-57 (2018).
15. Nguyen, C.H. *et al.* Translational control by RGS2. *The Journal of cell biology* **186**, 755-765 (2009).
16. Ma, T. & Klann, E. PERK: a novel therapeutic target for neurodegenerative diseases? *Alzheimer's research & therapy* **6**, 30-30 (2014).
17. Zhou, J. *et al.* N(6)-Methyladenosine Guides mRNA Alternative Translation during Integrated Stress Response. *Mol Cell* **69**, 636-647 e7 (2018).
18. Sun, H. *et al.* m(6)Am-seq reveals the dynamic m(6)Am methylation in the human transcriptome. *Nature communications* **12**, 4778-4778 (2021).
19. Selberg, S. *et al.* Discovery of Small Molecules that Activate RNA Methylation through Cooperative Binding to the METTL3-14-WTAP Complex Active Site. *Cell Rep* **26**, 3762-3771.e5 (2019).

20. Zhang, X. *et al.* Structural insights into FTO's catalytic mechanism for the demethylation of multiple RNA substrates. *Proceedings of the National Academy of Sciences* **116**, 2919 (2019).
21. Feng, C. *et al.* Crystal structures of the human RNA demethylase Alkbh5 reveal basis for substrate recognition. *The Journal of biological chemistry* **289**, 11571-11583 (2014).
22. Wang, L. *et al.* NADP modulates RNA m(6)A methylation and adipogenesis via enhancing FTO activity. *Nat Chem Biol* **16**, 1394-1402 (2020).
23. Liu, J. *et al.* A METTL3-METTL14 complex mediates mammalian nuclear RNA N6-adenosine methylation. *Nat Chem Biol* **10**, 93-5 (2014).
24. Liu, J. *et al.* m6A mRNA methylation regulates AKT activity to promote the proliferation and tumorigenicity of endometrial cancer. *Nature Cell Biology* **20**, 1074-1083 (2018).
25. Kim, J. & Lee, G. Metabolic Control of m6A RNA Modification. *Metabolites* **11**(2021).
26. Zhang, Y. *et al.* RNA-binding protein YTHDF3 suppresses interferon-dependent antiviral responses by promoting FOXO3 translation. *Proc Natl Acad Sci USA* **116**, 976-981 (2019).
27. Lin, Z. *et al.* RNA m6A methylation regulates sorafenib resistance in liver cancer through FOXO3-mediated autophagy. *The EMBO Journal* **39**(2020).
28. Mammucari, C. *et al.* FoxO3 controls autophagy in skeletal muscle in vivo. *Cell Metab* **6**, 458-71 (2007).
29. Warr, M.R. *et al.* FOXO3A directs a protective autophagy program in haematopoietic stem cells. *Nature* **494**, 323-327 (2013).
30. Wang, P., Doxtader, K.A. & Nam, Y. Structural Basis for Cooperative Function of Mettl3 and Mettl14 Methyltransferases. *Mol Cell* **63**, 306-317 (2016).

REVIEWER COMMENTS

Reviewer #1 (Remarks to the Author):

The authors thoroughly addressed my comments. The effects between YTHDF1/YTHDF3 versus YTHDF2 is quite interesting. This work will add new knowledge to the existing literature.

Reviewer #2 (Remarks to the Author):

The authors have addressed most of my concerns. I have no further suggestion.

Reviewer #3 (Remarks to the Author):

The authors added many experiments and the overall quality of the work has been improved. However, the overall significance of the work remains modest.

The authors have focused on Fox3 among many 86 peaks showing YTHDF3 RIP-seq upon nutrient deficiency without compelling reasons. The authors could have shown more lines of evidence to show that a modest increase in YTHDF3 during starvation activates autophagy primarily through FoxO3 translation. The rescue experiments to overexpress the unphysiological level of FoxO3 in Figure 5fg do not necessarily indicate the critical involvement of Foxo3. It is quite likely that YTHDF3 mediates autophagy through other molecules, which could be more critical in mediating autophagy. In addition, FoxO3 is regulated by posttranslational mechanisms and, thus, it is questionable whether translational upregulation is sufficient.

The study lacks quantifications in many results. For example, this reviewer is not convinced by the authors' argument regarding the METTL3 stabilization by starvation shown in Fig. 3 without any quantification. In addition, what is the underlying mechanism through which ubiquitination of METTL3 is decreased in response to starvation? Many molecules, such as BECN1, ATG5, BNIP3 and others are not regulated by YTHDF3 or METTL3 in Figure 5.

What is the overall significance of the proposed mechanism in autophagy activation by starvation in vivo? In addition, Fig. 3 suggests that activation of METTL3 is only transient. How does this mechanism affect the function of cells, including cell survival and metabolism?

Reviewer #4 (Remarks to the Author):

The authors satisfactorily answered my questions. While the authors appear to have designed their quantitative experiments well, I disagree strongly with their decision to NOT use FDR correction for the quantitative proteomics experiment. The argument was that this filtration would be too stringent and would remove proteins that are truly regulated. Unfortunately, the converse is also true: the methodology used would likely result in the inclusion of more false positives. With that being said, it is reasonable to perform pathway analyses without this filter in place, as some proteins that fall just below the significance filters with regard to corrected p-value and/or fold change might contribute to a significantly regulated biological pathway. For this reason, lack of p-value correction is not something that should preclude publication of this work.

I apologize for not catching this earlier, but there should be a supplementary table that lists all 955 differentially bound proteins (Uniprot accession and gene name at a minimum), number of peptides, measured fold changes, and p-values. The analysis of results from this experiment are given in Supplementary Figure 14, but no supporting data is provided. This data is present to some extent in the data repository but not in a form that is particularly user friendly.

Point-by-point responses (NCOMMS-21-11358A)

Reviewer #1 (Remarks to the Author):

The authors thoroughly addressed my comments. The effects between YTHDF1/YTHDF3 versus YTHDF2 is quite interesting. This work will add new knowledge to the existing literature.

Our response: It is highly appreciated that the reviewer acknowledged the significance of our study.

Reviewer #2 (Remarks to the Author):

The authors have addressed most of my concerns. I hve no further suggestion.

Our response: We thank the reviewer for showing her/his support in acceptance of our manuscript.

Reviewer #3 (Remarks to the Author):

The authors added many experiments and the overall quality of the work has been improved. However, the overall significance of the work remains modest.

Our response: We thank the reviewer for appreciating the improvement of our study, and we are happy to address her/his concern with more detailed responses as follows.

The authors have focused on Fox3 among many 86 peaks showing YTHDF3 RIP-seq upon nutrient deficiency without compelling reasons. The authors could have shown more lines of evidence to show that a modest increase in YTHDF3 during starvation activates autophagy primarily through FoxO3 translation.

Our response: We thank the reviewer for this kind suggestion. Among all 86 peaks, there are 7 genes annotated to GO-term Autophagy (0006914), including FOXO3, BMF, DDIT3, AP4M1, SESN2, ZFYVE1, and ZFYVE26 (**Supporting Figure 1a, see also Fig.5e in the revised manuscript**). In these genes, FOXO3 is one of the most important transcriptional factors which regulates autophagy according to the literatures¹. By performing RIP-qPCR, we verified most of these autophagy-related transcripts except ZFYVE26 were more enriched by YTHDF3 upon nutrient deficiency and their interactions were dependent on METTL3, importantly, the FOXO3 enrichment was most prominent (**Supporting Figure 1b, d, see also Fig.5i and Supplementary Fig.11a in the revised manuscript**). Furthermore, by western blotting, we found silencing YTHDF3 could reduce the expressions of FOXO3 and ZFYVE1, while YTHDF3 overexpression had an opposite effect (**Supporting Figure 1c, e, see also Fig.6a and Supplementary Fig.11b in the revised manuscript**). In contrast, BMF, DDIT3, and SESN2 did not show obviously corresponding changes as FOXO3 in both YTHDF3-deficient and YTHDF3-overexpressed MEFs (**Supporting Figure 1e, see also Supplementary Fig.11b in the revised manuscript**). Of note, due to the lack of a proper antibody targeting AP4M1 and the role of AP4M1 in autophagy was not well defined, we did not test AP4M1 expression. In addition, we examined 5 more autophagy-related genes in the list which was reported previously, including CDKN1B, PLD6, CCR4, FNIP2, and TXNIP. Our data showed silencing or overexpression of YTHDF3 had no significant effect on these

genes' expressions (**Supporting Figure 1f**, see also **Supplementary Fig.11c** in the revised manuscript). Thus, FOXO3 and ZFYVE1 were the most promising candidates mediating YTHDF3-promoted autophagy among the 86 peaks. However, as shown in a previous study that knocking down ZFYVE1 does not suppress autophagy², we therefore selected and focused on FOXO3 for further investigation.

Supporting Figure 1. Identification of FOXO3 as the candidate of YTHDF3. (a, see also **Fig.5e** in the revised manuscript) Venn diagram showing the number of overlapping up-enriched YTHDF3-binding targets and hyper-m⁶A-methylated mRNAs upon nutrient deprivation. Then the resultant 86 peaks were mapped to the gene list included in GO term autophagy (0006914) and obtained 7 genes. (b, see also **Fig.5i** in the revised manuscript) YTHDF3-RIP followed by qRT-PCR confirmed the interaction between YTHDF3 and FOXO3 mRNA in shNS and shMETTL3 MEFs upon nutrient deficiency. Two-tailed unpaired Student's t-test. Bars represent mean \pm SEM. *** $P < 0.001$. (c, see also **Fig.6a** in the revised manuscript) Immunoblot analysis of FOXO3 in shNS, shYTHDF3, control, and YTHDF3-OE MEFs, with or without nutrient starvation, respectively. (d, see also **Supplementary Fig.11a**) YTHDF3-RIP followed by qRT-PCR detected the interaction between YTHDF3 and the indicated mRNAs in shNS and shMETTL3 MEFs upon nutrient deficiency. Two-tailed unpaired Student's t-test. Bars represent mean \pm SEM. * $P < 0.05$, ** $P < 0.01$, and ns, not significant. (e, f, see also **Supplementary Fig.11b, c**) Immunoblot analyses of indicated proteins in shNS, shYTHDF3, control, and YTHDF3-OE MEFs, with or without nutrient starvation, respectively.

The rescue experiments to overexpress the unphysiological level of FoxO3 in Figure 5fg do not necessarily indicate the critical involvement of Foxo3. It is quite likely that YTHDF3 mediates autophagy through other molecules, which could be more critical in mediating autophagy.

Our response: We thank the reviewer for raising this important concern. To further determine the role of FOXO3 in YTHDF3 regulated autophagy, we knocked down FOXO3 and examined if YTHDF3-promoted autophagy flux was attenuated. Our data showed that the upregulations of LC3-II levels and p62 degradation promoted by YTHDF3 were remarkably impeded when silencing FOXO3 (**Supporting Figure 2a**, see also **Fig.6m** in the revised manuscript). Consistently, the mCherry-GFP-LC3 reporter assay showed that knocking down FOXO3 suppressed the starvation-induced autophagosome and autolysosome formations promoted by YTHDF3 overexpression (**Supporting Figure 2b, c**, see also **Fig.6n, o** in the revised manuscript). These data together with the rescue experiments of overexpressing FOXO3 (**Supporting Figure 2d-f**, see also **Fig.6h, k, l** in the revised manuscript) support that FOXO3 serves as a key target of YTHDF3 in promoting autophagy.

Supporting Figure 2. FOXO3 serves as key target of YTHDF3 in promoting autophagy. (a, see also 6m in the revised manuscript) Immunoblot analyses of FOXO3-silenced YTHDF3-OE MEFs (YTHDF3+siFOXO3) and control MEFs (YTHDF3+siNC) following nutrient starvation for the indicated time periods, with or without Baf.A1 treatment (20nM). (b, c, see also Fig.6n, o in the revised manuscript) mCherry-GFP-LC3 was transfected into FOXO3-silenced YTHDF3-OE MEFs and control MEFs, and the formation of autophagosomes (yellow) and autolysosomes (red) was examined. Scale bar, 20 μ m. (d, see also Fig.6h in the revised manuscript) Immunoblot analyses of FOXO3 rescued MEFs (shYTHDF3+FOXO3) and control MEFs (shYTHDF3+Con) following nutrient starvation for the indicated time periods, with or without Baf.A1 treatment

(20nM). (e, f, see also 6k, l in the revised manuscript) mCherry-GFP-LC3 was transfected into FOXO3 rescued and control MEFs, and the formation of autophagosomes (yellow) and autolysosomes (red) was examined. Scale bar, 20 μ m.

In addition, FoxO3 is regulated by posttranslational mechanisms and, thus, it is questionable whether translational upregulation is sufficient.

Our response: We totally agree with this argument that posttranslational modification of FoxO3 is important and our data couldn't exclude the effect of all types of posttranslational regulation on FoxO3 activity. To address the reviewer's concern partially, we examined phosphorylation, the most predominant posttranslational mechanism in regulating FOXO3 activity. It is reported the phosphorylation of FOXO3 at the S413 residue lead to transcriptional activity promotion³. Therefore, we tested whether YTHDF3 affected FOXO3 phosphorylation at this site. Our data showed p-FOXO3(S413) levels were increased in the nucleus and decreased in the cytoplasm upon nutrient starvation, in a similar manner to the total fractions (**Supporting Figure 3a, see also Fig.6a in the revised manuscript**). On the other hand, we detected a decrease in p-FOXO3(S413) levels in YTHDF3 KD cells, whereas YTHDF3 overexpression led to an opposite effect. However, the ratio of phosphorylated FOXO3 to the pan-FOXO3 was not obviously affected (**Supporting Figure 3a, see also Fig.6a in the revised manuscript**). These results suggested YTHDF3 may have a key role in regulating FOXO3 translation rather than FOXO3 posttranslational modification such as phosphorylation. We have also added such description in the "Discussion" section of the revised manuscript.

Supporting Figure 3. YTHDF3 has no effect on FoxO3 phosphorylation. (a, see also Fig.6a in the revised manuscript) Immunoblot analyses of FOXO3 and p-FOXO3 in shNS, shYTHDF3, control, and YTHDF3-OE MEFs, with or without nutrient starvation, respectively.

The study lacks quantifications in many results. For example, this reviewer is not convinced by the authors' argument regarding the METTL3 stabilization by starvation shown in Fig. 3 without any quantification.

Our response: We thank the reviewer for sharing us with this important suggestion. As required,

we have added the quantification results in Fig.3 regarding the METTL3 stabilization (**Supporting Figure 4a, b, see also Fig.4c, d in the revised manuscript**) and other experiments such as determining LC3-II and p62 levels. Please refer to our revised manuscript and figures for more details.

Supporting Figure 4. (a, see also Fig.4c in the revised manuscript) Left, MEFs with and without nutrient deficiency were treated with 20 μ M MG132 for the indicated time periods. Levels of METTL3 were examined by immunoblot analyses. GAPDH was used as a loading control. Right, relative METTL3 protein levels were quantitatively defined. **(b, see also Fig.4d in the revised manuscript)** Left, MEFs with and without nutrient deficiency were treated with 100 μ g/ml cycloheximide (CHX) for the indicated time periods. Levels of METTL3 were examined by immunoblot analyses. GAPDH was used as a loading control. Right, protein half-life of METTL3 was quantitatively defined.

In addition, what is the underlying mechanism through which ubiquitination of METTL3 is decreased in response to starvation?

Our response: We thank the reviewer for this suggestion. To identify the potential regulators interacting with METTL3 and accounted for its de-ubiquitination in response to starvation, we overexpressed FLAG-tagged METTL3 in MEFs and subjected the anti-FLAG immunoprecipitates to mass spectrometry. Surprisingly, we found 6 ubiquitination-related proteins are co-purified with METTL3, including RPS27a, RPL23, RPL11, RPS2, RPS3, and HSPA5 (**Supporting Figure 5a, see also Fig.4f in the revised manuscript**). Among these proteins, RPS2, RPS3, and HSPA5 were only reported to be ubiquitinated^{4, 5}, but no evidence indicated they had ubiquitylation regulatory roles. Therefore, we focused on RPS27a, RPL23 and RPL11 for further investigation. Interestingly, our data revealed nutrient starvation attenuated the RPS27a-METTL3 interactions, and increased the interactions of RPL11 and RPL23 with METTL3 (**Supporting Figure 5b, see also Fig.4f, g in the revised manuscript**). Moreover, RPS27a, RPL23 and RPL11 expressions were not significantly affected by starvation (**Supporting Figure 5d, see also Fig.4h in the revised manuscript**). These results indicated that the interactions of these proteins to METTL3 might affect METTL3 protein ubiquitination and stabilization. To further confirm this notion, we examined the METTL3 expressions in MEFs transfected with siRNAs targeting the RPS27a, RPL23, and RPL11, respectively. The results showed METTL3 were upregulated in siRPS27a MEFs but not significantly changed in siRPL11 and siRPL23 MEFs (**Supporting Figure 5e, see also Fig.4i in the revised manuscript**). Since RPS27a is a ubiquitin fusion protein, it can release active ubiquitin monomers, which mediate the protein ubiquitously degradation, we further examined whether RPS27a lead to METTL3 ubiquitination. Our result showed the inhibition of RPS27a strongly attenuated METTL3 ubiquitination (**Supporting Figure 5f, see also Fig.4j in the revised manuscript**). These results suggest that the impaired RPS27a-METTL3 interaction upon nutrient starvation results in METTL3 ubiquitination suppression thus increasing METTL3 stabilization.

Supporting Figure 5. Impaired RPS27a-METTL3 interaction upon nutrient starvation results in METTL3 ubiquitination suppression thus increasing METTL3 stabilization. (a, see also Fig.4f in the revised manuscript) Identification of METTL3-interacting proteins by quantitative mass spectrometry. The ubiquitination-related proteins are labeled in red. The rest of the proteins are shown in bright blue. **(b, see also Fig.4g in the revised manuscript)** Interactions between METTL3 and the indicated proteins were analyzed. **(c, see also Fig.4h in the revised manuscript)** Immunoblot analyses of RPS27a, RPL11 and RPL23 in MEFs following nutrient starvation for the indicated time periods. **(d, see also Fig.4i in the revised manuscript)** MEFs knocked down of indicated proteins were subjected to immunoblotting. **(e, see also Fig.4j in the revised manuscript)** In vivo ubiquitination assay of METTL3 in RPS27a KD and control MEFs.

Many molecules, such as BECN1, ATG5, BNIP3 and others are not regulated by YTHDF3 or METTL3 in Figure 5.

Our response: We thank the reviewer for raising this concern. By silencing or overexpression of YTHDF3, most of the FOXO3 target genes involved in autophagy including ULK1, ATG13, ATG14, PI3KIII, ATG7, ATG10, ATG5, BNIP3, and RAB7 were significantly changed at both the mRNA and protein levels. However, 4 genes (BECN1, ATG12, ATG4A and ATG4C) displayed very mild changes in protein levels, whereas significant changes were detected in their transcriptional levels (**Supporting Figure 6a, b, see also Fig.6b, c in the revised manuscript**). We assumed this might due to the difference in gene-specific and celltype-dependent gene expression regulations. Consistently, we found MEFs either silencing or overexpressing FOXO3 had significant effects on the transcription of above-mentioned genes, but had no clear effects on their protein levels (**Supporting Figure 6c, d, see also Supplementary Fig.12a, b in the revised manuscript**). In contrast, other FOXO3 targets such as ATG14, PI3KIII, ATG7, and BNIP3 were significantly changed in protein levels when manipulating FOXO3 genetically, similar to the effect of

YTHDF3(Supporting Figure 6d, see also Supplementary Fig.12b in the revised manuscript). Such discrepancies in expressions of FOXO3 targets were also detected in the MEFs silencing or overexpressing METTL3 (Supporting Figure 6e, see also Fig.6e in the revised manuscript). In addition, METTL3 might probably regulate these genes' expression via different m⁶A readers. For instance, a recent study illustrated METTL3 could induce decay of ATG7 transcript in a YTHDF2-dependent manner in senescent fibroblast-like synoviocytes (FLSs)⁶. Another study found YTHDF1 promoted BECN1 mRNA stability via recognizing the m⁶A sites within BECN1 transcripts in hepatic stellate cells (HSCs)⁷. Therefore, how METTL3 regulates different autophagy-related genes' expression and autophagic activity might be a comprehensive result via different regulatory mediators and layers, particularly emerging evidence indicated that METTL3 had other function independent of its m⁶A role⁸. In a word, YTHDF3 and METTL3 might not have exactly same impacts on expression levels of those FOXO3 target genes involved in autophagy. Nevertheless, since YTHDF3-regulated autophagy was METTL3-dependent, we observed most of the FOXO3 targets promoted by YTHDF3 overexpression were reduced when silencing METTL3, even for those unchanged genes in METTL3-silenced cells (Supporting Figure 6f, see also Supplementary Fig.12c in the revised manuscript). Above all, these data further suggested YTHDF3-promoted autophagic induction was dependent on METTL3.

Supporting Figure 6. YTHDF3-promoted autophagic induction is dependent on METTL3. (a, see also Fig.6b in the revised manuscript) qRT-PCR analysis of mRNA levels of FOXO3 target autophagy-related genes in shNS, shYTHDF3, control, and YTHDF3-OE MEFs. Two-tailed unpaired Student's t-test. Bars represent mean \pm SEM. * $P < 0.05$, ** $P < 0.01$, *** $P < 0.001$. (b, see also Fig.6c in the revised manuscript) Immunoblot analyses of protein levels of FOXO3 targeted autophagic genes in shNS, shYTHDF3, control, and YTHDF3-OE MEFs. (c, see also Supplementary Fig.12a in the revised manuscript) qRT-PCR analysis of mRNA levels of FOXO3 target autophagy-related genes in siNC, siFOXO3, control, and FOXO3-OE MEFs. Two-tailed unpaired Student's t-test. Bars represent mean \pm SEM. * $P < 0.05$, ** $P < 0.01$, *** $P < 0.001$. (d, see also Supplementary Fig.12b in the revised manuscript) Immunoblot analyses of protein levels of the indicated FOXO3 targeted autophagic genes in siNC, siFOXO3, control, and FOXO3-OE. (e, see also Fig.6e in the revised manuscript) Immunoblot analyses of protein levels of

FOXO3 targeted autophagic genes in shNS, shMETTL3, control, and METTL3-OE MEFs. MEFs. **(f, see also Supplementary Fig.12c in the revised manuscript)** Immunoblot analyses of protein levels of FOXO3 targeted autophagic genes in METTL3-silenced YTHDF3-OE MEFs.

What is the overall significance of the proposed mechanism in autophagy activation by starvation in vivo?

Our response: We thank the reviewer for pointing out this issue. It is known that under physiological conditions, autophagy has a homeostatic function in disposal of excessive protein aggregates, lipids and dysfunctional organelles in vivo. Attenuated autophagic response to starvation may lead to metabolic inflexibility. Interestingly, we observed although YTHDF3^{-/-} mice were comparable in body weight to wild-type mice under normal diet conditions, they exhibited significantly less body weight loss than their age-matched wild-type controls after a 24-hour fasting (**Supporting Figure 7a-c, see also Supplementary Fig.14a-c in the revised manuscript**). Meanwhile, we noticed that the weight of the fasted livers seemed heavier in YTHDF3^{-/-} mice than wild-type mice, although the p value was not statistically significant (**Supporting Figure 7d, e, see also Supplementary Fig.14d, e in the revised manuscript**). Indeed, YTHDF3^{-/-} mice showed more depositions of glycogen and lipid droplets within the hepatocytes than wild-type mice upon fasting (**Supporting Figure 7f-h, see also Supplementary Fig.14f-h in the revised manuscript**). These results indicate that YTHDF3 may play important roles in facilitating nutrient utilization probably by promoting autophagy. Next, we examined the expressions of LC3B-II and FOXO3 in livers derived from wild-type and YTHDF3^{-/-} mice. Compared to wild-type mice, fasted YTHDF3^{-/-} mice livers showed marked attenuated expressions in LC3B-II and FOXO3 levels (**Supporting Figure 7i, see also Supplementary Fig.14i in the revised manuscript**). Above all, these data indicate that YTHDF3-regulated autophagy might play important roles in nutrient homeostasis in vivo.

Due to limitations of observing period and detection methods, we haven't observed more apparent phenotypes in YTHDF3^{-/-} mice. However, since autophagy play important roles in maintaining metabolic homeostasis in vivo, particularly under some stressful conditions, we propose that our in vivo study might provide novel insights in understanding the role of YTHDF3 mediated autophagy in physiological and pathological metabolic rewirements.

Supporting Figure 7. The YTHDF3^{-/-} mice show less sensitivity to starvation in vivo. (a, see also **Supplementary Fig.14a** in the revised manuscript) Representative image of 24h-fasted wild-type and YTHDF3^{-/-} mice. (b, c, see also **Supplementary Fig.14b, c** in the revised manuscript) Body weights were measured in wild-type and YTHDF3^{-/-} mice before and after a 24h fast (b), and the body weight changes were compared between the groups (c). (d, see also **Supplementary Fig.14d** in the revised manuscript) Representative image of livers from the fasted wild-type and YTHDF3^{-/-} mice. (e, see also **Supplementary Fig.14e** in the revised manuscript) Liver weights in fasted wild-type and YTHDF3^{-/-} mice were measured. (f, see also **Supplementary Fig.14f** in the revised manuscript) Representative images of H&E, PAS, and Oil Red O staining of liver sections from fasted wild-type and YTHDF3^{-/-} mice. (g, h, see also **Supplementary Fig.14g, h** in the revised manuscript) The percentages of the PAS staining positive cells (g) and Oil Red O staining positive areas (h) were compared between groups. (i, see also **Supplementary Fig.14i** in the revised manuscript) Left, immunoblot analysis of FOXO3 and LC3-II in liver tissues derived from wild-type and YTHDF3^{-/-} mice. GAPDH is used as a loading control. Right, FOXO3 and LC3-II expressions were quantitatively defined, respectively. Two-tailed unpaired Student's t-test (n=6 mice per group, male, 8 weeks old). The data are represented as the means ± SEM. **P < 0.01.

In addition, Fig. 3 suggests that activation of METTL3 is only transient. How does this mechanism affect the function of cells, including cell survival and metabolism?

Our response: We thank the reviewer for this question. Autophagy is an adaptive cellular response to ensure cellular homeostasis and survival, but excessive autophagy is detrimental to life. During short-term starvation, autophagy sustains basic metabolism which is vital to live, while in long-term starvation, high levels of continuous autophagy may aggravate the cellular metabolically dysfunctional status. Interestingly, the self-compensatory mechanisms always try to avoid the occurrence of latter condition. In this study, we observed the protein abundance and catalytic activity of METTL3 were both up-regulated during nutrient deficiency (**Supporting Figure 8a, b, see also Fig.3d, f in the revised manuscript**). However, upon longer starvation, although the protein level of METTL3 remained relatively stable, its enzymatic activity was down-regulated (**Supporting Figure 8b, c, see also Fig.3f, g in the revised manuscript**). We speculated this might be a compensatory mechanism for cells under nutrients exhaustion. As noticed, the autophagy activity of MEFs is only transiently activated upon nutrient starvation. The LC3-II levels with Baf.A1 increased at 1-2-hour starvation but declined after 6 hours (**Supporting Figure 8d, see also Supplementary Fig.6a in the revised manuscript**). Consistent with this, the viability of MEFs did not change so much within 2-hour starvation, however, longer starvation caused cellular viability loss with morphological changes (**Supporting Figure 8e, f, see also Supplementary Fig.6b, c in the revised manuscript**). Furthermore, short-term starvation increased cellular ATP levels, but with longer starvation period, the ATP levels decreased apparently (**Supporting Figure 8g, see also Supplementary Fig.6d in the revised manuscript**). These results suggest MEFs within short-term starvation may be under a compensatory status that rapid autophagy can favor its living and metabolism; however, prolonged starvation may lead a decompensated status where reduced autophagy is beneficial to delay metabolic exhaustion and cell death in MEFs. Under these circumstances, controlling m⁶A might be an ideal strategy to control autophagic activity, thus maintaining cell survival and metabolism. The double safeguard mechanism, including adequate METTL3 protein maintenance and its dynamic catalytic activity fine-tuning, ensures proper m⁶A levels at different starvation periods, which is beneficial for cell metabolism and survival.

Supporting Figure 8. Transient activation of METTL3 favors cell survival and metabolism. (a, see also Fig.3d in the revised manuscript) Immunoblot analyses of nuclear fractions from MEFs following nutrient starvation for the indicated time periods. **(b, see also Fig.3f in the revised manuscript)** The relative m⁶A methylation catalytic activities of purified METTL3 from the MEFs starved for the indicated time periods were determined using an RNA probe and d₃-m⁶A. The

methylation yields were calculated based on the molar ratio of newly formed d₃-m⁶A to digested RNA probes. G was used as an internal control to calculate the amount of RNA probes. Two-tailed unpaired Student's t-test. The data are represented as the means ± SEM. *P < 0.05 and ***P < 0.001. **(c, see also Fig.3g in the revised manuscript)** Immunoblot analysis of METTL3 in MEFs following nutrient starvation for the indicated time periods. **(d, see also Supplementary Fig.6a in the revised manuscript)** Immunoblot analysis of LC3-II in MEFs following nutrient starvation for the indicated time periods with Baf.A1 treatment(20nM). GAPDH was used as a loading control. **(e, see also Supplementary Fig.6b in the revised manuscript)** Percentage of surviving cells after nutrient starvation for the indicated time periods. Two-tailed unpaired Student's t-test. The data are represented as the means ± SEM. ***P < 0.001. **(f, see also Supplementary Fig.6c in the revised manuscript)** Representative phase-contrast images of MEFs following nutrient starvation for the indicated time periods. **(g, see also Supplementary Fig.6d in the revised manuscript)** Measurement of cellular ATP levels in MEFs following nutrient starvation for the indicated time periods. Two-tailed unpaired Student's t-test. The data are represented as the means ± SEM. **P < 0.01, ***P < 0.001.

Reviewer #4 (Remarks to the Author):

The authors satisfactorily answered my questions. While the authors appear to have designed their quantitative experiments well, I disagree strongly with their decision to NOT use FDR correction for the quantitative proteomics experiment. The argument was that this filtration would be too stringent and would remove proteins that are truly regulated. Unfortunately, the converse is also true: the methodology used would likely result in the inclusion of more false positives. With that being said, it is reasonable to perform pathway analyses without this filter in place, as some proteins that fall just below the significance filters with regard to corrected p-value and/or fold change might contribute to a significantly regulated biological pathway. For this reason, lack of p-value correction is not something that should preclude publication of this work.

Our response: We thank the reviewer for his acknowledgement that excluding FDR correction might yield more false positives. Accordingly, we re-analyzed our proteomic data using the Benjamini-Hochberg (BH) corrections. Unfortunately, no protein reached the conventional 0.05 threshold when using the BH-FDR adjustments. One possible reason for this issue is label-free proteomics experiments had low power due to limited effect size and few replicates⁹, while multiple testing corrections may fail to detect true positives despite many exist candidates (e.g., YTHDF3 validated by Western blotting).

I apologize for not catching this earlier, but there should be a supplementary table that lists all 955 differentially bound proteins (Uniprot accession and gene name at a minimum), number of peptides, measured fold changes, and p-values. The analysis of results from this experiment are given in Supplementary Figure 14, but no supporting data is provided. This data is present to some extent in the data repository but not in a form that is particularly user friendly.

Our response: We thank the reviewer for this important suggestion. We would also apologize for miscalculating the difference of YTHDF3-binding proteins previously. In fact, the number of

YTHDF3 differentially bound proteins should be 1065 after careful examination when duplicate names were excluded. The data present in revised Fig. 8b and Supplementary Fig.18-21 were corrected accordingly. As requested, a supplementary table including detail information of 1065 YTHDF3 differentially bound proteins were provided as **Supplementary Table 6**. In addition, the newly identified METTL3 interacting proteins were also attached in **Supplementary Table 6**.

Reference

1. Liu L, Cheng Z. Forkhead Box O (FoxO) Transcription Factors in Autophagy, Metabolic Health, and Tissue Homeostasis. In: *Autophagy in Health and Disease: Potential Therapeutic Approaches* (ed Turksen K). Springer International Publishing (2018).
2. Axe EL, *et al.* Autophagosome formation from membrane compartments enriched in phosphatidylinositol 3-phosphate and dynamically connected to the endoplasmic reticulum. *The Journal of cell biology* **182**, 685-701 (2008).
3. Calissi G, Lam EW, Link W. Therapeutic strategies targeting FOXO transcription factors. *Nat Rev Drug Discov* **20**, 21-38 (2021).
4. Meyer C, Garzia A, Morozov P, Molina H, Tuschl T. The G3BP1-Family-USP10 Deubiquitinase Complex Rescues Ubiquitinated 40S Subunits of Ribosomes Stalled in Translation from Lysosomal Degradation. *Mol Cell* **77**, 1193-1205.e1195 (2020).
5. Kim HJ, *et al.* Crosstalk between HSPA5 arginylation and sequential ubiquitination leads to AKT degradation through autophagy flux. *Autophagy* **17**, 961-979 (2021).
6. Chen X, *et al.* METTL3-mediated m6A modification of ATG7 regulates autophagy-GATA4 axis to promote cellular senescence and osteoarthritis progression. *Annals of the Rheumatic Diseases*, (2021).
7. Shen M, *et al.* N6-methyladenosine modification regulates ferroptosis through autophagy signaling pathway in hepatic stellate cells. *Redox Biology* **47**, (2021).
8. Lin S, Choe J, Du P, Triboulet R, Gregory RI. The m(6)A Methyltransferase METTL3 Promotes Translation in Human Cancer Cells. *Mol Cell* **62**, 335-345 (2016).
9. Pascovici D, Handler DC, Wu JX, Haynes PA. Multiple testing corrections in quantitative proteomics: A useful but blunt tool. *Proteomics* **16**, 2448-2453 (2016).

REVIEWERS' COMMENTS

Reviewer #3 (Remarks to the Author):

No further concerns.